# De novo pyrimidine biosynthesis inhibition synergizes with BCL-X$_L$ targeting in pancreatic cancer

Huan Zhang[1,13], Naiara Santana-Codina [1,2,13] ✉, Qijia Yu[1,13], Clara Poupault [1], Claudia Campos[1], Xingping Qin [3,4,5], Nicole Sindoni [1], Marina Ciscar[2], Aparna Padhye[1], Miljan Kuljanin[1], Junning Wang[6], Matthew J. Dorman[1], Peter Bross [7,8], Andrew J. Aguirre [6,9], Stephanie K. Dougan [10,11], Kristopher A. Sarosiek [3,4,5,6] & Joseph D. Mancias [1,12] ✉

Oncogenic KRAS induces metabolic rewiring in pancreatic ductal adeno-carcinoma (PDAC) characterized, in part, by dependency on de novo pyrimidine biosynthesis. Pharmacologic inhibition of dihydroorotate dehydrogenase (DHODH), an enzyme in the de novo pyrimidine synthesis pathway, delays pancreatic tumor growth; however, limited monotherapy efficacy suggests that compensatory pathways may drive resistance. Here, we use an integrated metabolomic, proteomic and in vitro and in vivo DHODH inhibitor-anchored genetic screening approach to identify compensatory pathways to DHODH inhibition (DHODHi) and targets for combination therapy strategies. We demonstrate that DHODHi alters the apoptotic regulatory proteome thereby enhancing sensitivity to inhibitors of the anti-apoptotic BCL2L1 (BCL-X$_L$) protein. Co-targeting DHODH and BCL-X$_L$ synergistically induces apoptosis in PDAC cells and patient-derived organoids. The combination of DHODH inhibition with Brequinar and BCL-X$_L$ degradation by DT2216, a proteolysis targeting chimera (PROTAC), significantly inhibits PDAC tumor growth. These data define mechanisms of adaptation to DHODHi and support combination therapy targeting BCL-X$_L$ in PDAC.

Metabolic rewiring is a hallmark of cancer[1]. In pancreatic ductal adenocarcinoma (PDAC), this metabolic shift is driven by oncogenic KRAS mutation that promotes glucose use for glycosylation and synthesis of nucleotides[2,3]. Nucleotide metabolism is critical for maintaining essential pathways that support tumor growth like DNA synthesis, DNA damage repair, and glycosylation[4]. Pyrimidine nucleotides (UTP, CTP, TTP) can be obtained through two main pathways: the de novo pyrimidine synthesis pathway, which requires precursors like glutamine, aspartate, phosphoribosyl pyrophosphate, and bicarbonate to form the pyrimidine ring, and the salvage pathway, which can import or recycle

[1]Division of Radiation and Genome Stability, Department of Radiation Oncology, Dana-Farber Cancer Institute, Harvard Medical School, Boston, MA, USA. [2]Department of Biomedicine, Aarhus University, Aarhus, Denmark. [3]John B. Little Center for Radiation Sciences, Harvard T.H. Chan School of Public Health, Boston, MA, USA. [4]Laboratory of Systems Pharmacology, Harvard Program in Therapeutic Science, Department of Systems Biology, Harvard Medical School, Boston, MA, USA. [5]Molecular and Integrative Physiological Sciences Program, Harvard T.H. Chan School of Public Health, Boston, MA, USA. [6]Department of Medical Oncology, Dana-Farber Cancer Institute, Harvard Medical School, Boston, MA, USA. [7]Research Unit for Molecular Medicine, Department of Clinical Medicine, Aarhus University, Aarhus, Denmark. [8]Department of Clinical Biochemistry, Aarhus University Hospital, Aarhus, Denmark. [9]Cancer Program, Broad Institute of MIT and Harvard, Cambridge, MA, USA. [10]Department of Immunology, Harvard Medical School, Boston, MA, USA. [11]Department of Cancer Immunology and Virology, Dana-Farber Cancer Institute, Boston, MA, USA. [12]Department of Radiation Oncology, Brigham and Women's Hospital, Harvard Medical School, Boston, MA, USA. [13]These authors contributed equally: Huan Zhang, Naiara Santana-Codina, Qijia Yu. ✉e-mail: nsc@biomed.au.dk; Joseph_Mancias@dfci.harvard.edu

nucleobases and nucleosides[5]. Drugs targeting nucleotide metabolism (e.g., gemcitabine, 5-fluorouracil) are cornerstones of the two most effective chemotherapy regimens for PDAC patients (e.g., FOLFIRINOX and gemcitabine with nab-paclitaxel); however, therapeutic resistance to these chemotherapies is a significant clinical challenge.

We and others have previously identified de novo pyrimidine nucleotide synthesis as an in vivo metabolic vulnerability in PDAC[2,3,6,7], which suggested that blocking pyrimidine synthesis may be a promising therapeutic strategy for PDAC. Dihydroorotate dehydrogenase (DHODH) is an enzyme in the de novo pyrimidine biosynthesis pathway with clinically available inhibitors. DHODH is located in the inner mitochondrial membrane and couples oxidation of dihydroorotate to orotate, a critical intermediate in the synthesis of pyrimidines, with reduction of ubiquinone. In addition, pyrimidines are essential for Complex I activity by promoting PDH and tricarboxylic acid cycle activities[8]. Therefore, DHODH inhibitors (DHODHi) target pyrimidine synthesis and mitochondrial respiration, an additional essential PDAC metabolic pathway. Prior unsuccessful clinical trials evaluating de novo pyrimidine biosynthesis inhibition as an anticancer strategy likely failed due to inadequate dosing strategies that prevented consistent pyrimidine depletion[9,10] and the use of low-potency inhibitors like leflunomide (NCT02509052, NCT01611675). More recently there has been a renewed interest in repurposing Brequinar (BQ), an FDA-approved DHODH inhibitor with improved potency compared to leflunomide, as well as other newly available DHODHi as anticancer therapeutics. We previously demonstrated that DHODHi reduced pyrimidine nucleotides and decreased clonogenic ability of PDAC cells with a therapeutic index over non-tumor cells[3]. However, despite in vivo on-target depletion of pyrimidines, DHODHi demonstrated only partial responses in a xenograft PDAC model[3] suggesting intrinsic or acquired therapeutic resistance. Therapeutic resistance is one of the most difficult clinical problems in PDAC with a wide variety of genetic and non-genetic mechanisms reported[11]. Identifying and targeting non-genetic mechanisms of resistance is a challenge that requires a multidisciplinary approach encompassing identification of metabolic, transcriptomic and proteomic alterations that promote survival despite on-target inhibition[12,13].

Here, we identify pathways of adaptation to DHODHi using an integrated mass spectrometry(MS)-based quantitative temporal proteomics workflow, liquid chromatography (LC)/MS-based metabolomics and an in vitro and in vivo DHODHi-anchored CRISPR/Cas9 genetic screen approach (Fig. 1a). Proteomics and metabolomics confirm on-target inhibition of pyrimidine synthesis as well as an expected compensatory upregulation of the nucleoside salvage pathway. Using a genome-wide in vitro DHODHi-anchored screen, we identify sgRNAs targeting the anti-apoptotic gene *BCL2L1* (*BCL-X$_L$*) as a high priority synthetic lethal combinatorial hit. To account for differences in the in vitro versus in vivo metabolic milieu and prioritize candidates for translation, we perform a DHODHi-anchored mini-pool in vivo PDAC xenograft CRISPR/Cas9 screen thereby validating BCL-X$_L$ as a high priority target. We demonstrate synergistic cytotoxicity and tumor growth inhibition in PDAC cells, patient-derived organoids, and mouse models of PDAC when DHODHi are combined with a BCL-X$_L$ inhibitor (A-1331852) or degrader (DT2216). Given the growing interest in targeting de novo pyrimidine synthesis as a PDAC dependency, we propose a combination strategy of DHODH and BCL-X$_L$ inhibition that can induce anti-tumor responses with potential clinical translatability.

## Results

### Defining adaptation to DHODHi by metabolomics and proteomics

To define mechanisms of adaptation to DHODH inhibition, we initially tested sensitivity to Brequinar (BQ) in a panel of human and murine PDAC cell lines (Supplementary Fig. 1a). Half maximal inhibitory concentration (IC50) measurements showed a range of responses

between 0.5–5 µM. To assess long-term responses that allow for analysis of adaptation at later time points, we analyzed doubling events in PaTu-8988T and PaTu-8902 cells treated with BQ for 12 days (Supplementary Fig. 1b). Despite slower proliferation, PDAC cells maintained their proliferative capacity even at higher doses of BQ. Based on the varying levels of sensitivity in each cell line, we selected 0.5 and 5 µM as a low and high dose, respectively, for subsequent experiments.

First, we analyzed the metabolic response to BQ acutely (24 h) or long-term (7 days) by targeted LC-MS/MS metabolomics (Supplementary Data 1). In agreement with previous studies[3], BQ efficiently blocked pyrimidine synthesis (Fig. 1b) with an accumulation of nucleotide precursors (Aspartate, N-Carbamoyl-Aspartate, Dihydroorotate) and a decrease in pyrimidine nucleotides (UMP, UDP, UTP, dTTP, CMP, CTP). Importantly, pyrimidine synthesis was consistently downregulated long-term, indicating adaptation to DHODHi was not due to loss of on-target DHODH inhibition and pathway reactivation. Long-term BQ increased metabolites in the nucleoside salvage pathway including uracil and uridine (Supplementary Fig. 1c), suggesting compensatory activation of nucleoside import under pyrimidine depletion conditions[14]. DHODH inhibition can affect mitochondrial respiration directly via disruption of ubiquinone reduction and indirectly by modulating PDH activity and the TCA cycle[3,8,15]. Short and long-term BQ treatments similarly suggest sustained disruption in mitochondrial metabolism as demonstrated by modulation of TCA cycle intermediates (Supplementary Fig. 1d). As measured by Seahorse, BQ impaired oxygen consumption rate (OCR) at 24 h but to a lesser extent than complex I inhibition with IACS-010759 (Supplementary Fig. 1e). Long-term BQ treatment demonstrated more profound inhibitory effects suggesting resistance to DHODHi is not related to reactivation of respiration in these cells. (Supplementary Fig. 1e–j) Finally, in agreement with the role of DHODH in redox balance[16], BQ decreased the reduced:oxidized glutathione ratio (GSH/GSSG) at higher doses (Fig. 1c). Overall, our data confirms sustained inhibition of DHODH-mediated metabolism within the range of selected BQ doses and timepoints.

We then used multiplexed isobaric tag-based quantitative MS-based proteomics[12,13] to determine the global temporal proteomic response to BQ (Supplementary Data 2). Principal component analysis (PCA) showed consistent sample clustering and a time-dependent shift in PC1 and PC2 (Supplementary Fig. 2a, b). Overall, we identified and quantified ~8000 proteins in PaTu-8988T and PaTu-8902 cells with a higher magnitude of changes in protein expression at 7 days compared to 24 h (Fig. 1d, e, Supplementary Fig. 2c, d). Among the consistently downregulated proteins, multiple ribonucleoproteins were decreased both acutely (Fig. 1d, Supplementary Fig. 2c) and long-term (Fig. 1e, Supplementary Fig. 2d), consistent with a role for DHODH in ribosome biogenesis[17]. In agreement with recent studies[18], we also identified an induction in HLA-I and proteins involved in antigen presentation (Supplementary Fig. 2d). DHODH inhibition upregulated expression of nucleoside importers like SLC29A1 in PaTu-8988T cells (Fig. 1d, e), confirming our metabolomics studies. Finally, we also identified a compensatory upregulation in GPX1, likely related to an increase in oxidative stress suggested by metabolomics (Fig. 1e).

We then performed gene set enrichment analysis (GSEA) as a strategy to identify pathways activated in response to long-term treatment with BQ (Fig. 1f, Supplementary Fig. 2e, Supplementary Data 3)). Functional pathway analysis identified downregulated enrichment map nodes related to glycolysis, mTOR pathway, inflammation, and Toll-like receptor (TLR)-related innate immunity, among others. Interestingly, upregulated nodes correlated with pathways associated with metabolic adaptations like mitochondrial translation, electron transport chain, TCA cycle, and fatty acid oxidation/sphingolipid metabolism, suggesting a compensatory response involving multiple mitochondrial-related processes (Fig. 1f, Supplementary Fig. 2e). Next, to identify drugs with similar or inverse patterns of response to DHODHi that may be useful to overcome DHODHi-

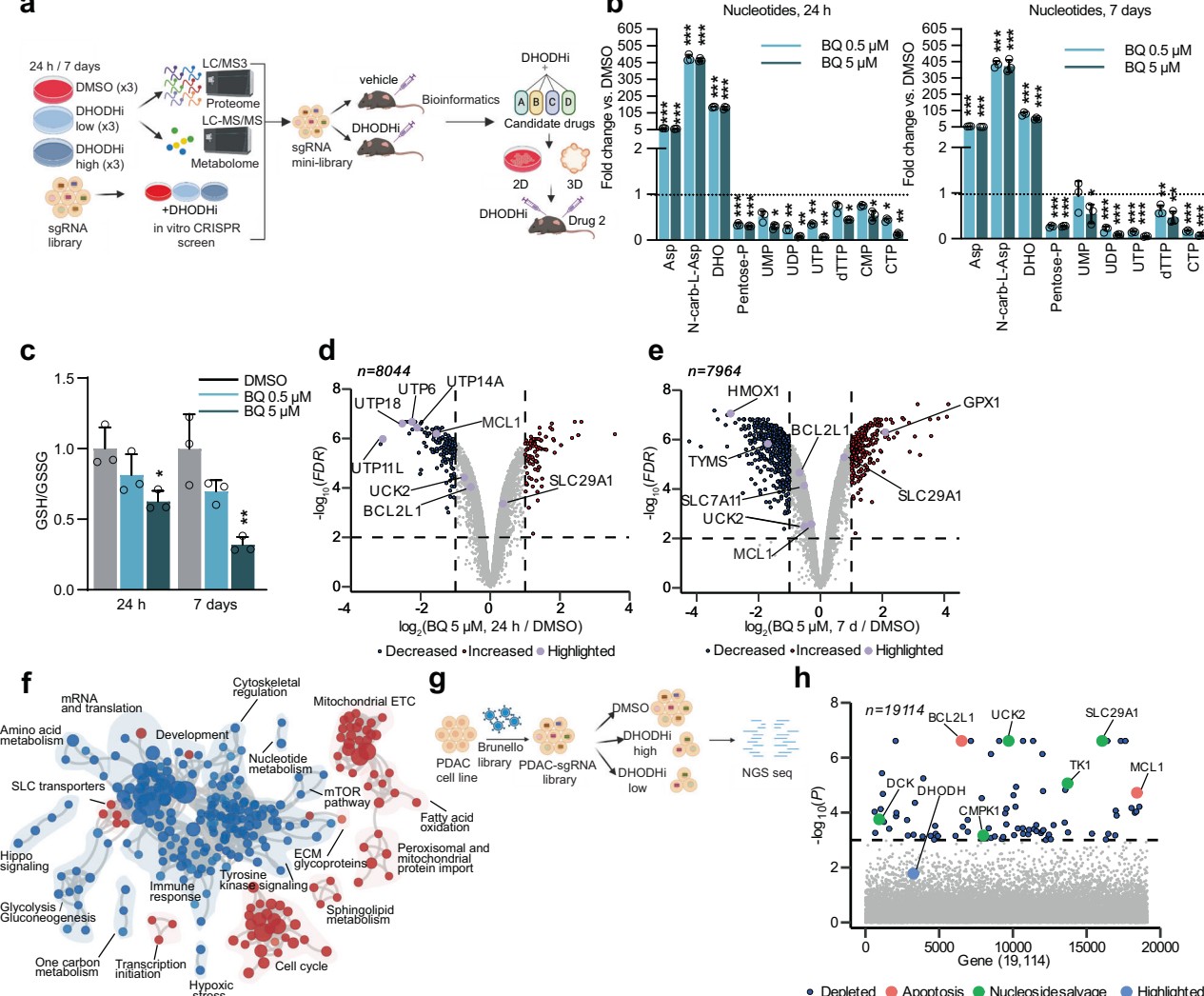

**Fig. 1 | Defining mechanisms of short- and long-term adaptation to DHODH inhibition in PDAC by an integrated multi-omics approach. a** Multi-omic workflow used to define adaptation to DHODH inhibition, see Methods and text for additional details (Santana-Codina, N. (2025) https://BioRender.com/a62aees). **b** Fold change of metabolites in the pyrimidine synthesis pathway (left: 24 h, right: 7 days). Asp: aspartate; N-Carb-L-Asp: N-Carbamoyl-L-Aspartate; DHO: dihydroorotate; UMP/UDP/UTP, uridine mono/di/triphosphate; CTP, cytidine triphosphate; dTTP, deoxythymidine triphosphate. Data are shown as mean values with error bars representing s.d. of $n = 3$ independent plates. Significance determined with t-test (unpaired, two-tailed) for cells treated with BQ vs. vehicle (*$p < 0.05$, **$p < 0.01$, ***$p < 0.001$). **c** Ratio of reduced-to-oxidized glutathione (GSH/GSSG) in PaTu-8988T cells. Ratios normalized to each DMSO condition and represented as fold change of 24 h and 7 day BQ-treated cells (0.5 and 5 μM), $n = 3$ independent plates. Significance determined with t-test (unpaired, two-tailed) for BQ vs. vehicle (*$p < 0.05$, **$p < 0.01$). **d, e** Volcano plot of protein abundance in PaTu-8988T cells treated with 5 μM BQ at 24 h (**d**) and 7 days (**e**). Plots display $-\log_{10}$ (FDR) versus

$\log_2$ relative protein abundance of mean BQ to DMSO-treated samples. Red circles: $\log_2$ fold change ≥ 1, FDR < 0.01; blue circles: $\log_2$ fold change ≤ −1, FDR < 0.01; data from 3 DMSO or 3 BQ-treated independent plates. **f** Enrichment map of gene set enrichment analysis (GSEA) of BQ-proteome (PaTu-8988T, 7 days, 0.5 μM). FDR < 0.01, Jaccard coefficient>0.25, node size related to number of components within a gene set, line width proportional to overlap between related gene sets. GSEA terms associated with upregulated (red), and downregulated (blue) proteins colored accordingly and grouped into nodes with associated terms. **g** BQ-anchored whole-genome in vitro screen in PaTu-8988T cells lentivirally transduced with the Brunello library (19,114 genes/76,441 sgRNAs), treated with BQ (0.5 or 5 μM, 2 weeks) followed by next-generation sequencing (Santana-Codina, N. (2025) https://BioRender.com/4lblqps). **h** Manhattan plot for depleted hits in PaTu-8988T (5 μM BQ) including significant genes ($-\log_{10}(P)$≥3) in BQ *vs.* DMSO (blue dots), apoptosis genes (orange), nucleoside salvage pathway genes (green). All source data including *p* values are provided as Source Data file.

associated adaptations, we used the Connectivity Map database[19], a resource we used in the past to define combinatorial strategies with KRAS[G12C] inhibitors[12]. Here, we identified similar patterns of response between BQ and bromodomain, HDAC, IMPDH, topoisomerase inhibition or HIF activation (Supplementary Fig. 2f).

**DHODHi-anchored genetic screens define synthetic lethalities**
To identify synthetic lethal interactions not predicted by metabolomics or proteomics, we performed a BQ-anchored CRISPR/Cas9 genome-wide[20] screen in PaTu-8988T cells (Fig. 1g, Supplementary

Data 4–6). Next-generation sequencing followed by MAGeCK/STARS analysis highlighted genes with depleted sgRNAs in the BQ-treated conditions, suggesting combinatorial targets for enhanced BQ sensitivity and genes with enriched sgRNAs that may promote further resistance to BQ. Gene Ontology (GO) analysis of the top 100 genes with depleted sgRNAs overlapped with pathways identified by metabolomics and proteomics, including nucleoside salvage and regulation of steroid metabolism (Supplementary Fig. 3a). Furthermore, GO also identified terms not appreciated in the proteome and metabolome analyses such as "positive regulation of apoptosis intrinsic pathways"

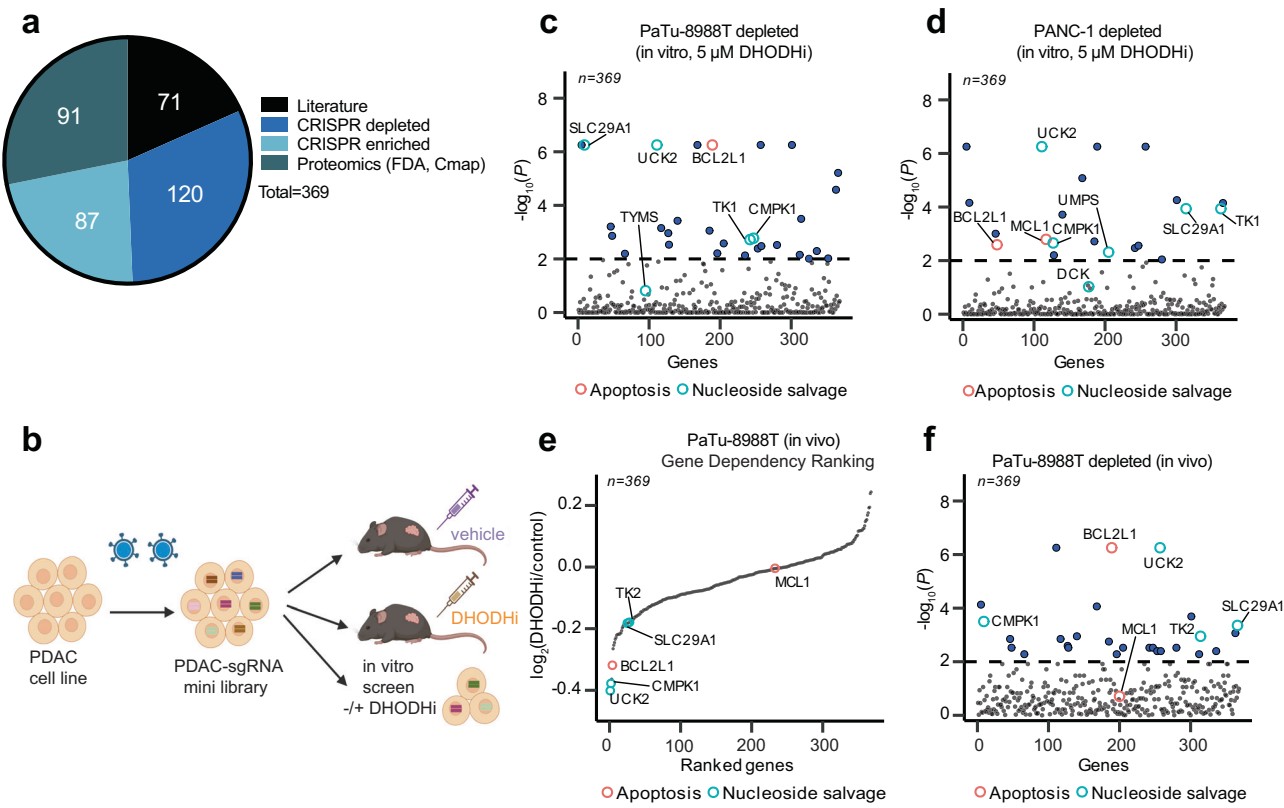

**Fig. 2 | Targeted in vivo CRISPR/Cas9 loss-of-function screen identifies synthetic lethalities with BQ in PDAC. a** A customized CRISPR/Cas9 mini-pool library (369 genes, 4 sgRNAs per gene, 528 control sgRNA) was generated prioritizing proteomic hits with FDA-approved available drugs, CRISPR depleted/enriched hits, and genes related to DHODH in published in vitro studies. **b** Schematic for in vivo and in vitro CRISPR screens with customized mini-pool library (Santana-Codina, N. (2025) https://BioRender.com/t12d46e). PDAC cells were infected with the mini-pool library and implanted in a xenograft NOG mouse model (PaTu-8988T, $n = 10$ mice/arm). Tumor-bearing mice (80–100 mm³) were treated with either vehicle or BQ, harvested 14 days later and analyzed by NGS. PaTu-8988T and PANC-1 cells containing the sgRNA mini-library were also evaluated in in vitro screens to confirm the whole genome screen results and directly contrast to in vivo results. **c, d** Manhattan plot for depleted hits in a mini-pool library in vitro CRISPR/Cas9 screen in PaTu-8988T (**c**) and PANC-1 (**d**) cells at 5 µM BQ. Blue dots highlight significant genes ($-\log_{10}(P) \geq 2$) in BQ versus DMSO, orange circles highlight genes related to apoptosis and cyan circles highlight genes in the nucleoside salvage pathway. **e** Rank-ordered graph of $\log_2$ (BQ/control) for each gene in PaTu-8988T tumors. **f** Manhattan plot for depleted hits in PaTu-8988T tumors. Blue dots highlight significant genes ($-\log_{10}(P) \geq 2$) in BQ versus vehicle, orange circles highlight genes related to apoptosis and cyan circles highlight genes in the nucleoside salvage pathway. Source data are provided as a Source Data file.

(Supplementary Fig. 3a). Among the significant depleted sgRNAs in the low-dose BQ screen (Supplementary Fig. 3b) were those targeting DHODH; however, these sgRNAs were not depleted in the high-dose BQ screen, (Fig. 1h) consistent with sub-maximal inhibition of BQ in the low-dose condition. Analysis of the sgRNAs consistently depleted at low and high doses of BQ (Fig. 1h, Supplementary Fig. 3b) confirmed nucleoside synthesis/salvage pathways (*UCK2, TK1, SLC29A1, CMPK1*) as synthetic lethalities, consistent with a compensatory upregulation in nucleoside salvage. The other most significantly depleted hits included the anti-apoptotic genes *BCL2L1* (*BCL-X_L*) and *MCL1*. Analysis of the top enriched sgRNAs (sgRNAs conferring resistance to DHODHi) identified multiple genes encoding mitochondrial complex subunits (Supplementary Fig. 3c, d) such as *NDUFA6, NDUFA1, NDUFS2, NDUFV1, NDUFA11* among others, which suggests Complex I inhibition may increase resistance to BQ.

The relative cellular dependency on de novo pyrimidine biosynthesis versus nucleoside salvage can shift depending on uridine availability[21,22] and uridine levels are different in cell culture versus in vivo conditions. To account for the physiological differences in uridine levels as well as any other differences between in vitro culture conditions and in vivo conditions, we designed a focused sgRNA library to evaluate high priority hits in a BQ-anchored mini-pool in vivo CRISPR/Cas9 screen. Our custom library contained sgRNAs targeting

369 genes (Fig. 2a) that we prioritized based on the following criteria: 1) proteomic hits with FDA drugs available, 2) most significant hits from the CRISPR screen, including depleted and enriched guides (for balancing the library), and 3) genes that had already been linked to DHODH biology in published in vitro studies but that had not yet been evaluated in in vivo studies. We first confirmed the performance of our custom library and correlation with the whole-genome screen data by testing the mini-library in vitro in PaTu-8988T and PANC−1 cells treated at low and high concentrations of BQ (Fig. 2b, Supplementary Data 7). Consistently, genes in nucleoside synthesis/salvage (*SLC29A1, UCK2, TK1, CMPK1*) were depleted across cell lines and doses (Fig. 2c, d, Supplementary Fig. 4a, b). We further confirmed anti-apoptotic *BCL2L1* as a depleted gene in all cell lines. Next, PaTu-8988T cells infected with our custom library were implanted subcutaneously in the flanks of NOG mice (NOD.Cg-*Prkdc^scid Il2rg^tm1Sug*/JicTac) and treated with vehicle or BQ 50 mg/kg per day for 14 days (Supplementary Data 7). The top depleted hits were in agreement with some of the in vitro dependencies, and highlighted genes in pyrimidine salvage (*UCK2, SLC29A1, CMPK1*) and apoptosis (*BCL2L1*) pathways (Fig. 2e, f) confirming the overlap between in vitro and in vivo metabolic dependencies for our top hits (Supplementary Fig. 4c, d). However, *MCL1* was no longer confirmed as a hit in the in vivo screen, suggesting a selective dependency on *BCL2L1* (BCL-X_L) and not on other anti-apoptotic proteins

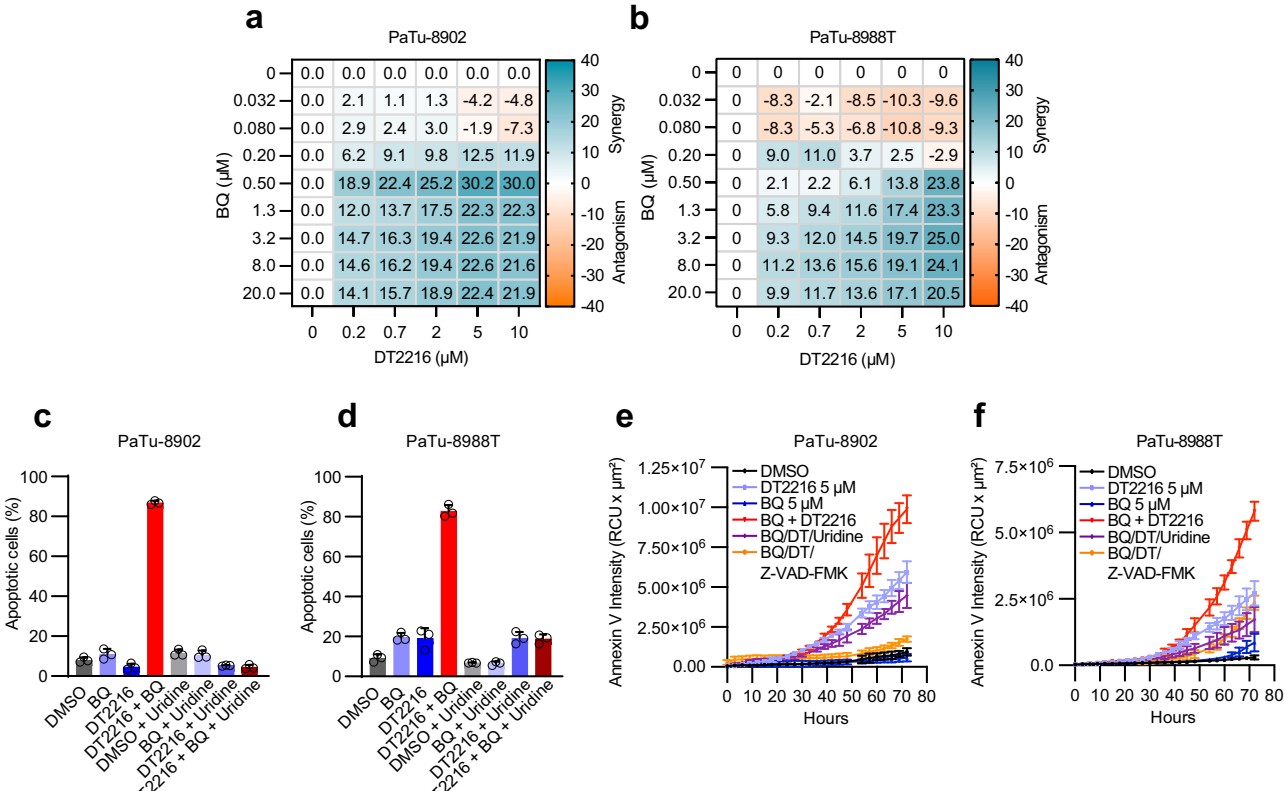

**Fig. 3 | Combination of BQ and DT2216 synergistically induces apoptosis in PDAC cell lines. a, b** Synergy score heatmaps of combination treatment with BQ and DT2216 in PaTu-8902 and PaTu-8988T cells. Synergy score between the two drugs was calculated using the HSA model implemented in SynergyFinder (antagonism: <−10; additive effect: from −10 to 10; synergistic effect: >10). Experiments performed in biological triplicate, data shown as mean HSA score of three technical replicates of one representative experiment. **c, d** Apoptosis induction was assessed using flow cytometry with FITC Annexin V staining in PaTu-8902 and PaTu-8988T cells following treatment with DMSO, DT2216 (2 μM), or BQ (5 μM), either alone or in combination, with or without 100 μM uridine for 72 hours. The percentage of apoptotic cells was determined as the sum of Annexin V-positive and Annexin V/ propidium iodide double-positive populations, representing early and late apoptosis, respectively. Error bars represent s.d. of three technical replicates, representative of three independent experiments. **e, f** Real-time accumulation of Annexin V fluorescence in PaTu-8902 and PaTu-8988T cells was monitored using the Incucyte system. Cells were labeled with Annexin V Red Dye and treated with the indicated concentrations of BQ or DT2216 alone or in combination or BQ + DT2216 with uridine (100 μM) or Z-VAD-FMK (50 μM) for 72 hours. (RCU: Red Calibrated Unit). Data are shown as mean values with error bars representing the s.d. from three technical replicates, representative of two independent experiments. Source data are provided as a Source Data file.

in vivo. Overall, our combinatorial platform identified vulnerabilities to target in combinatorial strategies and suggests that inhibiting nucleoside salvage or anti-apoptotic proteins may synergize with DHODH inhibitors.

## Co-targeting DHODH and BCL-X$_L$ in PDAC induces apoptosis

Our CRISPR/Cas9 screens nominated inhibition of *BCL2L1*, which encodes for BCL-X$_L$, as a target capable of inducing synthetic lethality in cells treated with BQ. Using CRISPR/Cas9 gene editing, we first generated PaTu-8902 *BCL2L1* knockout cells (Supplementary Fig. 5a). BCL-X$_L$ KO cells demonstrated no decrease in proliferation in comparison to sgControl cells (Supplementary Fig. 5b). However, as predicted by the CRISPR/Cas9 screens, BCL-X$_L$ KO cells were more sensitive to BQ with a decrease in proliferation and an increase in Annexin V staining and cleaved PARP in comparison to sgControl cells treated with BQ (Supplementary Fig. 5c–e). Next, we evaluated pharmacologic combination targeting of DHODH and BCL-X$_L$ by using BQ and DT2216[23], a BCL-X$_L$ proteolysis targeting chimera (PROTAC) that is currently being tested in Phase I clinical trials (NCT04886622). This PROTAC approach to targeting BCL-X$_L$ is particularly promising as it can spare platelet toxicity associated with BCL-X$_L$ inhibitors[23]. Combination treatment with BQ and DT2216 demonstrated synergy in a panel of human and murine PDAC cell lines (Fig. 3a, b, Supplementary

Fig. 6a–e), which was replicated using a distinct DHODH inhibitor, BAY-2402234 (Supplementary Fig. 6f, g). We then measured cell death by PI and Annexin V staining after 3 days of treatment (Fig. 3c, d) and real-time tracking of Annexin V intensity accumulation (Fig. 3e, f). Combination BQ and DT2216 increased apoptosis in human and murine PDAC cell lines compared to control and monotherapy and the effects were rescued by high-dose uridine and inhibition of apoptosis with Z-VAD-FMK (Fig. 3c–f, Supplementary Fig. 6h, i). To assess the potential of this combination in more physiological media preparations, we measured the relative growth of PaTu-8902 cells in Plasmax media that contains 3 μM uridine which more closely matches levels in human plasma in comparison to DMEM media preparations (Supplementary Fig. 6j–m). Consistent with previous studies, the anti-proliferative effects of BQ and BAY-2402234 were less striking in closer-to-physiological conditions. Nevertheless, BQ or BAY-2402234 and DT2216 resulted in more growth inhibition than single treatments (Supplementary Fig. 6j–m), suggesting that targeting de novo pyrimidine biosynthesis under physiologic uridine microenvironmental conditions still potentiates to BCL-X$_L$ targeting and may maintain combination efficacy in vivo.

We further investigated if co-dependency with BCL-X$_L$ was selective to targeting DHODH or is more broadly co-dependent with targeting pyrimidine biosynthesis or potentially nucleotide synthesis in general. Targeting UMPS, an enzyme downstream of DHODH in the de

novo pyrimidine biosynthesis pathway, with pyrazofurin recapitulated the effects of DHODHi as it decreased growth of PDAC cells (Supplementary Fig. 7a, b) without evident induction of cell death (Supplementary Fig. 7c, d). Remarkably, UMPSi synergized with DT2216 in growth suppression and Annexin V induction (Supplementary Fig. 7a–d), suggesting sensitization to BCL-X$_L$ targeting is not specific to DHODH inhibition alone but rather is an effect of pyrimidine synthesis pathway inhibition more broadly. Next, we assessed if the synergy with BCL-X$_L$ degraders was specific to pyrimidine biosynthesis inhibition or if it was a general consequence of disrupting nucleotide pools. To assess this, we evaluated the effects of AG-2037, a purine synthesis inhibitor targeting GART (phosphoribosylglycinamide formyltransferase), a trifunctional enzyme that catalyzes multiple steps in the purine synthesis pathway[24]. AG-2037 alone decreased proliferation of PDAC cells (Supplementary Fig. 7e, f) but did not increase Annexin V staining (Supplementary Fig. 7g, h). This data is in agreement with previous studies using in vivo CRISPR screens that defined purine and pyrimidine synthesis as essential pathways for PDAC growth[6,7]. Combination treatment of AG-2037 and DT2216 demonstrated synergistic effects on growth (Supplementary Fig. 7e, f) and Annexin V staining (Supplementary Fig. 7g, h) in two human PDAC cell lines, suggesting that co-dependency with DT2216 was a more general consequence of disrupting nucleotide pools.

Finally, as DHODH plays a role in maintaining mitochondrial respiration and complex I activity, we also explored whether co-dependency with BCL-XL may be related to the disruption of mitochondrial respiration by DHODH[8]. IACS-010759, a complex I inhibitor, in combination with DT2216 showed no additive effects on growth or Annexin V staining (Supplementary Fig. 7i-l) suggesting DHODHi effects on mitochondrial respiration are not responsible for sensitization to BCL-X$_L$ inhibition. Together with our studies using UMPSi and purine synthesis inhibitors, we conclude that co-dependency observed between DHODH and BCL-X$_L$ targeting is more likely related to depletion of nucleotide pools and not to impaired mitochondrial respiration.

## BQ modulates the PDAC apoptosis regulatory proteome

Previous studies have shown that BCL-X$_L$ expression increased from the preinvasive pancreatic intraepithelial (PanIN) stage to invasive PDAC in mouse models and patient tumors[25]. Importantly, compared to other anti-apoptotic proteins, higher levels of BCL-X$_L$ were observed in 90% of tumors from PDAC patients[26]. Monotherapy targeting of BCL-X$_L$ has modest effects in solid tumors; however, combinations of BCL-X$_L$ with gemcitabine and nab-paclitaxel or MEK inhibition have been reported to enhance PDAC anti-tumor efficacy[27,28]. Our data suggests DHODH inhibition sensitizes PDAC to apoptosis inducers targeting BCL-X$_L$. As the apoptotic cascade is a complex process regulated by multiple BCL-2 family proteins, we investigated how DHODH may rewire the apoptosis pathway in PDAC. To assess if BQ modulates the overall apoptotic priming as well as dependencies on specific pro-survival proteins in PDAC cells, we performed flow cytometry-based BH3 profiling[29] (Fig. 4a, b). At basal levels, PaTu-8988T was more primed for apoptosis than PaTu-8902, as shown by greater cytochrome c release in response to pro-apoptotic BIM, BID, and PUMA BH3 peptides[30,31]. Although BQ did not alter the priming of either cell line, BQ increased dependency on BCL-X$_L$ and BCL2 in both cell lines as indicated by the increased response to BAD and HRK peptides.

An increase in mitochondrial membrane potential is considered an early event in apoptosis that may be independent from caspase activation, precedes outer mitochondrial membrane disruption[32], and is associated with increased ROS production. As DHODHi decrease GSH/GSSG ratio (Fig. 1c), indicative of a decrease in antioxidant capacity, we next measured effects of DHODHi on mitochondrial membrane potential by Tetramethylrhodamine methyl ester (TMRM)

staining in PDAC cells. DHODHi induced an increase in mitochondrial membrane potential (Supplementary Fig. 8a, b), suggesting that DHODHi-induced mitochondrial dysfunction and hyperpolarization may sensitize PDAC cells to apoptosis inducers. Interestingly, BCL-X$_L$ is known to prevent changes in mitochondrial membrane potential in response to stimuli[32,33].

To understand if BQ changes the dependency on different pro-apoptotic proteins by modulation of their levels, we quantified expression of the BCL-2 family proteins after BQ treatment. Proteomic quantification of proteins in the apoptotic pathways demonstrated BQ decreased expression of anti-apoptotic proteins (BCL-X$_L$, MCL1) while increasing expression of pro-apoptotic proteins BAX, BAK, BID, NOXA and PUMA (Supplementary Fig. 8c, d). Western blot analyses confirmed the pro-apoptotic shift induced by BQ with decreases in anti-apoptotic proteins (BCL-X$_L$, MCL1, BCL2) and increased levels of cleaved-PARP and pro-apoptotic proteins BIM and PUMA (Fig. 4c). Treatment with a UMPSi also modulated expression of BCL-2 family proteins in a similar manner (Supplementary Fig. 8e). However, purine synthesis inhibition did not recapitulate the shift in BCL2 family proteins observed with pyrimidine synthesis inhibitors (Supplementary Fig. 8f), suggesting that even though synergy with BCL-X$_L$ targeting might be common to nucleotide deprivation in PDAC, the mechanism of apoptosis sensitization differs between blocking purine and pyrimidine metabolism.

We next evaluated the BCL-2 protein profile induced by DT2216 alone (Fig. 4d, Supplementary Fig. 8g) or in combination with BQ (Fig. 4e, Supplementary Fig. 8h). DT2216 efficiently degraded BCL-X$_L$ in the range of doses tested (0.1-5 μM) in human and murine PDAC cells. Of note, BCL-X$_L$ degradation induced compensatory increases in MCL1 and BCL2 (Fig. 4d, Supplementary Fig. 8g). Finally, combination of BQ and DT2216 significantly increased PARP cleavage and BIM compared to single treatments (Fig. 4e, Supplementary Fig. 8h) and these effects were rescued by exogenous uridine (Supplementary Fig. 8i).

To determine if BQ-induced changes at the protein level correlated with transcriptional changes in apoptosis-related genes, we performed qRT-PCR (Fig. 4f). Consistent with protein-level expression changes, PDAC cells treated with BQ downregulated expression of anti-apoptotic genes (BCL2L1 (BCL-X$_L$), MCL1 and BCL2) and upregulated expression of pro-apoptotic genes (BCL2L11 (BIM)). These data demonstrate that the pro-apoptotic shift induced by DHODHi is regulated at the transcriptional level. We next investigated whether DHODH inhibition has been previously linked to transcription factor pathways that regulate apoptosis genes. Prior research demonstrated that DHODH inhibition in the liver of mice downregulated NF-κB activity, a known pro-survival transcription factor[34,35]. Similarly, patients with Miller Syndrome, caused by mutation of DHODH, have decreased NF-κB pathway activity. NF-κB is a pro-survival effector that increases expression of anti-apoptotic genes, including BCL-X$_L$, MCL1, and BCL2[36]. Conversely, downregulation of NF-κB is associated with increased sensitivity to apoptosis[37].

Interestingly, our GSEA proteome analysis of BQ-treated PDAC lines demonstrated significant downregulation of pathways associated with inflammation and TLR-related innate immunity (Fig. 1f, Supplementary Fig. 2e, Supplementary Data 3), both of which are regulated by NF-κB transcription factor activity. To directly evaluate NF-κB pathway activity in response to DHODHi, we analyzed phosphorylation of p65 and p105, key proteins in the canonical and non-canonical NF-κB pathways, respectively. PDAC cells treated with DHODHi demonstrated a decrease in phosphorylation of both p65 and p105, consistent with decreased activation of the NF-κB pathway (Fig. 4g). Based on these findings, we propose a model in which DHODHi suppresses NF-κB activity, leading to a pro-apoptotic transcriptional response. This transcriptional shift downregulates anti-apoptotic genes, such as BCL-X$_L$, and upregulates pro-apoptotic genes, sensitizing PDAC cells to BCL-X$_L$ inhibition and apoptosis induction. Furthermore, DHODHi

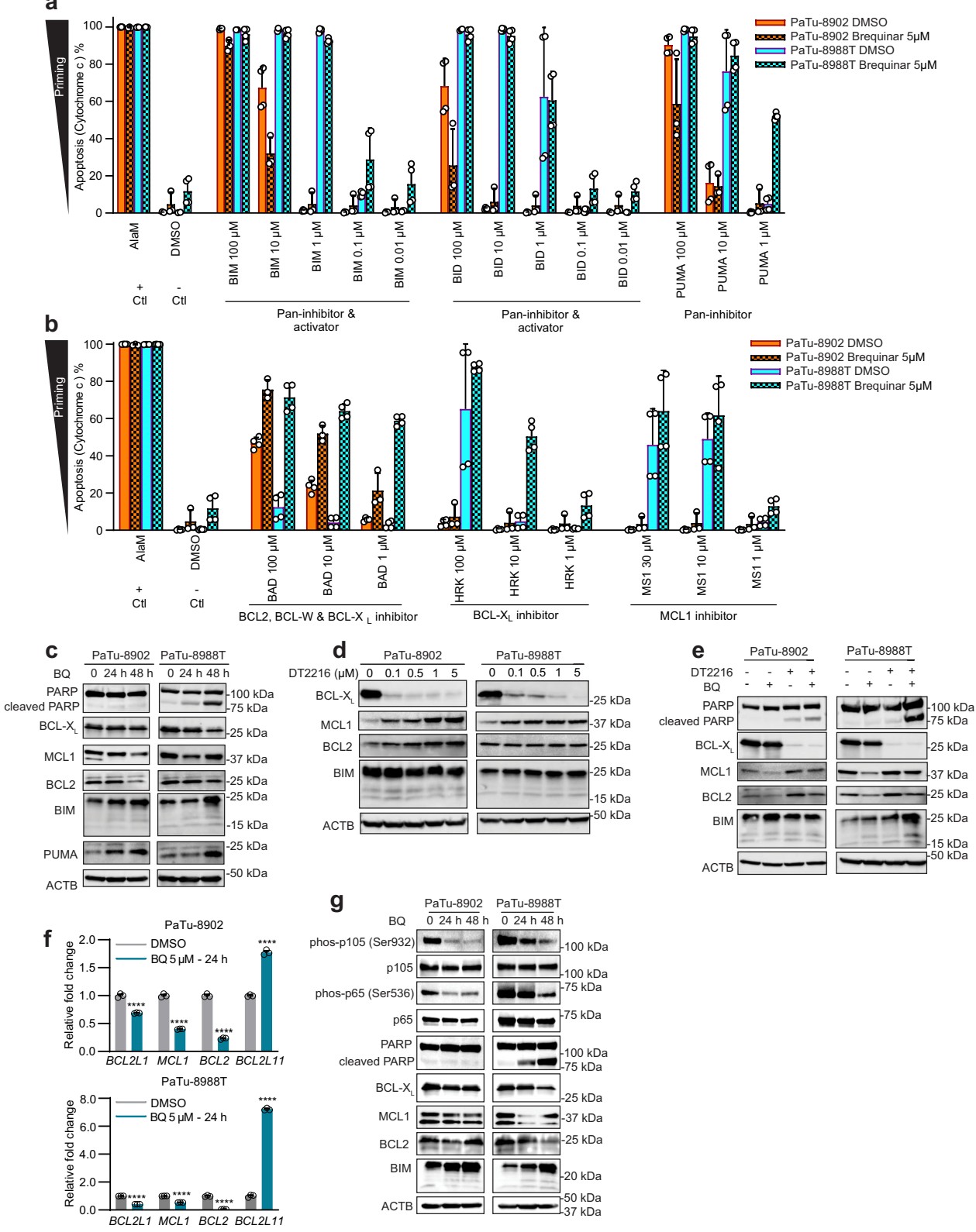

## BQ synergizes selectively with BCL-X_L inhibition

Given the effects of BQ on multiple BCL-2 family proteins and identification of *MCL1* sgRNAs as depleted in the in vitro CRISPR/Cas9 screen, we assessed if combination with other BCL-2 protein family inhibitors similarly synergized with BQ. Cell viability assays confirmed synergistic effects of BQ and BCL-X_L inhibition (DT2216 or A-1331852) using either DT2216 (Fig. 5a, d) or BQ as an anchor (Supplementary Fig. 9a–d). Interestingly, inhibitors against other BCL-2 family proteins such as AZD-5591, an MCL1 inhibitor (Fig. 5b, e, g, h) or venetoclax, a BCL2 inhibitor (Fig. 5c, f, g, h), demonstrated no synergy

directly or indirectly via downregulation of BCL-X_L induces mitochondrial hyperpolarization, potentiating sensitivity to apoptosis induction.

**Fig. 4 | BQ modulates levels of BCL-2 family proteins and enhances sensitivity to BCL-X$_L$ PROTAC-based degradation in PDAC. a**, **b** Flow cytometry-based BH3 profiling of PaTu-8902 and PaTu-8988T cells treated with BQ for 7 days shows priming (**a**) and specificity of BCL-2 protein family apoptotic dependency (**b**). Apoptosis is measured by Cytochrome c release (y-axis). Alamethicin (AlaM) and DMSO were used as positive and negative control for apoptosis induction, respectively. Experimental data sets (**a**, **b**) were derived from the same experiments and analyzed using same ALaM and DMSO controls. Conditions: BIM, BID and PUMA BH3 peptides inhibit all anti-apoptotic BCL-2 family proteins (BIM and BID also directly activate BAX and BAK), BAD BH3 peptide inhibits BCL2, BCL-W and BCL-X$_L$, HRK and MS1 peptides inhibit BCL-X$_L$ and MCL1, respectively. Data are presented as mean relative Cytochrome c release of two biologically independent experiments each with technical duplicate measurements, $n = 4$ (3 measurements for PaTu-8902 BQ 5 μM include two duplicates from one experiment and one

measurement from the second experiment). Error bars represent s.d. of all measurements. **c**, **d** Immunoblot analysis of BCL-2 family proteins in lysates from PaTu-8902 and PaTu-8988T cells treated with 5 μM BQ for 24 and 48 hours (**c**) or the indicated concentrations of DT2216 for 16 hours (**d**). **e** Immunoblot analysis of BCL-2 family proteins in lysates from PaTu-8902 and PaTu-8988T cells treated with 5 μM BQ or 5 μM DT2216 alone or BQ in combination with DT2216 for 24 hours. **f** Apoptosis regulatory gene expression assessed by qRT-PCR in PaTu-8902 and PaTu-8988T cells treated with 5 μM BQ, 24 hours. Expression levels normalized to *GAPDH* and presented as mean ± s.d. of 3 independent replicates (representative of three independent experiments). Significance was determined by t-test, **** $p < 0.0001$ (*BCL2L1* encodes BCL-X$_L$; *BCL2L11* encodes BIM). **g** Immunoblot analysis of p65 (RelA subunit of NF-κB) and p105 (precursor of the NF-κB p50 subunit) activation in lysates from PaTu-8902 and PaTu-8988T cells treated for 24 and 48 hours with BQ (5 μM). Source data are provided as a Source Data file.

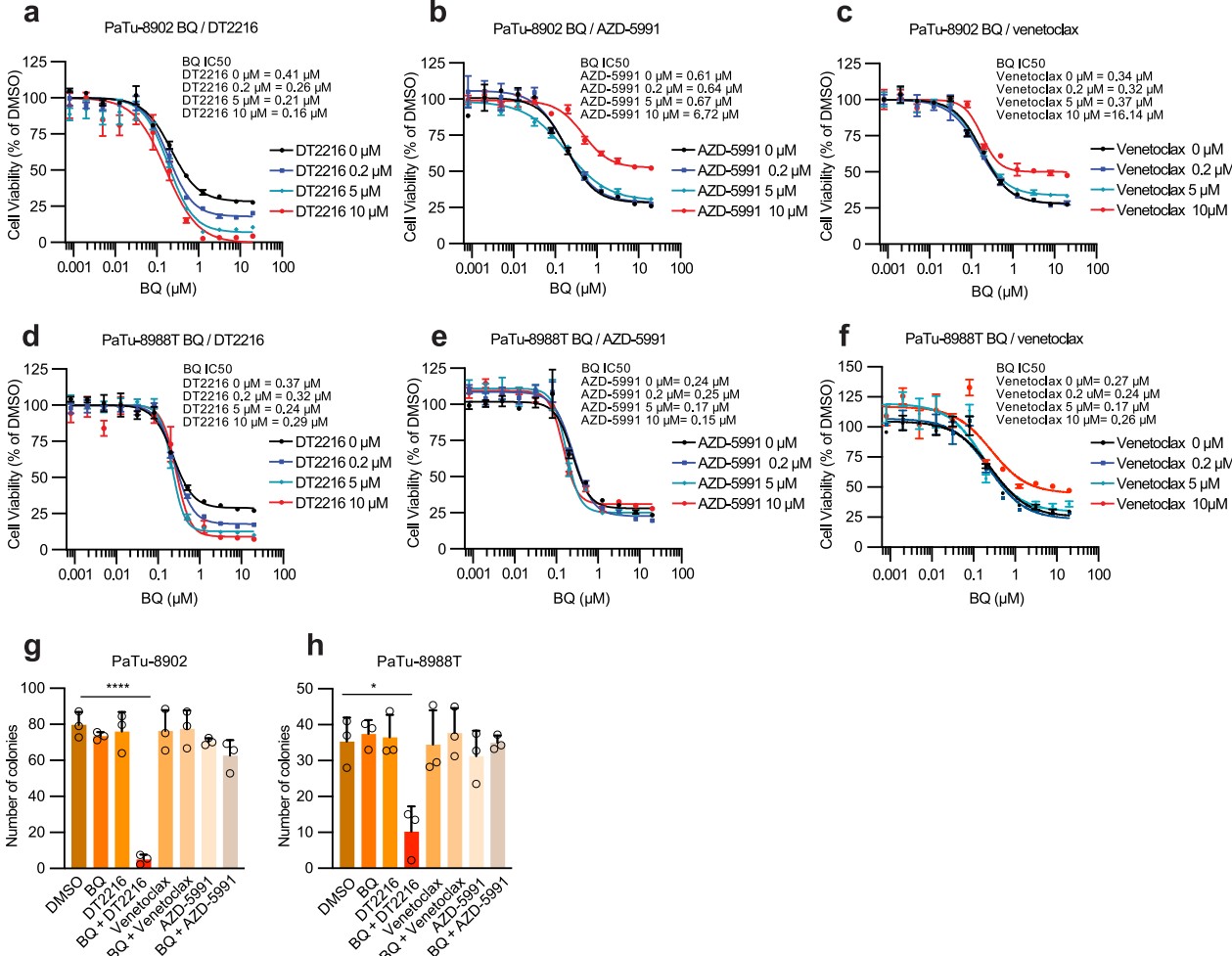

**Fig. 5 | Synergistic effects with BQ are specific to BCL-X$_L$ inhibition in PDAC. a–f** Percentage cell viability of PaTu-8902 and PaTu-8988T cells after treatment with increasing concentrations of BQ with DT2216 (BCL-X$_L$ PROTAC), AZD-5991 (MCL1 inhibition) or Venetoclax (BCL2 inhibition) for 5 days. IC50 values are shown for a representative experiment out of three independent experiments. Error bars represent s.d. of two-three technical replicates (two technical replicates for BQ alone conditions). **g**, **h** Clonogenic growth of PaTu-8902 and PaTu-8988T cells

treated with BQ (5 μM), DT2216 (5 μM), Venetoclax (5 μM), AZD-5991 (5 μM) or DT2216, Venetoclax or AZD-5991 in combination with BQ. Data are shown as mean values with error bars representing the s.d. of average of three independent experiments (each in three technical replicates). Significance was determined with *t* test (unpaired, two-tailed). *$p < 0.05$, ****$p < 0.0001$. Source data are provided as a Source Data file.

with BQ and in fact demonstrated protective effects at high concentrations. The discordance between BQ synergizing with *MCL1* genetic ablation in the CRISPR/Cas9 screen but not MCL1 pharmacologic inhibition may reflect that pharmacologic targeting of the MCL1 BH3-binding domain does not inhibit non-apoptosis-related functions of MCL1, such as maintenance of mitochondrial dynamics and

bioenergetics, whereas genetic ablation targets all MCL1 activities[38]. Overall, our data suggest the BQ-induced pro-apoptotic shift induces sensitivity to BCL-X$_L$ inhibition and not other BCL-2 family members. In fact, *BCL-X$_L$* is a strong PDAC dependency compared with *MCL1* and *BCL2* across 45 PDAC cell lines in the Cancer Dependency Map (DepMap), a library of genome-wide CRISPR/Cas9 loss-of-function screens[39]

(Supplementary Fig. 9e). Querying DepMap highlighted PaTu-8902 as the cell line with highest level of BCL-X$_L$ expression and dependency on BCL-X$_L$ across the 6 human PDAC lines we tested (Supplementary Fig. 9f), which correlated with the basal BCL-X$_L$ level detected by immune blot (Supplementary Fig. 9g) and the synergistic response to BQ and DT2216 (Fig. 3). Our results demonstrate that BQ promotes synergistic responses with BCL-X$_L$ inhibitors and not inhibitors of BCL2 or MCL1, likely due to BQ further increasing dependency on BCL-X$_L$, a critical mediator of PDAC cell survival[40].

## Co-targeting DHODH and BCL-X$_L$ inhibits PDAC tumor growth

To bridge the gap between our results in established PDAC monolayer culture systems and the potential in vivo effects of this combination, we tested the combination of BQ and DT2216 in a collection of patient-derived PDAC organoids that can more closely reflect in vivo tumor phenotypes and response to therapy (Supplementary Data 8). Drug sensitivity and synergy in patient-derived organoids was consistent with our results in established PDAC cell culture models (Fig. 6a-d), suggesting that combination targeting of DHODH and BCL-X$_L$ may inhibit tumor growth in tumor models of PDAC. Interestingly, three of the organoids used were KRAS$^{G12D}$ mutant but a priori resistant to pharmacologic KRAS$^{G12D}$ inhibition suggesting BQ and DT2216 may be useful even for KRAS inhibitor resistant tumors[41]. Indeed, a KRAS$^{G12D}$-mutant PDAC cell line exposed to escalating doses of MRTX1133 to the point of resistance, maintained sensitivity to the combination of DHODH and BCL-X$_L$ inhibition (Supplementary Fig. 9h).

Next, to assess the potential for tumor-specific targeting of BCL-X$_L$ in vivo, we implanted sgControl (non-targeting) and sgBCL2L1 (BCL-X$_L$ KO) PaTu-8902 cells in NOG mice and treated with vehicle or BQ. BCL-X$_L$ knockout tumors grew slightly slower compared to non-targeting tumors (Fig. 6e, f). BQ-treated BCL-X$_L$ KO tumor bearing mice demonstrated significant tumor growth inhibition compared to BQ-treated sgControl tumor bearing mice (Fig. 6e, f). Importantly, mice maintained stable body weight throughout treatment (Supplementary Fig. 10a). Additionally, BQ treatment in mice with BCL-X$_L$ KO tumors did not lead to significant anemia or thrombocytopenia (Supplementary Data 9).

To further assess combination DHODH and BCL-X$_L$ targeting as a therapeutic strategy for PDAC, we first evaluated for maximum tolerated doses of the combination of BQ and DT2216 in non-tumor bearing animal models. We noted that full dose BQ in combination with published regimens of DT2216 was associated with dose-limiting hematologic toxicity, in line with both DHODH inhibition and BCL-X$_L$ inhibition having effects on the bone marrow. We established that DT2216 at 15 mg/kg two times per week effectively degraded BCL-X$_L$ in the pancreas and liver (Supplementary Fig. 10b) and combination with BQ at 10 mg/kg three times per week was a tolerable regimen (Supplementary Fig. 10c). We next investigated the efficacy of BQ and DT2216 using PaTu-8902 and HPAC cell line-derived xenograft tumor models (Fig. 6g–i). DT2216 alone had minimal effects on tumor growth in vivo whereas BQ had some monotherapy efficacy in PaTu-8902 but none in HPAC xenografts. However, the combination of BQ and DT2216 led to a significant inhibition of tumor growth (Fig. 6g–i), including in HPAC xenografts where there was no single agent efficacy. Next, we examined the anti-tumor activity of the combination in an allograft KPCY C57BL/6 mouse that allows for evaluation in an immune competent setting. The poorly immunogenic murine PDAC cell line KPCY 6694 C2[42] was implanted subcutaneously (Fig. 6j, k) and allowed to establish tumors (75 mm³) prior to dosing. Similar to HPAC there was minimal single agent efficacy but a significant growth delay with the combination. To model the pancreatic microenvironment where nutrient supply may be differential compared to the subcutaneous environment, we orthotopically implanted KPCY 6694 C2 cells[43] and allowed tumors to establish prior to dosing (Fig. 6l). The BQ and DT2216 combination significantly suppressed the growth of

orthotopic KPCY 6694 C2 tumors (Fig. 6m). The combination treatment was well tolerated in the subcutaneous tumor mouse models (Supplementary Fig. 10d–f), with no significant change in mouse body weight after 3 weeks whereas, we observed a modest weight drop in the orthotopic tumor model (Supplementary Fig. 10g). While hematologic parameters were largely unaffected in immunocomprimised mouse models, the BQ and DT2216 combination demonstrated modest anemia and thrombocytopenia in C57BL/6 mice (Supplementary Data 10, 11). To evaluate potential toxicity of the combination beyond effects on hematolorecal parameters, we performed serum-based evaluations of the C57BL/6 subcutaneous syngenic model. There were no significant elevations in serum ALT in the combination treatment arm suggesting no effect on liver function (Supplementary Fig. 11a). While serum BUN levels were mildly elevated in the combination-treated group, they remained within the normal range, indicating no significant effect on kidney function (Supplementary Fig. 11b). Finally, there were no significant changes in serum Brain Natriuretic Peptide (BNP) levels, a marker of heart failure and myocardial stress, observed in any of the treatment groups (Supplementary Fig. 11c). Together, this toxicity profile is in line with other standard of care cytotoxic chemotherapy regimens, nevertheless, we discuss below potential strategies to mitigate toxicity. To confirm the on-target effects of DT2216, we evaluated BCL-X$_L$ degradation in tumors by immunoblotting. DT2216 decreased tumor BCL-X$_L$ levels in xenograft models and to a lesser degree in syngeneic models (Supplementary Fig. 11d–k), which may, in part, explain reduced tumor growth inhibition in the syngeneic experiments. Finally, we found higher levels of cleaved caspase 3 in PaTu-8902 flank tumors, indicating the drug combination induces apoptosis in vivo (Supplementary Fig. 11l). Overall, we propose combinatorial targeting of DHODH and BCL-X$_L$ as a therapeutic strategy with potential clinical translatability to restrict PDAC growth.

## Discussion

There is renewed interest in repurposing DHODHi as anti-cancer agents given the importance of de novo pyrimidine biosynthesis as a cancer dependency. While DHODHi failed as a monotherapy in previous phase I clinical trials for solid tumors[9,44], an important caveat of these studies was the use of intermittent dosing patterns, similar to cytotoxic chemotherapy regimens, that allow for recovery of pyrimidine pools and may be insufficient to consistently deplete pyrimidines in the tumor[4]. New trials are currently evaluating the efficacy of more potent DHODHi in hematological malignancies (NCT04609826 and NCT02509052) given their ability to induce differentiation of diverse AML subtypes[45,46]. Targeting DHODH in solid tumors is also a promising strategy; however, this will likely rely on the identification of combinatorial strategies to improve monotherapy responses.

DHODH inhibition has been previously linked with apoptosis induction in different types of cancers including hematological malignancies[47], colorectal cancer[48], neuroblastoma[49], renal cell carcinoma[50] and glioma[51,52]. For example, silencing of DHODH or DHODH inhibition by high-dose BQ sensitized the small cell lung cancer U1690 cell line to TRAIL-induced apoptosis[53]. Replication stress induced by DHODH inhibition has also been shown to contribute to its role in apoptosis induction[40]. However, in pancreatic cancer it is unclear how DHODH inhibition can regulate apoptosis sensitivity.

We propose a model in which pyrimidine synthesis inhibition suppresses NF-κB activity, leading to a pro-apoptotic transcriptional response. This transcriptional shift downregulates anti-apoptotic genes, such as *BCL-X$_L$*, and upregulates pro-apoptotic genes, sensitizing PDAC cells to BCL-X$_L$ degradation and promoting apoptosis. A similar profile of anti-apoptotic protein alteration was previously demonstrated in response to gemcitabine suggesting a conserved response to nucleotide metabolism targeting[32]. Functionally, alteration in anti-apoptotic proteins in PDAC, whether by DHODH inhibition

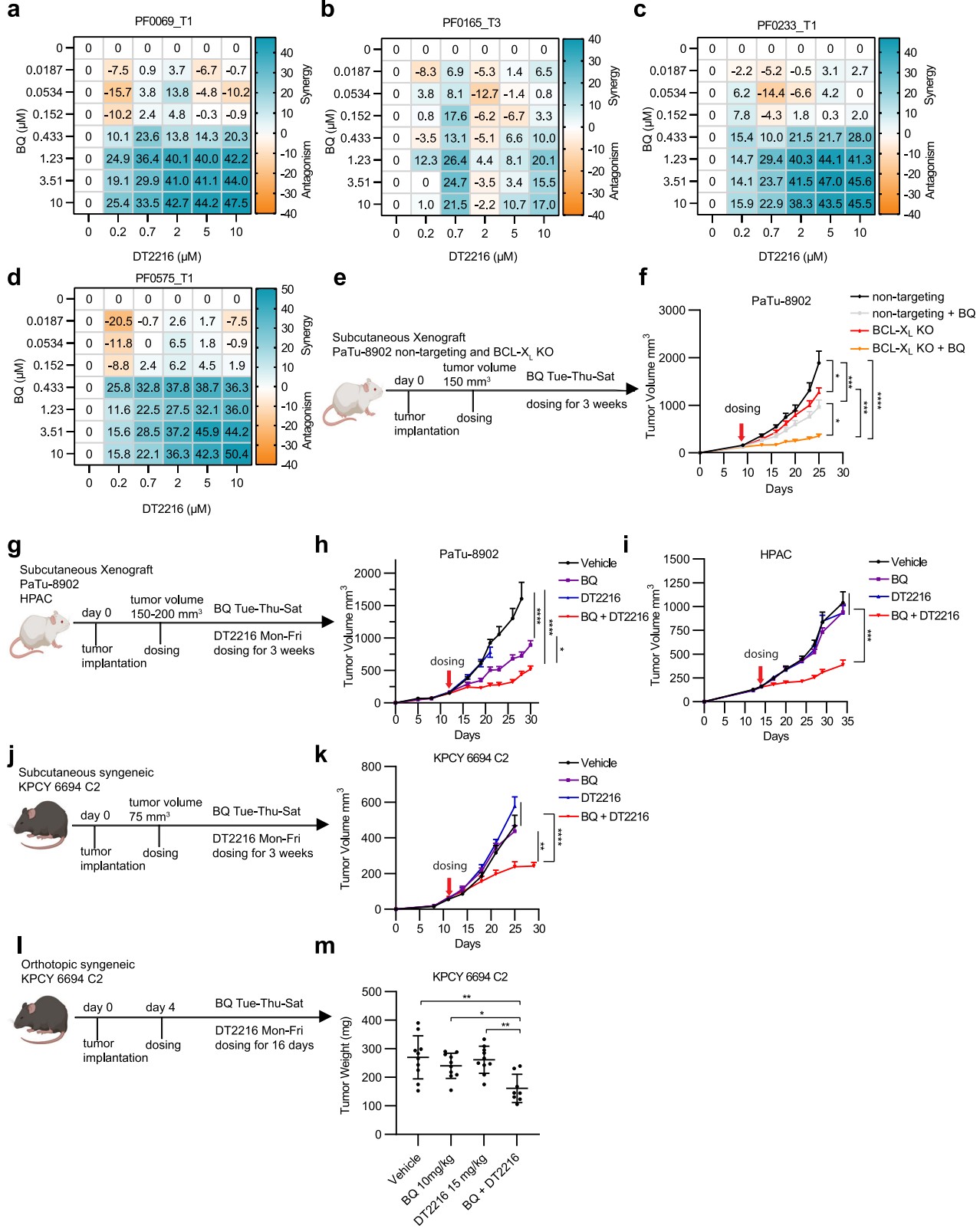

or gemcitabine, conferred increased sensitivity specifically to BCL-X$_L$ targeting. Given DHODHi is also associated with an increase in mitochondrial membrane potential, it is possible that in addition to altering NF-κB pathway activity to increase apoptosis sensitivity, DHODHi also primes cells for apoptosis by inducing mitochondrial dysfunction linked to ROS accumulation and mitochondrial hyperpolarization. Whether the increase in mitochondrial membrane potential is due to

direct DHODHi effects on mitochondrial respiration and antioxidant function or it is a downstream consequence of modulation in BCL-X$_L$[54] will require further investigation.

The specificity of enhanced sensitivity to BCL-X$_L$ targeting conferred by DHODH inhibition in PDAC appears be ingrained in PDAC tumor physiology at baseline. Previous studies have shown increased expression of BCL-X$_L$ with the progression of pancreatic neoplasia

**Fig. 6 | DT2216 increases the antitumor efficacy of BQ in patient-derived PDAC organoids and PDAC mouse models. a–d** Synergy score heatmaps based on cell viability data from patient-derived PDAC organoids treated with DT2216 and BQ. Synergy score calculated using the HSA model in SynergyFinder. Experiments were repeated two independent times, and data are shown as mean HSA score of three technical replicates from one experiment. **e** Experimental design of (**f**). **f** Tumor volume of PaTu-8902 BCL-X$_L$ KO tumors (non-targeting $n = 7$, non-targeting + BQ $n = 7$, BCL-X$_L$ KO $n = 8$, BCL-X$_L$ KO + BQ $n = 9$ per arm) after treatment with vehicle or 10 mg/kg BQ three times a week for 3 weeks. Error bars represent s.e.m. Statistical significance determined by ordinary one-way ANOVA test. *$p < 0.05$, ***$p < 0.001$, ****$p < 0.0001$ ($p = 0.024$, non-targeting vs. BCL-X$_L$ KO; $p = 0.020$, non-targeting + BQ vs. BCL-X$_L$ KO + BQ; $p = 0.0007$, non-targeting vs. non-targeting + BQ; $p = 0.0002$, BCL-X$_L$ KO vs. BCL-X$_L$ KO + BQ). **g** Experimental design of PaTu-8902 and HPAC xenograft flank tumor studies. **h, i** Tumor growth curves from PaTu-8902 (**h**, $n = 13$ per arm) and HPAC (**i**, $n = 7$ per arm) xenograft mouse models treated with vehicle, BQ, DT2216, or combination for 3 weeks. Error bars represent s.e.m. Statistical significance determined by one-way ANOVA. **h** *$p = 0.014$, BQ vs BQ + DT2216; ****$p < 0.0001$. **i** ***$p < 0.001$ ($p = 0.0003$, BQ vs BQ + DT2216; $p = 0.0004$, DT2216 vs BQ + DT2216; $p < 0.0001$, Vehicle vs BQ + DT2216. **j** Experimental design of KPCY 6694 C2 syngeneic flank allograft study in (**k**). **k** Tumor growth curves of KPCY 6694 C2 syngeneic flank model treated with vehicle ($n = 12$), BQ ($n = 13$), DT2216 ($n = 11$) or combination ($n = 11$) for 3 weeks. For flank-based tumor in vivo studies, tumors were measured twice a week. Error bars represent s.e.m. Statistical significance determined by one-way ANOVA (**$p = 0.0034$: BQ. vs. BQ + DT2216; ****$p < 0.0001$: Vehicle or DT2216 vs. BQ + DT2216). **l** Experimental design of KPCY 6694 C2 syngeneic orthotopic tumor study. **m** Tumor weight of KPCY 6694 C2 orthotopic tumors ($n = 10$ per arm, $n = 8$ for BQ + DT2216 group) after treatment with vehicle, BQ, DT2216, or combination for 16 days. Error bars represent ± s.e.m. Statistical significance determined by one-way ANOVA test, *$p < 0.05$, **$p < 0.01$ ($p = 0.025$, BQ vs BQ + DT2216; $p = 0.001$, Vehicle vs. BQ + DT2216; $p = 0.003$, DT2216 vs. BQ + DT2216). (Mancias, J. (2025) https://BioRender.com/i4zww2l). Source data provided as Source Data file.

from pancreatic intraepithelial neoplasia (PanIN)-1 to PDAC, while MCL1 slightly increased and BCL2 did not change[25]. Furthermore, BCL-X$_L$ but not MCL1 or BCL2 scored as a PDAC cell line dependency in DepMap data (Supplementary Fig. 9e). The importance of BCL-X$_L$ in PDAC physiology was further highlighted by Stanger and colleagues who demonstrated that BCL-X$_L$ enforces a slow-cycling state allowing survival of quiescent PDAC cells present in the austere nutrient and oxygen deprived PDAC tumor microenvironment[31]. These findings suggest an extra-reliance on BCL-X$_L$ in PDAC in comparison to MCL1 and BCL2 and may in part explain the selective synergy of DHODH inhibition with BCL-X$_L$ inhibition. Future studies should evaluate the potential of this combination in other types of tumors accounting for a differential reliance on other anti-apoptotic dependencies.

Our DHODH inhibition multi-omic data also demonstrate that activation of nucleoside salvage is a compensatory adaptation to DHODHi and suggests a combination therapeutic strategy. However, our studies present some limitations. First, our initial screens were performed in standard culture conditions, which differ in the amount of uridine and other metabolites available in comparison to the in vivo tumor metabolic milieu. Uridine levels in tumor interstitial fluid of subcutaneous and orthotopic PDAC tumors can be as high as 30–50 μM (10 times more than plasma concentrations). In fact, activation of nucleoside salvage is likely a contributor to the poor anti-tumor effects of monotherapy DHODHi in the uridine rich TME[55]. Indeed, exogenous uridine rescues growth defects in DHODHi-treated cells cultured in standard media conditions[21,22,56] and partially rescues our combinatorial treatment (Fig. 3). In addition, recent studies suggest glucose levels dictate apoptosis induction downstream of pyrimidine synthesis blockade[57]. To account for differences in metabolite availability in standard DMEM versus in vivo[55], cells can be cultured in media that mimics the physiological metabolic conditions of adult human plasma (e.g. Plasmax, HPLM) or the PDAC TME. Prior studies have shown that cancer cells grown in HPLM not only utilize uridine for nucleoside salvage, but they also suppress de novo pyrimidine synthesis by uric acid-mediated inhibition of UMPS activity, decreasing sensitivity to pyrimidine synthesis inhibitors like 5-FU[22]. To account for these known differences in standard DMEM media preparations, we performed a secondary CRISPR/Cas9 screen in vivo in the context of a fully formed tumor. Interestingly, both our in vitro and in vivo CRISPR screens highlighted the nucleoside salvage pathway as a co-dependency with DHODH inhibition. Despite likely differences in the relative usage of de novo pyrimidine biosynthesis versus nucleoside salvage pathways in vitro versus in vivo at baseline, DHODH inhibition in vivo still demonstrated synergy with nucleoside salvage. This suggests that despite the high levels of uridine in the TME there remains a metabolic consequence of DHODH inhibition in vivo in PDAC tumors, namely pushing tumors to rely more heavily on nucleoside salvage. One therapeutic strategy suggested by dual reliance on pyrimidine synthesis and

salvage pathways in PDAC is combination targeting of pyrimidine synthesis and salvage. Indeed, combination BQ and gemcitabine has been described as a potential combinatorial strategy in PDAC with tolerable in vivo toxicity[58]. Our CRISPR screen identified SLC29A1, a nucleoside importer, as an in vivo co-dependency, in agreement with previous studies showing synergistic in vitro efficacy[14]. However, to date there are no potent, selective, and in vivo capable inhibitors available for SLC29A1. Future studies are required to assess the feasibility of this combination since systemic dual targeting of nucleoside import and de novo pyrimidine synthesis may result in systemic toxicities affecting cells with high biosynthetic needs in the bone marrow and intestine as well as most non-tumor cells that mostly rely on salvage.

A recent report suggested DHODH acts as a ferroptosis defense mechanism by mitigating mitochondrial lipid peroxidation; however, the doses of Brequinar (500 μM) used in that study are associated with off-target inhibition of FSP1, an NAD(P)H-dependent oxidoreductase that is a critical cellular ferroptosis defense node[59,60]. In our study, we used lower concentrations of BQ (0.5-5 μM), in the range of IC50 for DHODH inhibition, and our unbiased screens highlight enhanced apoptosis sensitivity as a reproducible result of DHODH inhibition in unbiased cell culture and in vivo-based screens. Furthermore, we have also demonstrated synergy between DHODH inhibition and BCL-X$_L$ targeting using a distinct DHODH inhibitor, BAY-2402234, that does not have an off-target effect on FSP1[60].

Combinatorial targeting of DHODHi with BCL-X$_L$ was effective through multiple cell culture conditions (DMEM, Plasmax), in vivo models of PDAC and even KRAS inhibitor-resistant organoid systems, suggesting a conserved combinatorial effect in different metabolic environments and a potential use of this strategy to sensitize KRASi-resistant PDAC cells. BCL-X$_L$ is a well-demonstrated cancer target; however, dose-limiting thrombocytopenia and cardiac toxicity limits the use of BCL-X$_L$ inhibitors clinically[61,62]. DT2216 is a BCL-X$_L$ PROTAC that targets BCL-X$_L$ to the Von Hippel-Lindau (VHL) E3 ligase for degradation[23]. Given the lower expression of VHL in platelets, DT2216 is less able to degrade BCL-X$_L$ thereby reducing thrombocytopenia[23]. No toxicity was reported when DT2216 was combined with gemcitabine in PDAC mouse models[28] or AZD8055 in small-cell lung cancer mouse models[63]. Similarly, we did not observe weight loss or thrombocytopenia in our NSG xenograft model (Supplementary Fig. 10d and Supplementary Data 11). However, the combination of BQ and DT2216 caused modest anemia, thrombocytopenia, and mild kidney toxicity in the C57BL/6 J model (Supplementary Data 10). One strategy to mitigate potential hematologic side effects of the BQ and DT2216 combination is administration of recombinant G-CSF (filgrastim, pegfilgrastim) and romiplostim, common supportive treatments administered with FOLFIRINOX chemotherapy[51]. Although G-CSF-induced neutrophils can suppress T cell activity[64], we have shown that the combination of BQ and DT2216 does not require adaptive

immunity. Notably, BQ-treated BCL-X$_L$ KO tumor-bearing mice exhibited significant tumor growth inhibition and minimal treatment-related toxicity (Fig. 6e, f), suggesting that a strategy combining tumor-specific BCL-X$_L$ targeting with systemic DHODHi offers a promising therapeutic approach with the potential for reduced toxicity. Indeed, targeting BCL-X$_L$ inhibitors to tumors via an antibody–drug conjugate (ADC) approach is being evaluated as a strategy to minimize non-tumor tissue effects[65]. ABBV-637, an ADC consisting of a monoclonal antibody directed against the epidermal growth factor receptor (EGFR) conjugated to an inhibitor of BCL-X$_L$, and ABBV-155, an ADC composed of a monoclonal antibody against the immunoregulatory protein B7-homologue 3 (B7-H3, CD276) conjugated to a BCL-X$_L$ inhibitor are now in Phase 1 clinical trials for lung cancer treatment[62] (NCT04721015 and NCT03595059). Overall, we have designed a multiomics approach that facilitates discovery of mechanisms of adaptation to targeted therapies and nominates candidates for combination targeting with improved anti-tumor efficacy. Using this strategy, we nominate DHODH inhibition and BCL-X$_L$ targeting as a potential therapeutic strategy in PDAC.

## Methods

### Institutional review board statement
Animal studies were performed in accordance with a Dana-Farber Cancer Institute Institutional Animal Care and Use Committee–approved protocol (10-055). Patient-derived pancreatic cancer organoids were obtained under Dana-Farber/Harvard Cancer Center IRB-approved protocols 11–104, 17–000, 03–189, or 14–408.

### Cell culture
PaTu-8988T and PaTu-8902 cells were obtained from DSMZ (ACC 162 and ACC 179), PANC-1, HPAC, Panc 02.03, and HEK-293T cells were from ATCC (CRL-1469, CRL-2119, CRL-2553, CRL-3216) and PK-1 and KP-4 were from RIKEN Cell Bank (RBC1972, RBC1005). The murine KPCY 6499 C4 and KPCY6694 C2 cell lines were from Kerafast (EUP015-FP and EUP006-FP, respectively). Cell lines were maintained in a centralized cell bank and authenticated by assessment of cell morphology as well as short tandem repeat fingerprinting. Cells were routinely tested for Mycoplasma contamination using PCR and were negative for Mycoplasma. After thawing, cell lines were cultured for no longer than 30 days. Cell lines were maintained at 37 °C with 5% CO2 and grown in DMEM or RPMI 1640 supplemented with 10% FBS and 1% penicillin/streptomycin, unless otherwise specified. Plasmax™ (CancerTools, 156371) was supplemented with 2.5% FBS and 1% penicillin/streptomycin before use. HPLM (Gibco, A4899101) was supplemented with 10% dialyzed FBS (Gibco, A3382001), RPMI 1640 1X Vitamins (MCE, R7256) and 1% penicillin/streptomycin before use. Mouse and human pancreatic cancer cell lines were derived from both male and females.

The organoids were derived from tissue specimens from patients with metastatic or localized PDAC who underwent resection or biopsy between November 2015 and July 2019 at DF/BWCC. Investigators obtained written, informed consent from patients at least 18 years old with pancreatic cancer on Dana-Farber/Harvard Cancer Center IRB-approved protocols 11–104, 17–000, 03–189, or 14–408 for tissue collection, molecular analysis, and organoid generation. Relevant population characteristics of the human research participants were previously defined[66]. All organoids used for this study were from female patients. Organoid cultures were established and maintained according to established techniques[66]. In brief, tumor cells from patients after dissociation were cultured in 3-dimensional (3D) Growth-factor Reduced Matrigel (Corning), added with human complete organoid medium containing Advanced DMEM/F12 (GIBCO), 10 mM HEPES (GIBCO), 1x GlutaMAX (GIBCO), 500 nM A83-01 (Tocris), 50 ng/mL mEGF (Peprotech), 100 ng/mL mNoggin (Peprotech), 100 ng/mL hFGF10 (Peprotech), 10 nM hGastrin I (Sigma), 1.25 mM N-acetylcysteine (Sigma), 10 mM Nicotinamide (Sigma), 1x

B27 supplement (GIBCO), RSPONDIN-1 conditioned media 10% final, WNT3A conditioned media 50% final, 100 U/mL penicillin/streptomycin (GIBCO), and 1x Primocin (Invivogen), and maintained at 37 °C in 5% CO2. For proliferation assays, organoids were dissociated with TrypLE Express (GIBCO) before re-seeding into fresh Matrigel and culture medium.

### Cell doubling
To assess the effects of different doses of Brequinar in PDAC cells, 300,000 cells were plated in 6 cm plates. On day 1, BQ was added at 0.5, 1 or 5 μM and cells were counted on days 3, 6, 9 and 12. Number of cells were quantified by flow cytometry (Beckman Coulter Cytoflex).

### Metabolomics
Steady state metabolomics experiments were performed as previously described[67]. PaTu-8988T cells were plated and medium was refreshed the next day with BQ/DMSO. Media was refreshed 2 h before metabolite collection at 24 h or 7 days treatment. Experiments were performed in 25 mM glucose/4 mM glutamine. Metabolites were extracted on dry ice with 2 mL of 80% methanol (LC-MS grade, cooled to −80 °C). Plates were incubated for 30 min at −80 °C and metabolites were centrifuged at 4200 rpm for 10 min (4 °C) twice to remove all insoluble materials. Samples were lyophilized using a SpeedVac and dried samples were kept at −80 °C. On the day of analysis, pellets were resuspended in 20 μl of LC/MS grade water and centrifuged at $20,000 \times g$ for 5 min at 4 °C. 5 μl of sample were injected onto the LC-MS/MS system comprised of a 5500 QTRAP hybrid dual quadrupole ion trap mass spectrometer (AB/SCIEX) and a Prominence HPLC (Shimadzu) with an autosampler outfitted with an Amide XBridge column (Waters; 3.5 μm particle size, 4.6 mm diameter (i.d.) × 100 mm length; Waters cat. No. 186004868). For the QTRAP mass spectrometer, the following settings were used: +4800 V positive/−4500 V negative; 475 °C; curtain N2 gas set to 25; high collision energy; ion source gas 1 and 2 set to 35; declustering potential +93 positive/−93 negative; entrance potential +10 positive/−10 negative; collision cell exit potential +10 positive/−10 negative[67,68]. HPLC gradient as has been described before[67]. Metabolites were identified by matching elution time with standards. SRM Q1/Q3 peak integration was performed using MultiQuant v2.0 (AB/SCIEX). Metabolite fractions were normalized to cell number in a parallel 6 cm plate.

### Quantitative proteomics
Cells were lysed using 8 M urea, 200 mmol/L EPPS, pH 8.5 with protease inhibitors. 100 μg of protein extracts were reduced using TCEP and alkylated with 10 mmol/L iodoacetamide followed by chloroform/methanol precipitation. Protein pellets were digested overnight with Lys-C and trypsin digested the next day. Peptides were labeled using 200 μg of TMT reagent (20 μg/μl). To equalize protein loading, a ratio check was performed by pooling 2 μg of each TMT-labeled sample. Pooled TMT-labeled peptide samples were fractionated by basic-pH reverse-phase HPLC. Samples were desalted using StageTips prior to analyses using LC-MS/MS/MS. All mass spectrometry data were acquired using an Orbitrap Lumos mass spectrometer in line with a Proxeon NanoLC-1200 UHPLC system[13]. All acquired data were processed using Comet[69] and a previously described informatics pipeline[70]. Spectral searches were done using fasta-formatted databases (Uniprot Human, 2020, or Uniprot Mouse, 2020). Protein quantitative values were normalized so that the sum of the signal for all proteins in each channel was equal to account for sample loading.

### Brequinar-anchored CRISPR/SpCas9 genetic screen
The all-in-one version of the Human CRISPR knockout Pooled Library (Addgene #73179) was applied for the in vitro genome-wide loss-function screen in PaTu-8988T cells[20]. The distribution degree of all guides in the amplified library were assessed via NGS to ensure the Gini

index is less than 0.1. 64 µg of pooled plasmid was produced from 100 million 293 T cells using Lipofectamine 3000 according to manufacturer's standard protocol. The molar ratio of pooled plasmid, psPAX2 (Addgene #12260) and pMD2.G (Addgene #12259) was 2.5:2:1. Culture media was refreshed 24 h after transfection, and virus-containing media were collected at 48 h and 72 h. Next, virus-containing media were pooled and filtered through a 0.45 µm filter. To determine viral efficiency, dilutions of virus were mixed with 1 million PaTu-8988T cells and cells were incubated in complete culture media containing 4 µg/ml polybrene and spun at 1850 rpm at 25 °C for 2 hours. Cells were incubated at 37 °C for 24 h followed by selection with 1 µg/ml puromycin. After a 4-day puromycin selection, cells were counted, and virus efficiency was calculated as puromycin-selected well/no-puromycin well. In the full-scale experiment, 108 mL virus were used to infect 180 million cells via the plate spin infection method and then incubated with 1 µg/ml puromycin for 5 days. The final infection ratio was 23%. Cells were amplified in complete media until 400 million cells were obtained. Cells were then divided into three arms (40 million cells as biological triplicates in each arm): DMSO, 0.5 µM BQ (low dose) and 5 µM BQ (high dose). Cells were split every 3 days and treated with BQ for 14 days in total. Cells were collected and genome DNA was extracted via Qiagen Blood & Cell Culture DNA Maxi Kit (#13362). Guide RNA sequences were amplified using Illumina sequencing specific primer sets with NEB Q5 High-Fidelity DNA polymerase (#M0491L). Amplified amplicon sequencing was performed using next-generation sequencing (NGS) at 500x depth of each guide RNA by Novogene. Acquired NGS reads were analyzed via the MAGeCK tool[71]. Candidate genes were selected based on MAGeCK's output, considering both effect sizes and statistical significance.

### In vitro and in vivo small library CRISPR–SpCas9

We generated a small library targeting 369 genes to test in a mini-pool in vivo CRISPR/Cas9 screen. We prioritized: 1) proteomics hits with FDA available drugs and hits from the CMap analysis, 2) most statistically significant CRISPR depleted/enriched hits, 3) genes that had been previously related to DHODH in published in vitro studies. Four guide RNA sequences were designed for each gene based on the Brunello library and 528 non-targeting control guide RNA sequences were included. After adding uniform adaptor sequences to each guide RNA, the oligo pool was synthesized via Twist Bioscience. 10 ng of synthesized oligo pool was amplified using Q5 polymerase then purified using the Qiagen PCR Purification Kit (#28104). 500 ng purified oligo were mixed with 5 µg lentiCRISPR v2 (Addgene #52961) backbone, which was linearized with Esp3I enzyme (NEB #R0734L). The ligation was performed by adding an equal volume of NEB HiFi DNA Assembly Kit (#E5520S) into the above mix and incubated at 50 °C for 60 min. Electrocompetent transformation was applied to the assembled products using ElectroMAX Stbl4 Competent Cells (Thermo #11635018) according to the standard manufacturer's protocol. Transformed cells were incubated in 1 L Terrific Broth at 225 RPM 30 C for 14 h. Zymo-PURE II Plasmid Maxiprep Kit (ZYMO #D4203) was used to extract plasmid DNA. The distribution degree of all guides in the amplified library was assessed via NGS to ensure the Gini index was less than 0.1. Lentivirus production and infection was performed as described in the in vitro genome-wide screen. Virus infection ratio was controlled at 10%. For cell culture-based screens, ×1000 depth of each guide was used. BQ was applied using the same approach as described in in vitro genome-wide screen. Genomic DNA was extracted at the end of treatment. For the in vivo screen, 5 million PaTu-8988T cells in 50 µl PBS and 50 µl matrigel were injected in bilateral flanks of NOD.Cg-*Prkdc*[scid] *Il2rg*[tm1Sug]/JicTac mice (NOG mice, 7 weeks, female, Taconic, n = 10 tumors per arm). This strategy ensured 4000x depth for each guide. Treatment was initiated when tumors reached 80-100 mm³. BQ was prepared by dissolving in 70% PBS 1x /30% PEG-400 and adjusting pH to 7.00 with NaOH[3]. Mice were injected once every other day for

14 days at 50 mg/kg with BQ or vehicle. At endpoint (4 h after last dose), tumor tissues were collected, and genomic DNA was extracted using DNeasy Blood & Tissue Kit (Qiagen #69504). NGS library preparation, sequencing, and analysis were performed as described for the genome-wide screen. All sgRNA sequences for the in vivo library come from the commercial Brunello library and are now included in new Supplementary Data 6 in addition to raw read counts.

### Real time cell proliferation and apoptosis

Real-time detection of cell growth and Annexin V was performed following Incucyte® S3 protocols per the manufacturer's guidelines. Cells were plated at 1000–2000 cells/well in 96-well plates. The next day, drugs were added at the indicated concentrations. For apoptosis monitoring, Incucyte® Annexin V Red Dye was added at the same time as indicated drugs. Plates were kept in the Incucyte® S3 system inside a cell culture incubator for 72 hours. Cell growth and accumulation of Annexin V fluorescent signal were captured with the Incucyte® Live-Cell Analysis System every 3-4 hours. The Annexin fluorescent signal was analyzed by Incucyte® Live-Cell Analysis System and Total Red Object Integrated Intensity (RCU x µm²/Image, RCU means Red Calibrated Unit) was used to plot the curve in GraphPad. For cell growth analysis, relative growth, normalized to Day 0, was calculated and expressed as a fold change ratio using the Incucyte® Live-Cell Analysis System. The normalized data were then used to generate growth curves using GraphPad Prism.

### Synergy testing

On day 0, 300 cells in 20 µL of medium were seeded in 384-well plates using a microplate dispenser (Type 836, Thermo Scientific Combi). HPAC was seeded at 500 cells/well. On day 1, drugs were administered with a Tecan D300e drug dispenser (Tecan, Männedorf, Switzerland). Plates were kept in an incubator for 5 days. 20 µL of CellTiter-Glo were added by microplate dispenser and plates were incubated on a shaker at room temperature for 1 h. Luminescence was measured with a Beckman Coulter Cytoflex. IC50 values were calculated using nonlinear regression analysis in GraphPad Prism. Synergy score between drugs was calculated using the viability (%) normalized to the DMSO control with analysis performed using the HSA model implemented in SynergyFinder (https://www.synergyfinder.org/)[72]. The mean HSA value of drug combination was used to plot the synergy score heatmap by GraphPad.

### Apoptosis analysis by flow cytometry

Cells were plated at 50,000 cells/well (24-well plate). Cells were treated the next day with the indicated compounds for 72 h. Adherent and non-adherent cells in the medium were collected and stained with Annexin-FITC and propidium iodide (PE) for 15 min (BD biosciences 556547) as per the manufacturer protocol. Cells were placed on ice and analyzed using a Beckman Coulter Cytoflex. Quantification of apoptotic cells (%) after treatment included cell populations which were in early apoptosis (FITC Annexin V positive and PI negative) as well as late apoptosis (FITC Annexin V and PI positive).

### BH3 profiling

BH3 profiling was conducted via flow cytometry following established protocols[73] Briefly, cultured cells were trypsinized, centrifuged at $500 \times g$ for 5 minutes, resuspended in mannitol experimental buffer (MEB; 10 mM HEPES (pH 7.5), 150 mM mannitol, 50 mM KCl, 0.02 mM EGTA, 0.02 mM EDTA, 0.1% BSA, and 5 mM succinate), and added to wells of prepared 96-well plates containing the indicated peptide conditions and 0.001% digitonin. Cells were then incubated for 60 min at 28 °C, followed by fixation for 15 minutes in 8% PFA. Fixation was neutralized using N2 buffer (containing 1.7 M tris base and 1.25 M glycine, pH 9.1), and the cells were subsequently stained overnight with DAPI and an Alexa Fluor 647-conjugated anti-cytochrome c antibody

(BioLegend, 612310) at 4 °C. Finally, the stained cells were analyzed using an Attune NxT flow cytometer. The percentage of cytochrome c negative cells was determined for each peptide treatment condition.

## Quantitative real-time PCR (qRT-PCR)

Cells were plated in a 10 cm dish at a density to ensure 60–70% confluency at the time of treatment. After overnight attachment, cells were treated with DMSO (control) or BQ (5 μM) for 24 hours. Cells were washed with cold PBS and processed immediately for RNA extraction (FastPure Cell/Tissue Total RNA Isolation Kit V2, Vazyme, RC112-01). cDNA was synthesized using the High-Capacity cDNA Reverse Transcription Kit (Applied Biosystems, 4368814). qPCR was performed using Taq Pro Universal SYBR qPCR Master Mix (Vazyme, Q712-02) on an Applied Biosystems. Technical triplicates were run for each sample. qPCR results were analyzed using the ΔΔCt method. Gene expression levels were normalized to housekeeping genes (GAPDH), and fold change was calculated relative to untreated controls. Data were plotted as mean ± standard deviation of biological replicates. Primer sequences used for qPCR are provided in Supplementary Data 12.

## Clonogenic assay

Cells were seeded at a density of 400 cells per well in six-well plates. One day after plating, the indicated drug concentrations were added. After 5 days, the drug treatments were refreshed. On day 10, cells were fixed and stained using a solution of 80% methanol/0.1% crystal violet. Colonies were subsequently counted across all treatment conditions.

## TMRM measurements

Cells were treated with 5 μM Brequinar for 24 hours or 7 days. On the day of analysis, cells were trypsinized and stained with 25 nM Image-iT™ TMRM Reagent (Thermo Scientific, Cat. #I34361) and 20 μM MitoTracker™ Green (Thermo Scientific, Cat. #M46750) in DMEM supplemented with 1% FBS at 37 °C for 30 minutes. FCCP at 1 μM was added 10 minutes prior to analysis as positive control. TMRM fluorescence was calculated by subtracting basal mean fluorescence intensity (FCCP-treated) and normalizing to mitochondrial content by dividing to Mean Fluorescence Intensity of MitroTracker™ Green. Dead cells were excluded using DAPI staining. Flow cytometry was performed on a NovoCyte Quanteon analyzer (Agilent Technologies, Santa Clara, CA, USA) at the FACS Core Facility of Aarhus University, Denmark. MitoTracker™ Green fluorescence was detected on the B530-A channel, while TMRM fluorescence was measured on the Y586-A channel. Data were analyzed using FlowJo™ v10.10 Software (BD Life Sciences).

## Seahorse assay

OCR measurements were performed using a Seahorse Metabolic Flux Analyzer XFe96 instrument (Seahorse, Agilent, USA). 20,000 untreated or 7 day-treated cells were seeded in sextuplicate in DMEM supplemented with 10% FBS 48 h prior to the assay. For 24-hour treatment, BQ was added 24 h after seeding. The day of the measurement, media was replaced with Seahorse assay media prepared by supplementing XF Base media, (Agilent, #103193-100; adjusted to pH = 7.4) with 25 mmol/L glucose (Sigma, G8760), 300 mg/L glutamine (Thermo Fisher, 25030081) and 1 mM sodium pyruvate (Gibco, 11360-039). The plate was equilibrated for 1 hour in a non-CO₂, 37 °C incubator. Mitochondrial stress assay was performed with sequential injections of 1 μmol/L oligomycin, 0.5 μmol/L FCCP, and 0.5 μmol/L rotenone/0.5 μmol/L antimycin A. Data were normalized to cell number determined by Hoescht staining using a Cytation1 Cell Imaging Software (Agilent). Normalization unit was set to 20,000 cells.

## Western blotting

Cells were collected in ice cold PBS and lysed in RIPA buffer containing 1× Halt Protease and Phosphatase Inhibitor (Thermo Scientific, 78446).

Lysates were centrifuged and supernatants collected. Protein (100 μg) was resolved on 4% to 12% SDS-PAGE gels and transferred to PVDF membranes. Membranes were blocked in 5% milk and incubated with primary antibodies followed by peroxidase-conjugated secondary antibodies. Membranes were developed using the ECL Detection System (Thermo, 32209). The following antibodies were used: BCL-2 (D17C4) (CST, Cat# 3498, RRID:AB_1903907, 1:500), BCL-X$_L$ (54H6) (CST, #2764, RRID:AB_222800, 1:1000), MCL-1 (D35A5) (CST, #5453S, RRID:AB_10694494, 1:1000), PARP (CST, 9542S, RRID:AB_2160739, 1:1000), PUMA (E2P7G) (CST, #98672S, RRID:AB_3096180, 1:1000), BIM (C34C5) (CTS, #2933, RRID:AB_1030947, 1:1000), cleaved caspase 3 (Asp175) (CST, #9661, RRID:AB_2341188, 1:1000), Phospho-NF-κB p65 (Ser536) (93H1) (CST, #3033S, RRID:AB_331284, 1:1000), NF-κB p65 (D14E12) (CST, #8242S, RRID:AB_10859369, 1:1000), Phospho-NF-κB p105 (Ser932) (18E6) (CST, #4806S, RRID:AB_2282911, 1:1000), NF-κB1 p105 Antibody (CST, #4717S, RRID:AB_2282895, 1:1000), ACTB (Sigma, A5441, RRID:AB_476744, 1:5,000), Anti-rabbit IgG (H1L) HRP conjugate (Thermo, 31460, RRID:AB_228341, 1:3,000), and Anti-mouse IgG (H1L) HRP conjugate (Promega, W4021, RRID:AB_43083, 1:7,000). Quantification was performed using ImageJ.

## Generation of BCL-X$_L$ knockout in PaTu-8902 cells

Two sgRNAs targeting BCL-X$_L$ and one sgRNA targeting GFP as a negative control (Supplementary Data 12) were cloned into the lenti-CRISPR v2 plasmid (Addgene #52961) using standard molecular cloning techniques. Lentiviral vectors were produced by transfecting HEK293T cells with 1 mg/mL PEI. Viral supernatants were collected at 48 and 72 hours post-transfection and filtered through a 0.45 μm filter. PaTu-8902 cells were seeded at 40–50% confluency and transduced with lentiviral supernatants in the presence of 4 μg/mL polybrene to enhance transduction efficiency. After 24 hours, the medium was replaced with fresh complete culture medium. Puromycin selection (1 μg/mL) was initiated 48 hours after transduction and continued for 3 days until all untransduced control cells were eliminated. Puromycin-resistant cells were seeded into 96-well plates by serial dilution to isolate single-cell clones. Individual clones were expanded and screened by PCR and Western blotting to confirm successful BCL-X$_L$ knockout.

## Subcutaneous mouse xenograft and allograft studies

PaTu-8902 cells ($1 \times 10^6$ in PBS:Matrigel (1:1)) were implanted subcutaneously in the flank of NOD.Cg-Prkdc$^{scid}$ Il2rg$^{tm1Sug}$/JicTac mice (NOG mice, 7 weeks, female, Taconic). Tumors were allowed to reach 150 mm³ before randomization to treatment groups. HPAC ($3 \times 10^6$ in PBS:Matrigel (1:1)) cells were implanted subcutaneously in the flank of the NOD.Cg-Prkdc$^{scid}$ Il2rg$^{tm1Wjl}$/SzJ mice (NSG mice, 8 weeks, female, Jackson Lab). Tumors were allowed to reach 150-200 mm³ before randomization to treatment groups. KPCY 6694 C2 cells ($0.5 \times 10^6$ in PBS:Matrigel (1:1)) were implanted subcutaneously in the flank of C57BL/6 J mice (8 weeks, female, Jackson Lab). Tumors were allowed to reach 75 mm³ before randomization to treatment groups. PaTu-8902 non-targeting cells, ($1 \times 10^6$ in PBS:Matrigel (1:1)) and PaTu-8902 BCL-X$_L$ KO cells ($1.3 \times 10^6$ in PBS:Matrigel (1:1)) were implanted subcutaneously in the flank of NOD.Cg-Prkdc$^{scid}$ Il2rg$^{tm1Sug}$/JicTac mice (NOG mice, 9 weeks, female, Taconic). Tumors were allowed to reach 100 mm³–150 mm³ before randomization to treatment groups. DT2216 was formulated in 50% phosal 50 PG/45% miglyol 810 N/5% polysorbate 80. BQ was formulated in 70% PBS 1x /30% PEG-400 and adjusting pH to 7.00 with NaOH. DT2216 (15 mg/kg, twice per week) and BQ (10 mg/kg, three times per week) were administered intraperitoneally for 21 days. Mice in vehicle-control arms received 100 μl 30% PEG-400/70% PBS and 100 μl 50% phosal 50 PG/45% miglyol 810 N/5% polysorbate 80 following the same schedule as DT2216 and BQ administration. Tumor volume was measured twice a week with calipers as follows: volume = (length × width²)/2. Mice were euthanized when the

tumor size approached 2 cm in diameter (maximal tumor size allowed by the Ethics Committee). Tumors were then removed and cut into several pieces which were frozen or fixed in 10% formalin for analysis. Drug toxicity was assessed by weighing mice twice a week and by complete blood counts at endpoint. Blood was drawn by submandibular bleeding and analyzed using GENESIS™ hematology analyzer from Oxford Science. For all in vivo experiments, mice were housed in pathogen-free animal facilities at Dana-Farber Cancer Institute on a standard 12 h dark/12 h light cycle under controlled environmental conditions (temperature: 20 + / − 2 °C; humidity: 35 + / − 10%). Prolab Isopro RMH 3000 diet (Labdiet 5P75 and 5P76) and water were provided *ad libitum*. All the mice for this study were female and sex was not a specified variable for analysis.

### Orthotopic models of PDAC
50,000 KPCY 6694 C2 cells in 30 µl of PBS:Matrigel 1:1 were implanted in pancreata of C57BL/6 J mice[74] (10 weeks, female, Jackson Lab). DT2216 (15 mg/kg, twice per week) and BQ (10 mg/kg, three times per week) were administered intraperitoneally starting on day 4 after implantation. Mice were weighed twice a week to assess drug-related toxicity. Mice were euthanized 20 days post-surgery and tumors were harvested at endpoint and weighed.

### Immunohistochemical staining
Tissues were fixed in formalin and paraffin embedded. After deparaffinization, primary antibody was incubated followed by secondary antibody and then developed by DAB. The following antibody was used: cleaved caspase 3 (Asp175) (CST, #9661, RRID:AB_2341188, 1:400). Positive cell staining was quantified by ImageJ. Four 20× fields per mouse were analyzed.

### Serum ALT, BUN and BNP analysis
Blood collected from mice was immediately transferred to microtainer tubes without anticoagulants and allowed to clot at room temperature for 30−60 minutes. The samples were centrifuged at $2000 \times g$ for 10 minutes at 4 °C to separate the serum. The resulting serum was aliquoted and stored at −80 °C until further analysis. Serum levels of alanine aminotransferase (ALT), blood urea nitrogen (BUN), and brain natriuretic peptide (BNP) were quantified using commercially available assay kits. ALT activity was measured using the ALT Activity Assay Kit (Abcam, #ab282882), BUN levels were determined using the BUN Colorimetric Detection Kit (RayBiotech, #MA-BUN-1), and BNP concentrations were assessed using the BNP Mouse ELISA Kit (Thermo Fisher Scientific, #EEL089). Assays were performed according to the manufacturers' protocols, incorporating standard curves and appropriate serum dilutions: 10x for BUN and BNP measurements, and 40×−80× for normal ALT samples. All measurements were conducted in technical duplicates to ensure reliability, and serum ALT, BUN, and BNP levels were calculated following the kit instructions.

### Bioinformatic analysis
Quantitative proteomics dataset analyses were performed as described[12]. Datasets were subject to analysis using LIMMA package (3.40.2)[75] in R platform. Between each pairwise comparison, LIMMA applied a standard linear model fitting and an empirical Bayes procedure to correct the distribution. After the correction, a moderated t-statistic was calculated for each gene using a simple Bayesian model, which served as the fundamental statistic. Type I error is corrected by calculating false discovery rate (FDR) using the Benjamini-Hochberg method. A volcano plot of $\log_2$ fold change and FDR served as a visualization method for each comparison. Broad GSEA software (3.0)[76] was used to perform gene set enrichment analysis (GSEA) against the collection containing all canonical pathways (c2.all.v2023.2.Hs.symbols.gmt) in MSigDB. Pathway clouds were

visualized in Cytoscape (3.7.2)[77] (with EnrichmentMap (3.2.1)[78] plugin, and each cluster was manually curated. A target list and a background list were created for each comparison and then used for Gene Ontology (GO) enrichment analyses[79]. Principal component analysis (PCA) was done with the prcomp function in R platform, and the two major components (PC1 and PC2) were used for visualization. The L1000 gene expression database in Connectivity Map (CMap) 2.0[19] was used as the reference and top 150 up- and downregulated proteins upon BQ treatments were queried against L1000 to calculate connectivity scores. To identify BCL-$X_L$, MCL1 and BCL2 dependencies and also BCL-$X_L$ expression across 45 PDAC cell lines, we analyzed data from pooled, genome-scale CRISPR–SpCas9 loss-of-function data within the Broad Institute's DepMap Public 23Q4+Score (Chronos) and RNAseq data within expression Public 23Q4[80].

### Chemicals
The following chemicals were used: Brequinar (MCE, HY-108325), BAY-2402234 (MCE, HY-112645), A-1331852 (MCE, HY-19741), DT2216 (Chemietek), uridine (Sigma, U3003), Z-VAD-FMK (Selleckchem, No.S7023), Pyrazofurin (MCE, HY-122502), AG_2037 (ADOOQ, A17217), IACS-010759 (MCE, HY-112037), TMRM (Thermo Scientific, #I34361) MitoTracker™ Green (Thermo Scientific, #M46750), FITC Annexin V Apoptosis Detection Kit I (BD Biosciences, 556547), Incucyte® Annexin V Dye (Sartorius, 4641), Normal goat serum (Vector Labs, S-1000), Vectastain ABC-HRP kit (Vector Labs, PK-4001), Goat Anti-Rabbit IgG(H + L), Biotinylated (Vector Labs, BP-9100), and ImmPACT DAB Substrate kit (Vector Labs, SK-4105), CellTiter-Glo® 2.0 Cell Viability Assay (Promega, G924).

### Statistical analysis
For comparisons between two groups, a Student t-test (unpaired, two-tailed) was performed as noted. Groups were considered different when $P < 0.05$. For multiple comparisons, ordinary one-way ANOVA tests (Tukey statistical hypothesis test was used for correlation) were performed using Prism 10 (GraphPad Software). R-based analyses as described above, and figure generation were performed in R v4.0.3.

### Illustrations and diagrams
Drawings for experimental set-up were created in Adobe Illustrator (v24.1.2) and BioRender. Biorender images: Fig. 1a (Santana codina, N. (2025) https://BioRender.com/a62aees), Fig. 1g (Santana codina, N. (2025) https://BioRender.com/4lblqps), Fig. 2b (Santana codina, N. (2025) https://BioRender.com/t12d46e), Fig. 6 (Mancias, J. (2025) https://BioRender.com/i4zww2l)

### Reporting summary
Further information on research design is available in the Nature Portfolio Reporting Summary linked to this article.

## Data availability
Source data for each graph is provided in a separate Source Data file. The mass spectrometry proteomics data used to generate Fig. 1 and Supplementary Fig. 2 have been deposited in the ProteomeXchange Consortium with the dataset identifier 059471: https://www.ebi.ac.uk/pride/archive/projects/PXD059471. The metabolomics data are available as Supplementary Data 1 and have been deposited to the MassIVE repository with the dataset identifier MSV000097577: ftp://massive-ftp.ucsd.edu/v09/MSV000097577/. The CRISPR screen data are available as Supplementary Data 4-6 and at the European Nucleotide Archive (ENA) under the following accession number PRJEB90251: https://www.ebi.ac.uk/ena/browser/view/PRJEB90251. The other data supporting the findings of this study are available within the article and its Supplementary Information files. Source data are provided with this paper.

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

## Acknowledgements

This research was supported by a Burroughs Wellcome Fund Career Award for Medical Scientists and the Sidney Kimmel Foundation Kimmel Scholar Program (to JDM), a Claudia Adams Barr Program for Innovative Cancer Research grant and an AUFF Recruiting Grant (to NSC), and the Hale Family Centre for Pancreatic Cancer Research (to JDM, SKD, AJA). We thank the Harvard Medical School Rodent Histopathology core for tissue processing. Flow cytometry was performed at the FACS Core Facility, Aarhus University, Denmark. We acknowledge Dr. Steven Gygi for the use of mass spectrometry data analysis software.

## Author contributions

Conceptualization: H.Z., N.S.C., J.D.M. Data curation: H.Z., N.S.C., Q.Y., M.K., J.D.M. Investigation: H.Z., N.S.C., Q.Y., C.P., C.C., X.Q., N.S., M.C., A.P., M.K., J.W., M.D., S.K.D., J.D.M. Methodology: H.Z., N.S.C., Q.Y., C.P, X.Q., P.B., M.K., J.W., S.K.D., J.D.M. Formal analysis: H.Z., N.S.C., Q.Y., X.Q., M.C., M.K., J.D.M. Visualization: H.Z., N.S.C., Q.Y., X.Q., J.D.M. Funding acquisition: N.S.C., A.J.A., S.K.D., J.D.M. Writing—original draft: H.Z., N.S.C., J.D.M. Writing—review & editing: All authors. Project administration: N.S.C., J.D.M. Supervision: A.J.A., S.K.D., K.A.S., N.S.C., J.D.M.

## Competing interests

J.D.M. reports research support to his institution from Novartis and Casma Therapeutics and has consulted for Third Rock Ventures and Skyhawk Therapeutics, all unrelated to the submitted work. A.J.A. has

consulted for Anji Pharmaceuticals, Affini-T Therapeutics, Arrakis Therapeutics, AstraZeneca, Boehringer Ingelheim, Kestrel Therapeutics, Merck & Co., Inc., Mirati Therapeutics Inc., Nimbus Therapeutics, Oncorus, Inc., Plexium, Quanta Therapeutics, Revolution Medicines, Reactive Biosciences, Riva Therapeutics, Servier Pharmaceuticals, Syros Pharmaceuticals, T-knife Therapeutics, Third Rock Ventures, and Ventus Therapeutics; holds equity in Riva Therapeutics and Kestrel Therapeutics; and has research funding from Amgen, Boehringer Ingelheim, Bristol Myers Squibb, Deerfield, Inc., Eli Lilly, Mirati Therapeutics Inc., Novartis, Novo Ventures, Revolution Medicines, and Syros Pharmaceuticals, all unrelated to the submitted work. S.K.D. has received research funding from Novartis, Bristol Myers Squibb, Casma Therapeutics and Takeda, has equity in Axxis Bio, and is a co-founder and SAB member for Kojin Therapeutics, all unrelated to the submitted work. All authors declare no competing interests.
