## [Transparent Peer Review file · Nature Communications]

De novo pyrimidine biosynthesis inhibition synergizes with BCL-X_L targeting in pancreatic cancer

Corresponding Author: Dr Joseph Mancias

Version 0:

Reviewer comments:

Reviewer #1

(Remarks to the Author)

Zhang, Yu, and colleagues report that inhibiting DHODH reprograms the apoptotic regulatory proteome, thereby increasing the sensitivity to inhibitors of the anti-apoptotic BCL-XL protein. They found that combining DHODH and BCL-XL inhibitors synergistically induces apoptosis in PDAC cell lines and patient-derived PDAC organoids. In vivo, DHODH inhibition using Brequinar, along with BCL-XL degradation via DT2216 (PROTAC), significantly limits the growth of pancreatic tumor cells. This study elucidates the adaptive mechanisms of DHODH inhibition and proposes a novel combination therapy strategy for PDAC.

This interesting and well-executed study identifies anti-apoptotic mechanisms helping PDAC tumors resist DHODH inhibition. These findings have significant therapeutic relevance as they identify potential new strategies to combat PDAC. However, some concerns need to be addressed before potential acceptance for publication in Nature Communications.

General comments:

1) The authors did not provide the molecular mechanisms underlying why DHODH inhibition renders PDAC cells sensitive to BCL-XL degradation. What makes the inhibition of DHODH unique in making the cancer cells more sensitive to apoptosis induction? The reviewer understands that these mechanistic questions might require considerable effort to address and might be out of the scope of this translational study. However, the following suggestions could provide more mechanistic insights to understand the underlying mechanisms of this combination:

1a- It would be interesting to assess whether the pro-apoptotic effects upon DHODH inhibition and BCL-XL degradation could be rescued or partially rescued by adding nucleosides such as uridine/cytidine.

1b- It would be valuable to assess whether purine synthesis inhibitors could mimic the effects of DHODH inhibitors on BCL-XL expression or the pro-apoptotic phenotype.

1c- Using inhibitors of pyrimidine synthesis beyond DHODH inhibitors could also validate whether this effect is specific to DHODH. For example, the authors could use pyrazofurin, a UMPS inhibitor (PMID: 29723133), to assess BCL-XL expression and the pro-apoptotic phenotype.

2) To ensure that the effects of DHODH inhibitors are not related to ETC inhibition, it is important to compare the pro-apoptotic effects of inhibitors targeting mitochondrial complexes, such as complex I inhibitors, in combination with the BCL-XL degrader.

Reviewer #2

(Remarks to the Author)

In manuscript "De novo pyrimidine biosynthesis inhibition synergizes with BCL-XL targeting in pancreatic cancer", the authors identified BCL2L1 gene (Bcl-XL) as a DHODH inhibition resistance gene using integrated multi-omics approach and CRISPR/Cas9 loss-of-function screen with DHODH inhibitor brequinar (BQ). Authors found that combined inhibition of DHODH and Bcl-XL, using BQ and Bcl-XL inhibitor DT2216, synergistically promotes apoptosis in various pancreatic cancer cells and suppresses PDAC growth in patient-derived organoids and in vivo models of PDAC. Interestingly, BQ promotes synergistic responses exclusively with BCL-XL inhibitor but not inhibitors of BCL2 or MCL1. These findings are

interesting and significant. The manuscript is, for most part, clearly written and experiments are presented in a logical way. However, the exact molecular mechanism by which Bcl-XL regulates BQ resistance remains unclear. Before publication of this manuscript in Nature communications, the following issues should be addressed:

1. CRISPR screen shows that BCL2L1 (Bcl-XL) and Mcl-1 are the top candidates contributing to BQ resistance in pancreatic cells (i.e. PaTu-8988T) (Fig. 1h), but in vivo screen using the same cell line only shows BCL2L1 (Bcl-XL) not Mcl-1 (Fig. 2f). Why data from in vitro and in vivo CRISPR screen are inconsistent, since xenografts were derived from same cell line PaTu-8988T as in vitro.
2. Why BQ only synergizes with Bcl-XL inhibitor DT2216 but not Mcl-1 inhibitor since Mcl-1 is also the top candidate in in vitro CRISPR screen.
3. What exact molecular mechanism(s) by Bcl-XL regulates BQ resistance in pancreatic cancer cells.
4. Effects of BQ in combination with Bcl-XL inhibitor DT2216, Bcl-2 inhibitor or Mcl-1 inhibitor on pancreatic cell growth should be measured by Colony formation assay.
5. The DHODH inhibitor BQ has been reported to have immunosuppressive and serious side effects. Therefore, combination of BQ and DT2216 in animal models, authors should observe immune profile and in vivo toxicity in more details, including blood cells (WBC, RBC, PLT), liver function (ALT, AST), kidney function (BUN), heart function, etc.

Reviewer #3

(Remarks to the Author)

This study by Zhang et al. is both interesting and clinically relevant. The authors conducted high-quality in vitro and in vivo screens, identifying BCL2L1 as a synergistic target with DHODH inhibition. Combination of DHODH and BCL2L1 inhibitors demonstrated a synergistic effect against cancer in both in vitro and in vivo settings. However, several weaknesses in the manuscript need to be addressed before it can be considered for publication.

Major:

1. The manuscript partially overlaps with the published paper: C. Mao et al. Nature <https://doi.org/10.1038/s41586-021-03539-7> (2021). Please address the similarities and differences between the two works, and discuss how your findings either complement or diverge from the conclusions of the Mao et al. study.
2. In Figure 1d and Figure 1e, is BCL2L1 upregulated? If not, please discuss its relevance to your findings. If it is upregulated, please ensure it is clearly labeled in the figures.
3. What is the library coverage of the library-transduced PDAC cells before and after the screen, as presented in Figures 1 and 2? Providing this information would help clarify the robustness of the screening results.
4. In Figure 2a, under criterion 3), why did the authors exclude genes that have been published in both in vitro and in vivo studies?
5. The genes involved in nucleotide salvage (UCK2, SLC29A1, CMPK1) and apoptosis (BCL2L1) are overlapped in both screens. Why did the authors choose not to focus on the nucleotide salvage genes for the subsequent experiments?
6. In Figure 6, the in vivo efficacy of combination targeting of DHODH and BCL-XL appears to be limited.

Minor:

1. Some labels in Figures 1d and 1e are unclear (e.g., SLC29A1, SLC7A11, and UCK2 in Figure 1e). Please clarify whether these genes are upregulated or downregulated in the figures.
2. The authors could consider testing different tumor models, such as AML, to enhance the clinical potential of the combination therapy of BD and DT2216.

Reviewer #4

(Remarks to the Author)

Reviewer #5

(Remarks to the Author)

In the manuscript by Zhang et al, the authors aim to identify mechanisms and potential therapeutic targets that contribute to the resistance to the inhibition of DHODH, a key enzyme for nucleotide synthesis and mitochondrial metabolism. The authors utilized unbiased and comprehensive approaches, including Integrated metabolomic, proteomic and CRISPR-mediated genetic screening in human PDAC cell lines, and revealed pathways critical of nucleotide salvage pathway and BCL-XL-mediated anti-apoptosis pathway for the adaptation to DHODH inhibition. The authors further demonstrated that pharmacological inhibition of DHODH and BCL-XL exhibits synergistic effect in inhibition PDAC growth both in vitro and in vivo. Overall the experiments are well-designed and well-executed. It not only offers molecular insight for the adaptive mechanisms to the inhibition of DHODH in PDAC, but also holds strong translational potential by providing preclinical

evidence for combination therapeutic strategy. The overall manuscript is well-written and easy to follow. Nevertheless, a few concerns need to be addressed.

1. Additional characterization of mitochondrial function following short-term and long-term inhibition of DHODH, such as OCR and membrane potential, will further enhance the molecular insight of the study.
2. Currently, the molecular mechanisms underlying the synergy between DHODH inhibition and BCL-XL depletion is not thoroughly elucidated. How does DHODH inhibition promote the sensitivity to BCL-XL? Is DNA damage or ROS induction involved? Does DHODH inhibition affect cytochrome c release?
3. The PDAC cells exhibit unique dependency on BCL-XL, but not BCL2 or MCL. What's the impact of BCL-XL, BCL2 or MCL depletion on cytochrome c release and caspase activation?
4. Additional genetic studies showing the synergy between BCL-XL depletion and DHODH inhibition are preferred.

Minor comment

On page 7, references are missing for the following statement, 'In agreement with recent studies, we also identified an induction in HLA-I and proteins involved in antigen presentation'.

Version 1:

Reviewer comments:

Reviewer #1

(Remarks to the Author)

The authors have addressed my concerns. I congratulate the authors on an excellent paper!

Reviewer #2

(Remarks to the Author)

Authors have addressed my comments properly by additional experiments and discussions.

Reviewer #5

(Remarks to the Author)

The authors conducted extensive experiments and have successfully addressed all my comments. I have no additional comment.

Reviewer #6

(Remarks to the Author)

This is an interesting paper with potential important implications. As far as I can assess the authors have adequately answered most of reviewer 3 comments.

I think this paper should be accepted with a few minor revisions:

1) In the CRISPR screen BCL2L1 and MCL1 score as targets but as reviewer 3 mentions, BCL2L1 is down regulated following BQ treatment. This suggests that BQ works by activating apoptosis (but it is not a complete response) and that further inhibition of either MCL or BCL2L1 is enough to get apoptosis. This is very interesting since these two genes are typically regulated differently and mutually exclusive. What about MCL1 this should be highlighted also in the proteomic data. Since BCL2L1 and MCL1 are very well-known co-dependencies, the authors should comment on the possibility of a trio combination or that MCL1 is restricting BCL2L1 suppression.

2) It is very critical that the raw read counts from the CRISPR screen are added as a supplementary table for re-analysis of this data.

Point-by-point Response to Reviews:

Below, we have copied each reviewer's comments (blue text) and present a point-by-point response (black text). For ease of review, we present data as "Response Figures" that integrate figure panels for the response to reviewers but also note where these were included in the revised manuscript. When appropriate, we have also included relevant text from the manuscript for ease of reference by the reviewers.

Reviewer #1 (Remarks to the Author):

Zhang, Yu, and colleagues report that inhibiting DHODH reprograms the apoptotic regulatory proteome, thereby increasing the sensitivity to inhibitors of the anti-apoptotic BCL-XL protein. They found that combining DHODH and BCL-XL inhibitors synergistically induces apoptosis in PDAC cell lines and patient-derived PDAC organoids. In vivo, DHODH inhibition using Brequinar, along with BCL-XL degradation via DT2216 (PROTAC), significantly limits the growth of pancreatic tumor cells. This study elucidates the adaptive mechanisms of DHODH inhibition and proposes a novel combination therapy strategy for PDAC.

This interesting and well-executed study identifies anti-apoptotic mechanisms helping PDAC tumors resist DHODH inhibition. These findings have significant therapeutic relevance as they identify potential new strategies to combat PDAC. However, some concerns need to be addressed before potential acceptance for publication in Nature Communications.

We thank the reviewer for their constructive comments. As below, we have now addressed the reviewer's points.

General comments:

1) The authors did not provide the molecular mechanisms underlying why DHODH inhibition renders PDAC cells sensitive to BCL-XL degradation. What makes the inhibition of DHODH unique in making the cancer cells more sensitive to apoptosis induction? The reviewer understands that these mechanistic questions might require considerable effort to address and might be out of the scope of this translational study. However, the following suggestions could provide more mechanistic insights to understand the underlying mechanisms of this combination:

We agree with the reviewer that understanding the molecular mechanism underlying why DHODH inhibition increases sensitivity to BCL-XL degradation is an important point. For the revised manuscript, we have performed experiments to understand the mechanistic basis of this combinatorial effect. Taking into consideration Reviewer #1's comments and those of the other reviewers, we investigated two hypotheses: 1) that DHODH inhibition increases sensitivity to BCL-XL targeting by altering the activity of a transcriptional regulator of apoptosis genes, and 2) that DHODH inhibition alters mitochondrial function thereby enhancing apoptosis sensitivity.

In our original manuscript, we showed that DHODH inhibition (DHODHi) downregulated protein-level expression of BCL-2-family proteins as determined by quantitative proteomics and immunoblotting (**Fig. 4c, Supplementary Fig. 8c-d, Supplementary Table 2**). To understand if the changes at the proteome level correlated with transcriptional responses, in new revision experiments, we quantified expression of apoptosis-related genes by qRT-PCR (**Response Fig. 1a; included in the manuscript as Fig. 4f**). Consistent with protein-level expression changes, PDAC cells treated with DHODHi downregulated expression of anti-apoptotic genes (*BCL2L1* (*BCL-X_L*), *MCL1* and *BCL2*) and upregulated expression of pro-apoptotic genes (*BCL2L11* (*BIM*)).

Response Figure 1 DHODHi modulates expression of apoptotic genes by impairing activation of the NF-κB pathway. (a) Apoptosis regulatory gene expression assessed by qRT-PCR in PaTu-8902 and PaTu-8988T cells treated as indicated (5 μM BQ, 24 hours). Expression levels are normalized to *GAPDH* and presented as mean ± s.d. of 3 independent replicates (representative of three independent experiments). Significance determined by t-test, **** p<0.0001. (b) Immunoblot analysis of p65 (RelA subunit of NF-κB) and p105 (precursor of the NF-κB p50 subunit) activation in lysates from PaTu-8902 and PaTu-8988T cells treated for 24 and 48 hours with BQ (5 μM).

These data suggest that the pro-apoptotic shift induced by DHODHi may be regulated at the transcriptional level. We investigated whether DHODH inhibition has been previously linked to transcription factor pathways that regulate apoptosis genes. Prior research demonstrated that DHODH inhibition in the liver of mice downregulated NF-κB activity, a known pro-survival transcription factor (PMID: 15455409, PMID: 9973483). Similarly, patients with Miller Syndrome, caused by mutation of DHODH, have decreased NF-κB pathway activity (PMID: 19915526). NF-κB is a pro-survival effector that increases the expression of anti-apoptotic genes, including *BCL-X_L*, *MCL1*, and *BCL2* (PMID: 32231206; PMID: 29379212). Conversely, downregulation of NF-κB is associated with increased sensitivity to apoptosis (PMID: 16751281).

Interestingly, our GSEA proteome analysis of BQ-treated PDAC lines demonstrated significant downregulation of pathways associated with TLR-related innate immunity and inflammation (Fig. 1f, Supplementary Fig. 2e, Supplementary Table 3), which activate or are regulated by NF-κB transcription factor activity, respectively. To directly evaluate NF-κB pathway activity in response to DHODHi, we analyzed phosphorylation of p65 and p105, key proteins in the canonical and non-canonical NF-κB pathways, respectively. PDAC cells treated with DHODHi demonstrated a decrease in phosphorylation of both p65 and p105, consistent with decreased activation of the NF-κB pathway (Response Fig. 1b; Fig. 4g). Based on these findings, we propose a model in which DHODHi suppresses NF-κB activity, leading to a pro-apoptotic transcriptional response. This transcriptional shift downregulates anti-apoptotic genes, such as *BCL-X_L*, and upregulates pro-apoptotic genes, sensitizing PDAC cells to *BCL-X_L* inhibition and apoptosis induction.

To evaluate the hypothesis that DHODH inhibition increases sensitivity to *BCL-X_L* targeting by affecting mitochondrial function and thereby mitochondrial apoptosis priming, we performed new experiments to measure Oxygen Consumption Rate (OCR) and membrane potential of PDAC cells treated with DHODHi (Response Fig. 2; Supplementary Fig. 1e-j). As measured by Seahorse, DHODHi impaired OCR at 24 h but to a lesser extent than complex I inhibition with IACS-010759. This is in line with previous studies showing a role for DHODH in respiration (PMID: 38547260). Long-term DHODHi treatment demonstrated more profound inhibitory effects suggesting resistance to DHODHi is not related to reactivation of respiration in these cells.

An increase in mitochondrial membrane potential is considered an early event in apoptosis that may be independent from caspase activation, precedes outer mitochondrial membrane disruption, and is associated with increased ROS production (PMID: 9973403, 9393856). As DHODHi decreased GSH/GSSG ratios (Fig. 1c),

indicative of a decrease in antioxidant capacity, we next measured the effects of DHODHi on mitochondrial membrane potential by Tetramethylrhodamine methyl ester (TMRM) staining in PDAC cells. DHODHi induced an increase in mitochondrial membrane potential (**Response Fig. 2g-h; Supplementary Fig. 8a-b**) suggesting that DHODHi-induced mitochondrial dysfunction and hyperpolarization may sensitize PDAC cells to apoptosis inducers. Interestingly, BCL-X_L is known to prevent changes in mitochondrial membrane potential in response to stimuli (PMID: 21987637, 9393856); therefore, whether the increase in mitochondrial membrane potential is due to a direct effect of DHODHi on mitochondrial respiration or antioxidant function or is a downstream consequence of modulation in BCL-X_L levels is unclear and will be the subject of future investigation.

a - (also Supp Fig. 1e) **b - (also Supp Fig. 1f)** **c - (also Supp Fig. 1g)** **d - (also Supp Fig. 1h)**

e - (also Supp Fig. 1i) **f - (also Supp Fig. 1j)** **g - (also Supp Fig. 8a)** **h - (also Supp Fig. 8b)**

Response Figure 2 DHODHi impairs mitochondrial respiration and mitochondrial membrane potential (a-b) Oxygen Consumption Rate of PaTu-8988T and PaTu-8902 cells treated with BQ for 24 h or 7 days. The complex I inhibitor IACS-010759 was used as a positive control for impaired mitochondrial respiration. **(c-f)** Measurements of basal respiration and ATP production for PaTu-8988T and PaTu-8902 cells. Error bars represent s.d. of six technical replicates (one representative of three independent experiments). Significance was determined with t-test. **** p<0.0001. **(g-h)** Mitochondrial membrane potential was measured in PaTu-8902 **(g)** and PaTu-8988T **(h)** after 24 h or 7 days of treatment with Brequinar by flow cytometry. Mean fluorescence Intensity (MFI) of TMRM (Tetramethylrhodamine methyl ester) was calculated by subtracting fluorescence of FCCP-treated cells, normalizing to mitochondrial content measured with MitoTracker Green and normalized to DMSO. Error bars represent s.d. of average of 6 independent experiments. Significance was determined by t-test. **p < 0.01, ***p < 0.001, **** p < 0.0001.

Finally, we explored whether co-dependency with BCL-X_L may be related to disruption of mitochondrial respiration by DHODHi⁸. IACS-010759 in combination with DT2216 showed no additive effects on growth or Annexin V staining (**Response Fig. 3a-d; Supplementary Fig. 7i-l**) suggesting DHODHi effects on mitochondrial respiration alone are not responsible for sensitization to BCL-X_L inhibition.

Response Figure 3 Complex I inhibitor does not recapitulate the pro-apoptotic phenotype induced by DHODHi (a-b) Percentage cell viability of PDAC cells after treatment with increasing concentrations of IACS-010759 with DT2216 for 5 days. IC₅₀ values are shown for a representative experiment out of two independent experiments. **(c-d)** Real-time accumulation of Annexin V fluorescence in PaTu-8902 and PaTu-8988T cells was monitored using the Incucyte system. Cells were labeled with Annexin V Red Dye and treated with DT2216 (2 μM) or IACS-010759 (10 nM) alone or in combination for 72 hours. Error bars represent s.d. of three technical replicates (representative of two experiments). Significance was determined using ordinary one-way ANOVA. ****p < 0.0001.

We have summarized these new findings in the discussion as follows:

“We propose a model in which pyrimidine synthesis inhibition suppresses NF-κB activity, leading to a pro-apoptotic transcriptional response. This transcriptional shift downregulates anti-apoptotic genes, such as *BCL-X_L*, and upregulates pro-apoptotic genes, sensitizing PDAC cells to *BCL-X_L* degradation and promoting apoptosis. A similar profile of anti-apoptotic protein alteration was previously demonstrated in response to gemcitabine suggesting a conserved response to nucleotide metabolism targeting³². Functionally, alteration in anti-apoptotic proteins in PDAC, whether by DHODH inhibition or gemcitabine, conferred increased sensitivity specifically to *BCL-X_L* targeting. Given DHODHi is also associated with an increase in mitochondrial membrane potential, it is possible that in addition to altering NF-κB pathway activity to increase apoptosis sensitivity, DHODHi also primes cells for apoptosis by inducing mitochondrial dysfunction linked to ROS accumulation and mitochondrial hyperpolarization. Whether the increase in mitochondrial membrane potential is due to direct DHODHi effects on mitochondrial respiration and antioxidant function or it is a downstream consequence of modulation in *BCL-X_L*⁵⁵ will require further investigation.”

1a- It would be interesting to assess whether the pro-apoptotic effects upon DHODH inhibition and *BCL-X_L* degradation could be rescued or partially rescued by adding nucleosides such as uridine/cytidine.

This is an excellent point by the reviewer. In our original manuscript, we demonstrated that the addition of uridine to PDAC cells treated with combination DHODH inhibition and *BCL-X_L* degradation partially rescued cell growth and Annexin V staining (**Fig. 3e-f, Supplementary Fig. 6h-m**). In new experiments, we have now expanded on our original findings by performing metabolic rescues of DHODH inhibition and *BCL-X_L* degradation using exogenous uridine (100 μM) to measure effects on cell death as measured by 1) propidium iodide(PI)/Annexin V staining and 2) cleaved PARP immunoblotting. First, our new data demonstrates that the addition of uridine

rescues apoptosis in response to BQ monotherapy in PaTu-8988T, as expected, but does not rescue the effect of DT2216 monotherapy (**Response Fig. 4a-b; Fig. 3c-d**). This indicates that the effects of DT2216 monotherapy are likely independent of pyrimidine metabolism. Importantly, uridine addition partially rescues cell death induced by the combination of BQ and DT2216 back to the level of DT2216 treatment alone in both PaTu-8988T and PaTu-8902 (**Response Fig. 4a-b; Fig. 3c-d**). Furthermore, uridine addition rescues cleaved PARP levels in BQ and DT2216 treated cells back to the level of cleaved PARP seen with DT2216 alone (**Response Fig. 4c; Supplementary Fig. 8i**). Overall, uridine was able to rescue the apoptotic phenotype induced by DHODH and BCL-X_L co-targeting.

Response Figure 4 Exogenous uridine rescues the pro-apoptotic phenotype induced by DHODHi. (a-b) Apoptosis induction was assessed using flow cytometry with FITC Annexin V staining in PaTu-8902 and PaTu-8988T cells following treatment with DMSO, DT2216 (2 μ M), or BQ (5 μ M), either alone or in combination, with or without 100 μ M uridine for 72 hours. The percentage of apoptotic cells was determined as the sum of Annexin V-positive and Annexin V / propidium iodide double-positive populations, representing early and late apoptosis, respectively. Error bars represent s.d. of three technical replicates (representative of two experiments). Significance was determined with ordinary one-way ANOVA test. *** $p < 0.001$, **** $p < 0.0001$. **(c)** Immunoblot analysis of BCL-X_L and cleaved PARP in lysates from PaTu-8902 and PaTu-8988T cells treated with DMSO, DT2216 (5 μ M) or BQ (5 μ M) alone or in combination with or without uridine (100 μ M) for 24 hours.

1b- It would be valuable to assess whether purine synthesis inhibitors could mimic the effects of DHODH inhibitors on BCL-X_L expression or the pro-apoptotic phenotype.

We thank the reviewer for the suggestion to address this important point. To assess if targeting purine synthesis could also sensitize to BCL-X_L degraders, we used Pelitrexol, (AG-2037), a purine synthesis inhibitor targeting GART (phosphoribosylglycinamide formyltransferase), a trifunctional enzyme that catalyzes multiple steps in the purine synthesis pathway (PMID: 38823389). AG-2037 alone decreased proliferation of PDAC cells (**Response Fig. 5a-b; Supplementary Fig. 7e-f**) but did not increase Annexin V staining (**Response Fig. 5c-d; Supplementary Fig. 7g-h**). This data is in agreement with previous studies using in vivo CRISPR screens that defined purine and pyrimidine synthesis as essential pathways for PDAC growth (PMID: 33152323, PMID: 33152324). Combination treatment of AG-2037 and DT2216 demonstrated synergistic effects on growth (**Response Fig. 5a-b; Supplementary Fig. 7e-f**) and Annexin V staining (**Response Fig. 5c-d; Supplementary Fig. 7g-h**) in two human PDAC cell lines. However, purine synthesis inhibition did not recapitulate the shift in BCL2 family proteins observed with pyrimidine synthesis inhibitors (**Response Fig. 5e; Supplementary Fig. 8f**), suggesting that even though synergy with BCL-X_L targeting may be a commonality with nucleotide deprivation in PDAC, the mechanism of apoptosis sensitization differs between pyrimidine and purine metabolism inhibition.

a - (also Supp Fig. 7e)**b - (also Supp Fig. 7f)****c - (also Supp Fig. 7g)****d - (also Supp Fig. 7h)****e - (also Supp Fig. 8f)**
Response Figure 5 Targeting purine synthesis sensitizes to BCL-X_L degraders. (a-b) Percentage viability of PaTu-8902 and PaTu-8988T cells after treatment with increasing concentrations of AG-2037 with DT2216 for 5 days. IC₅₀ values are shown for a representative experiment out of two independent experiments. (c-d) Real-time accumulation of Annexin V fluorescence in PaTu-8902 and PaTu-8988T cells was monitored using the Incucyte system. Cells were labeled with Annexin V Red Dye and treated with the of DT2216 (2 µM) or AG-2037 (0.1 µM) alone or in combination for 72 hours. Error bars represent s.d. of three technical replicates (representative of two experiments). significance determined with ordinary one-way ANOVA test. ***p < 0.001. (e) Immunoblot analysis of BCL-2 family proteins in lysates from PaTu-8988T and PaTu-8902 cells treated with indicated concentrations of AG-2037 for 24 hours.

1c- Using inhibitors of pyrimidine synthesis beyond DHODH inhibitors could also validate whether this effect is specific to DHODH. For example, the authors could use pyrazofurin, a UMPS inhibitor (PMID: 29723133), to assess BCL-X_L expression and the pro-apoptotic phenotype.

We thank the reviewer for this excellent question and suggestion. To define if sensitization to BCL-X_L targeting is specific to DHODH inhibition or more broadly related to pyrimidine synthesis inhibition, we have performed new experiments using pyrazofurin, an inhibitor of UMPS that targets pyrimidine synthesis downstream of DHODH (PMID: 38547260). UMPSi monotherapy recapitulated the effects of DHODHi as it decreased growth of PDAC cells (Response Fig. 6a-b; Supplementary Fig. 7a-b) without significant induction of cell death (Response Fig. 6c-d; Supplementary Fig. 7c-d). Pyrazofurin also decreased expression of BCL-2 family proteins similar to what we observed with BQ treatment (Response Fig. 6e; Supplementary Fig. 8e). Finally, UMPSi synergized with DT2216 in growth suppression and Annexin V staining (Response Fig. 6a-d; Supplementary Fig. 7a-d). Overall, our data suggests that sensitization to BCL-X_L targeting is not specific to DHODH inhibition alone but rather is an effect of pyrimidine synthesis pathway inhibition more broadly.

a- (also Supp Fig. 7a)**b- (also Supp Fig. 7b)****c- (also Supp Fig. 7c)****d- (also Supp Fig. 7d)****e- (also Supp Fig. 8e)**
Response Figure 6 Targeting pyrimidine synthesis with UMPSi sensitizes to BCL-X_L degraders. (a-b) Percentage cell viability of PDAC cells after treatment with increasing concentrations of pyrazofurin with DT2216 for 5 days. IC₅₀ values are shown for a representative experiment out of two independent experiments. **(c-d)** Real-time accumulation of Annexin V fluorescence in PaTu-8902 and PaTu-8988T cells was monitored using the Incucyte system. Cells were labeled with Annexin V Red Dye and treated with DT2216 (1 µM) or pyrazofurin (1 µM) alone or in combination for 66 hours **(c)** and 72 hours **(d)**. Error bars represent s.d. of three technical replicates (representative of two experiments). significance determined with ordinary one-way ANOVA test. ***p < 0.001. **(e)** Immunoblot analysis of BCL-2 family proteins in lysates from PaTu-8902 and PaTu-8988T cells treated with indicated concentrations of pyrazofurin for 24 hours.

2) To ensure that the effects of DHODH inhibitors are not related to ETC inhibition, it is important to compare the pro-apoptotic effects of inhibitors targeting mitochondrial complexes, such as complex I inhibitors, in combination with the BCL-XL degrader.

We thank the reviewer for this important consideration. DHODH couples oxidation of dihydroorotate to orotate with reduction of ubiquinone to ubiquinol (PMID: 20566882) and therefore, is required to support mitochondrial respiration. In addition, recent studies have shown pyrimidines support pyruvate dehydrogenase and complex I function (PMID: 38547260). To understand if the DHODHi-induced pro-apoptotic effects were due to nucleotide depletion or to inhibition of mitochondrial respiration, we performed combinatorial studies using IACS-010759, a complex I inhibitor (PMID: 36658425). IACS-010759 alone impaired PDAC cell growth (**Response Fig. 3a-b (above); Supplementary Fig. 7i-j**) without an increase in Annexin V staining (**Response Fig. 3c-d; Supplementary Fig. 7k-l**). However, combination treatment of DT2216 and IACS-010759 demonstrated no synergistic effects on relative growth or Annexin V induction (**Response Fig. 3a-d; Supplementary Fig. 7i-l**). Together with our studies using UMPSi, we conclude that the pro-apoptotic effects observed with DHODHi are most likely related to depletion of nucleotide pools and not to impaired mitochondrial respiration.

Reviewer #2 (Remarks to the Author):

In manuscript “De novo pyrimidine biosynthesis inhibition synergizes with BCL-XL targeting in pancreatic cancer”, the authors identified *BCL2L1* gene (Bcl-XL) as a DHODH inhibition resistance gene using integrated multi-omics approach and CRISPR/Cas9 loss-of-function screen with DHODH inhibitor brequinar (BQ). Authors found that combined inhibition of DHODH and Bcl-XL, using BQ and Bcl-XL inhibitor DT2216, synergistically promotes apoptosis in various pancreatic cancer cells and suppresses PDAC growth in patient-derived organoids and in vivo models of PDAC. Interestingly, BQ promotes synergistic responses exclusively with BCL-XL inhibitor but not inhibitors of BCL2 or MCL1. These findings are interesting and significant. The manuscript is, for most part, clearly written and experiments are presented in a logical way. However, the exact molecular mechanism by which Bcl-XL regulates BQ resistance remains unclear. Before publication of this manuscript in Nature communications, the following issues should be addressed:

We thank the reviewer for their positive comments.

1. CRISPR screen shows that *BCL2L1* (Bcl-XL) and *MCL1* are the top candidates contributing to BQ resistance in pancreatic cells (i.e. PaTu-8988T) (Fig. 1h), but in vivo screen using the same cell line only shows *BCL2L1* (Bcl-XL) not *MCL1* (Fig. 2f). Why data from in vitro and in vivo CRISPR screen are inconsistent, since xenografts were derived from same cell line PaTu-8988T as in vitro.

We thank the reviewer for this question. We designed our study to prioritize investigation of hits that validated in both in vitro and in vivo growth conditions. The particular reason that we included an in vivo screen in our study was noted in our original manuscript: “To account for the physiological differences in uridine levels as well as any other differences between in vitro culture conditions and in vivo conditions, we designed a focused sgRNA library to evaluate high priority hits in a BQ-anchored mini-pool in vivo CRISPR/Cas9 screen.”

As noted by the reviewer, *BCL2L1* and *MCL1* were hits in our in vitro screens. However, *BCL2L1* but not *MCL1* validated in our in vivo screens. While the exact reason for this difference is unclear, we speculate that it is due to different level of dependency on *MCL1* in in vitro versus in vivo growth conditions and in response to DHODHi in in vitro versus in vivo growth conditions. First, in PDAC in vivo models (including patients), BCL-X_L has been demonstrated to be the most important anti-apoptosis BCL-2 family member in comparison to *MCL1*. In our original manuscript, we highlighted prior research to this effect:

“Previous studies have shown that BCL-X_L expression increased from the preinvasive pancreatic intraepithelial (PanIN) stage to invasive PDAC in mouse models and patient tumors²⁶. Importantly, compared to other anti-apoptotic proteins, higher levels of BCL-X_L were observed in 90% of tumors from PDAC patients²⁷.”

Second, it was recently demonstrated that metabolic vulnerabilities of cancer cells, including PDAC, differ in vitro versus in vivo largely due to differences in metabolite availability in cell culture conditions versus in vivo (PMID: 28388410; PMID: 30990168, PMID: 33152323; PMID: 33152324). Therefore, it is possible that some of the hits identified in the DHODHi anchored in vitro screen, such as *MCL1*, might not appear as significant in the DHODHi anchored in vivo screen due to differences in the in vivo metabolic environment.

In summary, we purposefully designed our experiment to prioritize hits with in vivo relevance and therefore, we focused on *BCL2L1* instead of *MCL1* for in vivo validation. As previously published, BCL-X_L is the critical anti-apoptotic BCL2 family member in vivo and our results further demonstrate that BCL-X_L is the critical node for combination targeting with DHODH inhibition in vivo.

2. Why BQ only synergizes with Bcl-XL inhibitor DT2216 but not *MCL1* inhibitor since *MCL1* is also the top candidate in in vitro CRISPR screen.

The reviewer brings up an important point. While *MCL1* was a candidate based on the in vitro CRISPR screens, our synergy experiments did not show additive effects when BQ was paired with an *MCL1* inhibitor. Of note, the results of our CRISPR screen are a read-out of genetic deletion of *MCL1* and not pharmacologic inhibition. To

investigate if there are potential differences between inhibiting MCL1 function or depleting the protein, we performed new experiments using a compound reported to be a MCL1 PROTAC degrader (AZD-5991) (PMID: 30559424, PMID: 39167622). However, in our experiments in PDAC cells using this compound, we were unable to demonstrate MCL1 degradation (data not shown) limiting our ability to address this question directly. We hypothesize that the discordance between BQ synergizing with *MCL1* genetic ablation in the CRISPR/Cas9 screen but not MCL1 pharmacologic inhibition may reflect that pharmacologic targeting of the MCL1 BH3-binding domain does not inhibit non-apoptosis related functions of MCL1, such as maintenance of mitochondrial dynamics and bioenergetics, whereas genetic ablation targets all MCL1 activities (PMID: 34475520). Further investigation of this hypothesis was outside the scope of our study given the focus in our manuscript on targets that were validated in vivo (e.g. BCL-X_L).

3. What exact molecular mechanism(s) by Bcl-X_L regulates BQ resistance in pancreatic cancer cells.

This is an excellent point that was also brought up by Reviewer 1. We have now performed further experiments to define the mechanistic links between DHODHi and sensitivity to apoptosis. We include here the response we provided to Reviewer 1 for ease of review:

Taking into consideration Reviewer #1's comments and those of the other reviewers, we investigated two hypotheses: 1) that DHODH inhibition increases sensitivity to BCL-X_L targeting by altering the activity of a transcriptional regulator of apoptosis genes, and 2) that DHODH inhibition alters mitochondrial function thereby enhancing apoptosis sensitivity.

In our original manuscript, we showed that DHODH inhibition (DHODHi) downregulated protein-level expression of BCL-2-family proteins as determined by quantitative proteomics and immunoblotting (**Fig. 4c, Supplementary Fig. 8c-d, Supplementary Table 2**). To understand if the changes at the proteome level correlated with transcriptional responses, in new revision experiments, we quantified expression of apoptosis-related genes by qRT-PCR (**Response Fig. 1a; included in the manuscript as Fig. 4f**). Consistent with protein-level expression changes, PDAC cells treated with DHODHi downregulated expression of anti-apoptotic genes (*BCL2L1* (BCL-X_L), *MCL1* and *BCL2*) and upregulated expression of pro-apoptotic genes (*BCL2L11* (*BIM*)).

Response Figure 1 DHODHi modulates expression of apoptotic genes by impairing activation of the NF-κB pathway. (a) Apoptosis regulatory gene expression assessed by qRT-PCR in PaTu-8902 and PaTu-8988T cells treated as indicated (5 μM BQ, 24 hours). Expression levels are normalized to *GAPDH* and presented as mean ± s.d. of 3 independent replicates (representative of three independent experiments). Significance determined by t-test, **** p<0.0001. **(b)** Immunoblot analysis of p65 (RelA subunit of NF-κB) and p105 (precursor of the NF-κB p50 subunit) activation in lysates from PaTu-8902 and PaTu-8988T cells treated for 24 and 48 hours with BQ (5 μM).

These data suggest that the pro-apoptotic shift induced by DHODHi may be regulated at the transcriptional level. We investigated whether DHODH inhibition has been previously linked to transcription factor pathways that regulate apoptosis genes. Prior research demonstrated that DHODH inhibition in the liver of mice downregulated

NF- κ B activity, a known pro-survival transcription factor (PMID: 15455409, PMID: 9973483). Similarly, patients with Miller Syndrome, caused by mutation of DHODH, have decreased NF- κ B pathway activity (PMID: 19915526). NF- κ B is a pro-survival effector that increases expression of anti-apoptotic genes, including *BCL-X_L*, *MCL1*, and *BCL2* (PMID: 32231206; PMID: 29379212). Conversely, downregulation of NF- κ B is associated with increased sensitivity to apoptosis (PMID: 16751281).

Interestingly, our GSEA proteome analysis of BQ-treated PDAC lines demonstrated significant downregulation of pathways associated with TLR-related innate immunity and inflammation (**Fig. 1f**, **Supplementary Fig. 2e**, **Supplementary Table 3**), which activate or are regulated by NF- κ B transcription factor activity, respectively. To directly evaluate NF- κ B pathway activity in response to DHODHi, we analyzed phosphorylation of p65 and p105, key proteins in the canonical and non-canonical NF- κ B pathways, respectively. PDAC cells treated with DHODHi demonstrated a decrease in phosphorylation of both p65 and p105, consistent with decreased activation of the NF- κ B pathway (**Response Fig. 1b**; **Fig. 4g**). Based on these findings, we propose a model in which DHODHi suppresses NF- κ B activity, leading to a pro-apoptotic transcriptional response. This transcriptional shift downregulates anti-apoptotic genes, such as *BCL-X_L*, and upregulates pro-apoptotic genes, sensitizing PDAC cells to *BCL-X_L* inhibition and apoptosis induction.

To evaluate the hypothesis that DHODH inhibition increases sensitivity to *BCL-X_L* targeting by affecting mitochondrial function and thereby mitochondrial apoptosis priming, we performed new experiments to measure Oxygen Consumption Rate (OCR) and membrane potential of PDAC cells treated with DHODHi (**Response Fig. 2**; **Supplementary Fig. 1e-j**). As measured by Seahorse, DHODHi impaired OCR at 24 h but to a lesser extent than complex I inhibition with IACS-010759. This is in line with previous studies showing a role for DHODH in respiration (PMID: 38547260). Long-term DHODHi treatment demonstrated more profound inhibitory effects suggesting resistance to DHODHi is not related to reactivation of respiration in these cells.

An increase in mitochondrial membrane potential is considered an early event in apoptosis that may be independent from caspase activation, precedes outer mitochondrial membrane disruption, and is associated with increased ROS production (PMID: 9973403, 9393856). As DHODHi decrease GSH/GSSG ratios (**Fig. 1c**), indicative of a decrease in antioxidant capacity, we next measured effects of DHODHi on mitochondrial membrane potential by Tetramethylrhodamine methyl ester (TMRM) staining in PDAC cells. DHODHi induced an increase in mitochondrial membrane potential (**Response Fig. 2g-h**; **Supplementary Fig. 8a-b**) suggesting that DHODHi-induced mitochondrial dysfunction and hyperpolarization may sensitize PDAC cells to apoptosis inducers. Interestingly, *BCL-X_L* is known to prevent changes in mitochondrial membrane potential in response to stimuli (PMID: 21987637, 9393856); therefore, whether the increase in mitochondrial membrane potential is due to a direct effect of DHODHi on mitochondrial respiration or antioxidant function or is a downstream consequence of modulation in *BCL-X_L* levels is unclear and will be the subject of future investigation.

a - (also Supp Fig. 1e)**b - (also Supp Fig. 1f)****c - (also Supp Fig. 1g)****d - (also Supp Fig. 1h)****e - (also Supp Fig. 1i)****f - (also Supp Fig. 1j)****g - (also Supp Fig. 8a)****h - (also Supp Fig. 8b)**
Response Figure 2 DHODHi impairs mitochondrial respiration and mitochondrial membrane potential (a-b) Oxygen Consumption Rate of PaTu-8988T and PaTu-8902 cells treated with BQ for 24 h or 7 days. The complex I inhibitor IACS-010759 was used as a positive control for impaired mitochondrial respiration. **(c-f)** Measurements of basal respiration and ATP production for PaTu-8988T and PaTu-8902 cells. Error bars represent s.d. of six technical replicates (one representative of three independent experiments). Significance was determined with t-test. **** $p < 0.0001$. **(g-h)** Mitochondrial membrane potential was measured in PaTu-8902 **(g)** and PaTu-8988T **(h)** after 24 h or 7 days of treatment with Brequinar by flow cytometry. Mean fluorescence Intensity (MFI) of TMRM (Tetramethylrhodamine methyl ester) was calculated by subtracting fluorescence of FCCP-treated cells, normalizing to mitochondrial content measured with MitoTracker Green and normalized to DMSO. Error bars represent s.d. of average of 6 independent experiments. Significance was determined by t-test. ** $p < 0.01$, *** $p < 0.001$, **** $p < 0.0001$.

Finally, we explored whether co-dependency with BCL- X_L may be related to disruption of mitochondrial respiration by DHODHi⁸. IACS-010759 in combination with DT2216 showed no additive effects on growth or Annexin V staining (**Response Fig. 3a-d; Supplementary Fig. 7i-l**) suggesting DHODHi effects on mitochondrial respiration alone are not responsible for sensitization to BCL- X_L inhibition.

Response Figure 3 Complex I inhibitor does not recapitulate the pro-apoptotic phenotype induced by DHODHi (a-b) Percentage cell viability of PDAC cells after treatment with increasing concentrations of IACS-010759 with DT2216 for 5 days. IC50 values are shown for a representative experiment out of two independent experiments. **(c-d)** Real-time accumulation of Annexin V fluorescence in PaTu-8902 and PaTu-8988T cells was monitored using the Incucyte system. Cells were labeled with Annexin V Red Dye and treated with DT2216 (2 μM) or IACS-010759 (10 nM) alone or in combination for 72 hours. Error bars represent s.d. of three technical replicates (representative of two experiments). Significance was determined using ordinary one-way ANOVA. ****p < 0.0001.

We have summarized these new findings in the discussion as follows:

“We propose a model in which pyrimidine synthesis inhibition suppresses NF-κB activity, leading to a pro-apoptotic transcriptional response. This transcriptional shift downregulates anti-apoptotic genes, such as *BCL-X_L*, and upregulates pro-apoptotic genes, sensitizing PDAC cells to *BCL-X_L* degradation and promoting apoptosis. A similar profile of anti-apoptotic protein alteration was previously demonstrated in response to gemcitabine suggesting a conserved response to nucleotide metabolism targeting³². Functionally, alteration in anti-apoptotic proteins in PDAC, whether by DHODH inhibition or gemcitabine, conferred increased sensitivity specifically to *BCL-X_L* targeting. Given DHODHi is also associated with an increase in mitochondrial membrane potential, it is possible that in addition to altering NF-κB pathway activity to increase apoptosis sensitivity, DHODHi also primes cells for apoptosis by inducing mitochondrial dysfunction linked to ROS accumulation and mitochondrial hyperpolarization. Whether the increase in mitochondrial membrane potential is due to direct DHODHi effects on mitochondrial respiration and antioxidant function or it is a downstream consequence of modulation in *BCL-X_L*⁵⁵ will require further investigation.”

4. Effects of BQ in combination with Bcl-XL inhibitor DT2216, BCL2 inhibitor or MCL1 inhibitor on pancreatic cell growth should be measured by Colony formation assay.

We thank the reviewer for this excellent suggestion. We have now performed new experiments to assess the synergistic effects of BQ and different apoptotic inducers on clonogenic growth. While single treatments did not impair clonogenic ability extensively, combinatorial treatment of BQ and DT2216 significantly reduced colony formation (**Response Fig. 7a-b; Fig. 5g-h**). Interestingly, combination treatment with BQ and BCL2 or MCL1 inhibitors did not demonstrate synergistic effects on colony formation. Overall our results agree with our previous

relative growth assays (**Fig. 5a-f**) demonstrating a synergistic effect on colony formation with combination BQ and BCL-X_L inhibition but not with BQ and BCL2 or MCL1 inhibition (**Response Fig. 7a-b; Fig. 5g-h**).

Response Figure 7 Synergistic effects with BQ are specific to BCL-X_L inhibition in clonogenic assays. (a-b) Clonogenic growth of PDAC cells treated with BQ (5 μM), DT2216 (5 μM), Venetoclax (5 μM) or AZD-5991 (5 μM) alone or in combination with BQ. Error bars indicate s.d. of three technical replicates in three independent experiments. Significance was determined with t-test. *p < 0.05, **p < 0.01, ***p < 0.001, ****p < 0.0001.

5. The DHODH inhibitor BQ has been reported to have immunosuppressive and serious side effects. Therefore, combination of BQ and DT2216 in animal models, authors should observe immune profile and in vivo toxicity in more details, including blood cells (WBC, RBC, PLT), liver function(ALT, AST), kidney function(BUN), heart function, etc.

We thank the reviewer for highlighting this important point. As noted in the original manuscript, hematologic parameters were largely unaffected in NSG mice treated with the combination (**Supplementary Table 10**) whereas C57BL/6 mice treated with the combination demonstrated moderate anemia and thrombocytopenia (**Supplementary Table 9**). Notably, there was no significant decrease in WBC in the combination treatment compared to vehicle treated mice (**Supplementary Table 9**). To investigate the potential for non-hematologic toxicity, we have performed new serum analyses of tumor-bearing C57BL/6 mice treated with vehicle, BQ, DT2216, or combination treatment (**Response Fig. 8a-c; Supplementary Fig. 11a-c**). While serum ALT levels were mildly elevated in BQ-treated mice, neither DT2216 nor the combination of BQ and DT2216 significantly altered ALT levels. As such, our data demonstrate that the combination treatment did not significantly impact liver function (**Response Fig. 8a; Supplementary Fig. 11a**). Serum BUN levels were mildly elevated in the combination-treated group; however, they remained within the normal range, indicating no significant effect on kidney function (**Response Fig. 8b; Supplementary Fig. 11b**). Finally, we measured serum Brain Natriuretic Peptide (BNP) levels, a marker of heart failure and myocardial stress. No significant changes in BNP levels were observed in any of the treatment groups (**Response Fig. 8c; Supplementary Fig. 11c**).

Response Figure 8 Toxicity evaluation of BQ Combined with DT2216 in C57BL/6J Mice. (a) Serum levels of alanine aminotransferase (ALT) in tumor-bearing C57BL/6J mice following treatment with Vehicle, BQ (10 mg/kg), DT2216 (15 mg/kg), or their combination. Error bars represent s.d. of 3 mice. (b) Serum levels of blood urea nitrogen (BUN) in tumor-bearing C57BL/6J mice following treatment with Vehicle, BQ (10 mg/kg), DT2216 (15 mg/kg), or their combination (normal range in C57BL/6J = 2–71 mg/dL). Error bars represent s.d. of 3 mice. (c) Serum levels of brain natriuretic peptide (BNP) in tumor-bearing C57BL/6J mice following treatment with Vehicle, BQ (10 mg/kg), DT2216 (15 mg/kg), or their combination. Error bars represent s.d. of 3 mice. Statistical significance was determined by ordinary one-way ANOVA test. * $p < 0.05$.

Overall, our data suggests BQ and DT2216 induces moderate anemia and mild renal toxicity. Together, this toxicity profile is in line with other standard of care cytotoxic chemotherapy regimens. We have updated our manuscript with these new findings as follows (line 455):

“While hematologic parameters were largely unaffected in NSG mice, the BQ and DT2216 combination demonstrated modest anemia and thrombocytopenia in C57BL/6 mice (**Supplementary Table 9-10**). To evaluate potential toxicity of the combination beyond effects on hematological parameters, we performed serum-based evaluations of the C57BL/6 subcutaneous syngenic model. There were no significant elevations in serum ALT in the combination treatment arm suggesting no effect on liver function (**Supplementary Fig. 11a, Supplementary Table 11**). While serum BUN levels were mildly elevated in the combination-treated group, they remained within the normal range, indicating no significant effect on kidney function (**Supplementary Fig. 11b**). Finally, there were no significant changes in serum Brain Natriuretic Peptide (BNP) levels, a marker of heart failure and myocardial stress, observed in any of the treatment groups, (**Supplementary Fig. 11c**).”

As discussed in our original manuscript, we envision multiple future directions for mitigating toxicity of this combination approach.

“One strategy to mitigate potential hematologic side effects of the BQ and DT2216 combination is administration of recombinant G-CSF (filgrastim, pegfilgrastim) and romiplostim, common supportive treatments administered with FOLFIRINOX chemotherapy⁵¹. Although G-CSF induced neutrophils can suppress T cell activity,⁶⁴ we have shown that the combination of BQ and DT2216 does not require adaptive immunity. Notably, BQ-treated BCL-X_L KO tumor-bearing mice exhibited significant tumor growth inhibition and minimal treatment related toxicity (**Fig. 6e-f**), suggesting that a strategy combining tumor-specific BCL-X_L targeting with systemic DHODH inhibition (DHODHi) offers a promising therapeutic approach with the potential for reduced toxicity. Indeed, targeting BCL-X_L inhibitors to tumors via an antibody–drug conjugate (ADC) approach is being evaluated as a strategy to minimize non-tumor tissue effects⁶⁵. ABBV-637, an ADC consisting of a monoclonal antibody directed against the epidermal growth factor receptor (EGFR) conjugated to an inhibitor of BCL-X_L, and ABBV-155, an ADC composed of a monoclonal antibody against the immunoregulatory protein B7-homologue 3 (B7-H3, CD276) conjugated to a BCL-X_L inhibitor are now in Phase 1 clinical trials for lung cancer treatment (NCT04721015 and NCT03595059⁶⁶).”

As a proof of concept of the anti-tumor combination efficacy of systemic DHODH inhibition and tumor-specific targeting of BCL-X_L, we used CRISPR/Cas9-based gene editing to generate BCL-X_L knockout (KO) clones in PaTu-8902 PDAC cells (**Response Fig. 9a; Supplementary Fig. 5a**). While BCL-X_L KO alone did not significantly impact cell growth in vitro (**Response Fig. 9b; Supplementary Fig. 5b**), treatment with BQ demonstrated a significant decrease in cell proliferation (**Response Fig. 9c; Supplementary Fig. 5c**) and increased Annexin V staining compared to PaTu-8902 cells transduced with a non-targeting sgRNA (sgControl) (**Response Fig. 9d-e; Supplementary Fig. 5d-e**). These results are consistent with our data using BCL-X_L inhibitors and degraders (**Supplementary Fig. 9a-d**). To assess the potential for tumor-specific targeting of BCL-X_L in vivo, we implanted sgControl (non-targeting) and BCL-X_L KO cells in mice and treated with vehicle or BQ. BCL-X_L KO tumors grew slightly slower compared to non-targeting tumors (**Response Fig. 9f-g; Fig. 6e-f**). BQ-treated BCL-X_L KO tumor-bearing mice demonstrated significant tumor growth inhibition compared to BQ-treated sgControl tumor bearing mice (**Response Fig. 9f-g; Fig. 6e-f**). Importantly, mice maintained stable body weight throughout treatment (**Supplementary Fig. 10b**). Additionally, BQ treatment in mice with BCL-X_L KO tumors did not lead to significant anemia or thrombocytopenia (**Supplementary Table 8**). Altogether, this suggests that a

strategy combining systemic DHODH inhibition with tumor-specific BCL-X_L targeting, such as with an antibody drug conjugate, offers a promising therapeutic approach with the potential for reduced toxicity.

a - (also Supp Fig. 5a)

b - (also Supp Fig. 5b)

c - (also Supp Fig. 5c)

d - (also Supp Fig. 5d)

e - (also Supp Fig. 5e)

f - (also Fig. 6e)

g - (also Fig. 6f)

Response Figure 9 BCL-X_L KO and BQ treatment show additive effects *in vivo*. (a) Immunoblot analysis of BCL-X_L in non-targeting cells and BCL-X_L KO PaTu-8902 cells (The numbers represent individual clones derived from the cell pool following lentivirus infection and subsequent selection. BCL-X_L KO clones 1-4 were derived from the BCL-X_L sgRNA-1 targeting cell pool. BCL-X_L KO clones 5-8 originated from the BCL-X_L sgRNA-2 targeting cell pool). (b) Cell proliferation was monitored in real-time using the Incucyte® Live-Cell Analysis System and represented as relative growth normalized to day 0 for PaTu-8902 non-targeting cell and BCL-X_L KO over 72 h. Non-targeting-1 and non-targeting-2 refer to clones 1 and 2 in (a) labeled as non-targeting. BCL-X_L KO-1 and BCL-X_L KO-2 correspond to clones 1 and 5 from (a) labeled as BCL-X_L KO. Clones non-targeting -1 and BCL-X_L KO-1 were used in panels (c), (d), (e) and (g). Error bars represent s.d. of three technical replicates (representative of two experiments). (c) Cell proliferation was monitored in real-time using the Incucyte® Live-Cell Analysis System and represented as relative growth normalized to day 0 for PaTu-8902 non-targeting cell and BCL-X_L KO with or without 5 μM BQ over 72 h. Error bars represent s.d. of three technical replicates (representative of two experiments). (d) Real-time accumulation of Annexin V fluorescence in PaTu-8902 non-targeting cells and BCL-X_L KO cells was monitored using the Incucyte system. Cells were labeled with Annexin V Red Dye and treated with the indicated concentrations of 5 μM BQ for 72 hours. Error bars represent s.d. of three technical replicates (representative of two experiments). (e) Immunoblot analysis of BCL-X_L and cleaved PARP in non-targeting cells and BCL-X_L KO PaTu-8902 with or without 5 μM BQ. (f) Experimental design of (g). (g) Tumor volume of PaTu-8902 BCL-X_L KO tumors (non-targeting n=7, non-targeting + BQ n=7, BCL-X_L KO n=7, BCL-X_L KO +BQ n=9 per arm) after treatment with vehicle or 10 mg/kg BQ three times a week for 3 weeks. Tumors were measured twice a week. Error bars represent ± s.e.m. For all panels, statistical significance was determined by ordinary one-way ANOVA test. *p < 0.05, **p < 0.01, ***p < 0.001, ****p < 0.0001.

Reviewer #3 (Remarks to the Author):

This study by Zhang et al. is both interesting and clinically relevant. The authors conducted high-quality in vitro and in vivo screens, identifying BCL2L1 as a synergistic target with DHODH inhibition. Combination of DHODH and BCL2L1 inhibitors demonstrated a synergetic effect against cancer in both in vitro and in vivo settings. However, several weaknesses in the manuscript need to be addressed before it can be considered for publication.

We thank the reviewer for their positive comments.

Major:

1. The manuscript partially overlaps with the published paper: C. Mao et al. *Nature* <https://doi.org/10.1038/s41586-021-03539-7> (2021). Please address the similarities and differences between the two works, and discuss how your findings either complement or diverge from the conclusions of the Mao et al. study.

We thank the reviewer for the suggestion. Mao et al. report that DHODH acts as a ferroptosis defense mechanism via its known role in reducing ubiquinone to ubiquinol in the mitochondria (PMID: 33981038). They propose that ubiquinol acts as a radical-trapping antioxidant in the mitochondria with anti-ferroptosis activity and that DHODH function is especially important in ferroptosis defense in cancers with low GPX4 levels (GPX4 is an anti-ferroptosis glutathione peroxidase). They propose combination DHODH inhibition and sulfasalazine as a ferroptosis inducing anti-cancer strategy (no PDAC models were evaluated in their study). In contrast, our study focuses exclusively on PDAC models and identifies that DHODH inhibition enhances selective vulnerability to BCL-X_L-targeted therapies to induce apoptosis and anti-tumor efficacy. Besides the fact that both studies identify DHODH inhibition as a potential cancer therapy approach, we respectfully feel that there is little significant overlap between these studies.

We would also note important differences in the methodology employed in our respective studies especially as it relates to the concentrations of Brequinar (BQ) used. In their study, Mao et al. employed high concentrations of BQ (500 μ M) that are well beyond the known IC₅₀ for DHODH inhibition (7 nm). These supratherapeutic doses of BQ have been highlighted in a comment in *Nature* by the Conrad Lab as a confounder of the suggested finding that DHODH acts as a ferroptosis defense mechanism (PMID: 37407687). The Conrad Lab identified that at the 500 μ M concentration of BQ used by Mao et al., BQ inhibits not only DHODH but also FSP1, a bona fide ferroptosis suppressor. These data suggest that the findings in the Mao et al. study regarding DHODH as a ferroptosis defense mechanism might stem from off-target FSP1 inhibition by high-dose BQ rather than selective DHODH inhibition. In our study, we specifically employ lower concentrations of BQ (500 nM-5 μ M) to minimize any off-target inhibition of FSP1. Our unbiased screens at these concentrations of BQ highlight enhanced apoptosis sensitivity and not ferroptosis sensitivity as a reproducible result of DHODH inhibition. Furthermore, we have also performed experiments using another DHODH inhibitor, BAY-2402234, which is specific to DHODH and does not inhibit FSP1 (PMID: 37407687). This distinction is crucial, as it indicates that our findings isolate DHODH effects without confounding off-target effects seen at higher concentrations of BQ.

We have updated our Discussion to reflect this distinction between our studies as follows:

“A recent report suggested DHODH acts as a ferroptosis defense mechanism by mitigating mitochondrial lipid peroxidation; however, the doses of Brequinar (500 μ M) used in that study are associated with off-target inhibition of FSP1, an NAD(P)H-dependent oxidoreductase that is a critical cellular ferroptosis defense node^{60,61}. In our study, we used lower concentrations of BQ (0.5-5 μ M), in the range of IC₅₀ for DHODH inhibition, and our unbiased screens highlight enhanced apoptosis sensitivity as a reproducible result of DHODH inhibition in unbiased cell culture and in vivo-based screens. Furthermore, we have also demonstrated synergy between DHODH inhibition and BCL-X_L targeting using a distinct DHODH inhibitor, BAY-2402234, that does not have an off-target effect on FSP1⁶¹.”

2. In Figure 1d and Figure 1e, is BCL2L1 upregulated? If not, please discuss its relevance to your findings. If it is upregulated, please ensure it is clearly labeled in the figures.

Our quantitative proteomic and immunoblot experiments presented in the original manuscript demonstrate that BCL2L1 levels are downregulated at 7 days in PaTu-8988T cell lines (**Fig. 4c, Supplementary Fig. 8c-d**). We have now labelled BCL2L1 in Fig 1d. and 1e (**Response Fig.10; Fig. 1d-1e**).

Response Figure 10 Volcano plot illustrates significant protein abundance differences in PaTu-8988T cells treated with 5 μ M BQ at 24h (a) and 7 days (b). Volcano plots display the $-\log_{10}$ (FDR) versus the \log_2 of the relative protein abundance of mean BQ to DMSO-treated samples. Red circles represent significantly upregulated proteins (\log_2 fold change ≥ 1 , FDR < 0.01), whereas blue circles represent significantly downregulated proteins (\log_2 fold change ≤ -1); data from 3 DMSO or 3 BQ-treated independent plates.

3. What is the library coverage of the library-transduced PDAC cells before and after the screen, as presented in Figures 1 and 2? Providing this information would help clarify the robustness of the screening results.

To ensure robust results from our CRISPR/Cas9 screening, in line with recommended coverage to ensure robust results (PMID: 38048220), we maintained a coverage of 500x for the in vitro whole-genome screen and 1000 x for the in vitro and 2500x for the in vivo mini-pool screens. **Supplementary Table 4** also demonstrates excellent library coverage (99.8% for whole genome screen and above 99.8% for the mini library) across all PDAC cell samples both before and after the screen, underscoring the robustness of our screening results. This consistent representation indicates that each construct was adequately maintained, minimizing potential sampling bias and enhancing the reliability of the identified hits.

Supplementary table 4 library coverage across all PDAC cell samples both before and after the screen.

Whole genome screening (PaTu-8988T)

Label	Total sgRNAs	Zero counts sgRNAs	Coverage
Initial sample - before screening	77441	50	99.935%
DM SO # 1	77441	152	99.804%
DM SO # 2	77441	180	99.768%
DM SO # 3	77441	145	99.813%
BQ low dose # 1	77441	123	99.841%
BQ low dose # 2	77441	113	99.854%
BQ low dose # 3	77441	127	99.836%
BQ high dose # 1	77441	89	99.885%
BQ high dose # 2	77441	78	99.899%
BQ high dose # 3	77441	123	99.841%

Mini-pool screening (PaTu-8988T)

Label	Total sgRNAs	Zero counts sgRNAs	Coverage
Initial sample - before screening	2000	0	100.000%
DM SO # 1	2000	0	100.000%
DM SO # 2	2000	0	100.000%
DM SO # 3	2000	0	100.000%
BQ low dose # 1	2000	0	100.000%
BQ low dose # 2	2000	0	100.000%
BQ low dose # 3	2000	0	100.000%
BQ high dose # 1	2000	0	100.000%
BQ high dose # 2	2000	0	100.000%
BQ high dose # 3	2000	0	100.000%

Mini-pool screening (PANC-1)

Label	Total sgRNAs	Zero counts sgRNAs	Coverage
Initial sample - before screening	2000	0	100.000%
DM SO # 1	2000	1	99.950%
DM SO # 2	2000	0	100.000%
DM SO # 3	2000	0	100.000%
BQ low dose # 1	2000	0	100.000%
BQ low dose # 2	2000	0	100.000%
BQ low dose # 3	2000	0	100.000%
BQ high dose # 1	2000	0	100.000%
BQ high dose # 2	2000	0	100.000%
BQ high dose # 3	2000	1	99.950%

In vivo mini-pool screening (PaTu-8988T)

Label	Total sgRNAs	Zero counts sgRNAs	Coverage
Initial sample - before screening	2000	0	100.000%
Vehicle # 1	2000	0	100.000%
Vehicle # 2	2000	0	100.000%
Vehicle # 3	2000	3	99.850%
Vehicle # 4	2000	2	99.900%
Vehicle # 5	2000	0	100.000%
Vehicle # 6	2000	0	100.000%
Vehicle # 7	2000	0	100.000%
Vehicle # 8	2000	2	99.900%
Vehicle # 9	2000	0	100.000%
Vehicle # 10	2000	1	99.950%
BQ # 1	2000	1	99.950%
BQ # 2	2000	0	100.000%
BQ # 3	2000	0	100.000%
BQ # 4	2000	0	100.000%
BQ # 5	2000	2	99.900%
BQ # 6	2000	0	100.000%
BQ # 7	2000	0	100.000%
BQ # 8	2000	0	100.000%
BQ # 9	2000	1	99.950%
BQ # 10	2000	0	100.000%

4. In Figure 2a, under criterion 3), why did the authors exclude genes that have been published in both *in vitro* and *in vivo* studies?

We apologize for the confusion. Our criteria were to include genes that had been linked to DHODH biology in *in vitro* studies but that had not yet been evaluated in *in vivo* studies. We have rephrased the sentence in the results section to clarify this point (line 219).

“Our custom library contained sgRNAs targeting 369 genes (**Fig. 2a**) that we prioritized based on the following criteria: 1) proteomic hits with FDA drugs available, 2) most significant hits from the CRISPR/Cas9 screen, including depleted and enriched guides (for balancing the library), and 3) genes that had already been linked to DHODH biology in published in vitro studies but that had not yet been evaluated in in vivo studies”.

5. The genes involved in nucleotide salvage (UCK2, SLC29A1, CMPK1) and apoptosis (BCL2L1) are overlapped in both screens. Why did the authors choose not to focus on the nucleotide salvage genes for the subsequent experiments?

We thank the reviewer for bringing up this excellent point and allowing us to clarify this further. We agree that our screens suggest that co-targeting pyrimidine biosynthesis and nucleoside salvage may be a useful therapeutic strategy. However, the potential of this combination has already been explored in several published studies. Mullen et al. showed that co-targeting DHODH with BQ and SLC29A1, a nucleoside importer, with genetic ablation, had additive combinatorial efficacy delaying tumor growth in vitro and in vivo (PMID: 36341997). Combination of BQ and dipyrindamole, an FDA-approved SLC29A1 inhibitor, in colon cancer and PDAC xenograft models showed no additive effect of the combination when lower doses of dipyrindamole were used (PMID: 33344900). However, the combination significantly reduced in vivo growth of MYC-driven neuroblastoma when high dipyrindamole doses were used (PMID: 34462431). These data suggest co-targeting synthesis and salvage might be a plausible anti-cancer strategy; however, currently available SLC29A1 inhibitors lack efficacy and require use at high doses, which increases the risk for potential toxicities. In our original manuscript, we included a discussion of the concept of co-targeting pyrimidine biosynthesis and nucleoside salvage, pasted here for ease of review (line 561):

“Our CRISPR screen identified SLC29A1, a nucleoside importer, as an in vivo co-dependency, in agreement with previous studies showing synergistic in vitro efficacy¹⁴. However, to date there are no potent, selective, and in vivo capable inhibitors available for SLC29A1. Future studies are required to assess the feasibility of this combination since systemic dual targeting of nucleoside import and de novo pyrimidine synthesis may result in systemic toxicities affecting cells with high biosynthetic needs in the bone marrow and intestine as well as most non-tumor cells that mostly rely on salvage.”

6. In Figure 6, the in vivo efficacy of combination targeting of DHODH and BCL-XL appears to be limited.

Combination targeting of DHODH and BCL-X_L demonstrated significant delays in tumor progression compared to vehicle-treated or monotherapy-treated tumors in all of the mouse models tested (**Fig. 6g-m**). However, we do agree there are limitations to our results. First, evaluation of the efficacy of DT2216 to degrade tumor BCL-X_L protein levels demonstrated only modest BCL-X_L degradation in the KPCY xenografts and orthotopic models (**Fig. 6k and m, Supplementary Fig. 11f-g**). We hypothesize the efficiency of this combination would increase with more potent BCL-X_L degraders. To demonstrate this point, we tested the efficacy of BQ in combination with tumor-specific *BCL2L1* (BCL-X_L) knockout, in vitro and vivo (**Response Fig. 9**). Our results demonstrate that optimal depletion of *BCL2L1* in tumor cells enhances the combinatorial effects of DHODH and BCL-X_L targeting on tumor growth.

a - (also Supp Fig. 5a)**b - (also Supp Fig. 5b)****c - (also Supp Fig. 5c)****d - (also Supp Fig. 5d)****e - (also Supp Fig. 5e)****f - (also Fig. 6e)****g - (also Fig. 6f)**
Response Figure 9 BCL-XL KO and BQ treatment show additive effects *in vivo*. (a) Immunoblot analysis of BCL-XL in non-targeting cells and BCL-XL KO PaTu-8902 cells (The numbers represent individual clones derived from the cell pool following lentivirus infection and subsequent selection. BCL-XL KO clones 1-4 were derived from the BCL-XL sgRNA-1 targeting cell pool. BCL-XL KO clones 5-8 originated from the BCL-XL sgRNA-2 targeting cell pool). (b) Cell proliferation was monitored in real-time using the Incucyte® Live-Cell Analysis System and represented as relative growth normalized to day 0 for PaTu-8902 non-targeting cell and BCL-XL KO over 72 h. Non-targeting-1 and non-targeting-2 refer to clones 1 and 2 in (a) labeled as non-targeting. BCL-XL KO-1 and BCL-XL KO-2 correspond to clones 1 and 5 from (a) labeled as BCL-XL KO. Clones non-targeting -1 and BCL-XL KO-1 were used in panels (c), (d), (e) and (g). Error bars represent s.d. of three technical replicates (representative of two experiments). (c) Cell proliferation was monitored in real-time using the Incucyte® Live-Cell Analysis System and represented as relative growth normalized to day 0 for PaTu-8902 non-targeting cell and BCL-XL KO with or without 5 μM BQ over 72 h. Error bars represent s.d. of three technical replicates (representative of two experiments). (d) Real-time accumulation of Annexin V fluorescence in PaTu-8902 non-targeting cells and BCL-XL KO cells was monitored using the Incucyte system. Cells were labeled with Annexin V Red Dye and treated with the indicated concentrations of 5 μM BQ for 72 hours. Error bars represent s.d. of three technical replicates (representative of two experiments). (e) Immunoblot analysis of BCL-XL and cleaved PARP in non-targeting cells and BCL-XL KO PaTu-8902 with or without 5 μM BQ. (f) Experimental design of (g). (g) Tumor volume of PaTu-8902 BCL-XL KO tumors (non-targeting n=7, non-targeting + BQ n=7, BCL-XL KO n=7, BCL-XL KO + BQ n=9 per arm) after treatment with vehicle or 10 mg/kg BQ three times a week for 3 weeks. Tumors were measured twice a week. Error bars represent ± s.e.m. For all panels, statistical significance was determined by ordinary one-way ANOVA test. *p < 0.05, **p < 0.01, ***p < 0.001, ****p < 0.0001.

Minor:

- Some labels in Figures 1d and 1e are unclear (e.g., SLC29A1, SLC7A11, and UCK2 in Figure 1e). Please clarify whether these genes are upregulated or downregulated in the figures.

We have now fixed labels in **Fig. 1d-e (Response Fig. 10)**.

Response Figure 10 Volcano plot illustrates significant protein abundance differences in PaTu-8988T cells treated with 5 μ M BQ at 24h (a) and 7 days (b). Volcano plots display the $-\log_{10}$ (FDR) versus the \log_2 of the relative protein abundance of mean BQ to DMSO-treated samples. Red circles represent significantly upregulated proteins (\log_2 fold change ≥ 1 , FDR < 0.01), whereas blue circles represent significantly downregulated proteins (\log_2 fold change ≤ -1); data from 3 DMSO or 3 BQ-treated independent plates.

2. The authors could consider testing different tumor models, such as AML, to enhance the clinical potential of the combination therapy of BD and DT2216.

We thank the reviewer for appreciating the potential importance of our strategy and for suggesting testing the combination of BQ and DT2216 in other tumor models, specifically acute myeloid leukemia (AML). Targeting DHODH in AML induces differentiation (PMID: 27641501) and triggers apoptosis (PMID: 29880605; PMID: 35514210). Therefore, co-targeting pyrimidine biosynthesis and apoptosis in AML is a rational strategy. However, AML dependency on anti-apoptotic proteins differs significantly from that of PDAC. AML cells are generally more dependent on BCL2 than BCL-X_L for survival, as shown in several studies like Pan et al. (PMID: 24346116) and Konopleva et al. (PMID: 27520294) which demonstrate the efficacy of BCL2 inhibitors such as venetoclax in AML cells. In addition, Souers et al. (PMID: 23291630) emphasized the critical role of BCL2 over BCL-X_L in AML therapy and demonstrated the potent antitumor activity of venetoclax in BCL2-dependent AML cells.

There have been attempts to explore combinations co-targeting BCL2 and pyrimidine metabolism in AML by combining venetoclax (BCL2 inhibitor) with cytaradine (VIALE-C study, PMID: 36112968) or the cytidine analog azacytidine (VIALE-A, PMID: 38343151). These combinations have demonstrated promising efficacy in AML by leveraging the high dependency of most AML cells on BCL2 for survival. In contrast, AML subtypes with erythroid or megakaryocytic differentiation exhibit a higher dependency on BCL-X_L, making BCL-X_L inhibitors particularly effective in these cases (PMID: 36508699). However, the exploration of co-targeting BCL-X_L and pyrimidine metabolism in AML has not been explored. This is likely due to the broader dependency of most AML subtypes on BCL2 and MCL1 for survival, which reduces the therapeutic potential of strategies targeting BCL-X_L in the majority of AML cases (PMID: 33353284).

Given the distinct anti-apoptotic dependencies, we believe that expanding this study to include AML could obscure our focus on PDAC and its BCL-X_L dependency. Instead, we have now added a discussion acknowledging that different cancer types might exhibit varied dependencies on anti-apoptotic proteins, which would influence the therapeutic effectiveness of combination DHODH and BCL-X_L targeting beyond PDAC. This clarification will recognize the difference in anti-apoptotic dependencies and how that informs the potential applicability of our combination therapy across various cancers. This has been incorporated in the discussion, line 526-530:

“... These findings suggest an extra-reliance on BCL-X_L in PDAC in comparison to MCL1 and BCL2 and may in part explain the selective synergy of DHODH inhibition with BCL-X_L inhibition. Future studies should evaluate

the potential of this combination in other types of tumors accounting for differential reliance on anti-apoptotic dependencies.”

Reviewer #4 (Remarks to the Author):

We thank the reviewer for their contributions to the review process.

Reviewer #5 (Remarks to the Author):

In the manuscript by Zhang et al, the authors aim to identify mechanisms and potential therapeutic targets that contribute to the resistance to the inhibition of DHODH, a key enzyme for nucleotide synthesis and mitochondrial metabolism. The authors utilized unbiased and comprehensive approaches, including Integrated metabolomic, proteomic and CRISPR-mediated genetic screening in human PDAC cell lines, and revealed pathways critical of nucleotide salvage pathway and BCL-XL-mediated anti-apoptosis pathway for the adaptation to DHODH inhibition. The authors further demonstrated that pharmacological inhibition of DHODH and BCL-XL exhibits synergistic effect in inhibition PDAC growth both in vitro and in vivo. Overall the experiments are well-designed and well-executed. It not only offers molecular insight for the adaptive mechanisms to the inhibition of DHODH in PDAC, but also holds strong translational potential by providing preclinical evidence for combination therapeutic strategy. The overall manuscript is well-written and easy to follow. Nevertheless, a few concerns need to be addressed.

We thank reviewer #5 for this positive review.

1. Additional characterization of mitochondrial function following short-term and long-term inhibition of DHODH, such as OCR and membrane potential, will further enhance the molecular insight of the study.

We thank the reviewer for this excellent question. To investigate the effects of acute versus long-term DHODH inhibition on mitochondrial function, we performed new experiments to measure Oxygen Consumption Rate (OCR) and membrane potential of cells treated with DHODHi for 24 h or 7 days (**Response Fig. 2; Supplementary Fig. 1e-j**). As measured by SeaHorse, DHODHi impaired OCR at 24h but to a lesser extent than complex I inhibition with IACS-010759. This is in line with previous studies showing a role for DHODH in respiration (PMID: 38547260). However, long-term treated cells showed more profound inhibitory effects suggesting resistance to DHODHi is not related to reactivation of respiration in these cells.

An increase in mitochondrial membrane potential is considered an early event in apoptosis that may be independent from caspase activation, precedes outer mitochondrial membrane disruption, and is associated with increased ROS production (PMID: 9973403, 9393856). As DHODHi decrease GSH/GSSG ratios (**Fig. 1c**), indicative of a decrease in antioxidant capacity, we next measured effects of DHODHi on mitochondrial membrane potential by Tetramethylrhodamine methyl ester (TMRM) staining in PDAC cells. DHODHi induced an increase in mitochondrial membrane potential (**Response Fig. 2g-h; Supplementary Fig. 8a-b**) suggesting that DHODHi-induced mitochondrial dysfunction and hyperpolarization may sensitize PDAC cells to apoptosis inducers. Interestingly, BCL-X_L is known to prevent changes in mitochondrial membrane potential in response to stimuli (PMID: 21987637, 9393856); therefore, whether the increase in mitochondrial membrane potential is due to a direct effect of DHODHi on mitochondrial respiration or antioxidant function or is a downstream consequence of modulation in BCL-X_L levels is unclear and will be the subject of future investigation.

a - (also Supp Fig. 1e) b - (also Supp Fig. 1f) c - (also Supp Fig. 1g) d - (also Supp Fig. 1h)

e - (also Supp Fig. 1i) f - (also Supp Fig. 1j) g - (also Supp Fig. 8a) h - (also Supp Fig. 8b)

Response Figure 2 DHODHi impairs mitochondrial respiration and mitochondrial membrane potential (a-b) Oxygen Consumption Rate of PaTu-8988T and PaTu-8902 cells treated with BQ for 24 h or 7 days. The complex I inhibitor IACS-010759 was used as a positive control for impaired mitochondrial respiration. **(c-f)** Measurements of basal respiration and ATP production for PaTu-8988T and PaTu-8902 cells. Error bars represent s.d. of six technical replicates (one representative of three independent experiments). Significance was determined with t-test. **** p < 0.0001. **(g-h)** Mitochondrial membrane potential was measured in PaTu-8902 **(g)** and PaTu-8988T **(h)** after 24 h or 7 days of treatment with Brequinar by flow cytometry. Mean fluorescence Intensity (MFI) of TMRM (Tetramethylrhodamine methyl ester) was calculated by subtracting fluorescence of FCCP-treated cells, normalizing to mitochondrial content measured with MitoTracker Green and normalized to DMSO. Error bars represent s.d. of average of 6 independent experiments. Significance was determined by t-test. **p < 0.01, ***p < 0.001, **** p < 0.0001.

2. Currently, the molecular mechanisms underlying the synergy between DHODH inhibition and BCL-XL depletion is not thoroughly elucidated. How does DHODH inhibition promote the sensitivity to BCL-XL? Is DNA damage or ROS induction involved? Does DHODH inhibition affect cytochrome c release?

This is an excellent point that has been raised by multiple reviewers. To assess if BQ modulates overall apoptotic priming as well as dependencies on specific pro-survival proteins in PDAC cells, we performed flow cytometry-based BH3 profiling (**Fig. 4a-b**) that ultimately measures cytochrome c release. Although BQ did not significantly increase the priming of either cell line, BQ sensitized cells to BCL-X_L targeting in PaTu-8988T and BCL2 targeting in both cell lines as indicated by the increased cytochrome c release in response to BAD and HRK peptides. These findings suggest that combining BQ with apoptosis inhibitors may further amplify cytochrome c release.

Regarding a potential role for ROS induction by DHODH inhibition as part of the mechanism of apoptosis sensitization, our metabolomics data demonstrates a decrease in the GSH/GSSG ratio (**Fig. 1c**), suggesting a decrease in antioxidant capacity after BQ treatment. In fact, prior work demonstrates that DHODH inhibition correlates with increased ROS levels (PMID: **38207075**). Our new preliminary data demonstrating that DHODH induces mitochondrial hyperpolarization may also fit with a role of ROS in apoptosis sensitization; however, as described in detail below we believe the primary mechanism of apoptosis sensitization by DHODH inhibition is via downregulation of NFKB leading to a pro-apoptotic shift in expression of the apoptosis proteome (**Response Fig. 1-2: Fig. 4f-g, Supplementary Fig. 1e-j, Supplementary 8a-b**).

An increase in DNA damage after DHODH inhibition has been associated with nucleotide imbalance and apoptosis in IDH mutant gliomas (PMID: 35985343, PMID: 35985342). In our study, our proteome GSEA analyses demonstrate an increase in DNA repair pathways in PaTu-8902 cells at 24 h, a decrease in PaTu-8988T cells at 24h and an increase at 7 d only at low doses. Given these discordant results and the lack of DNA damage repair related co-dependencies in the CRISPR/Cas9 screen, we concluded that it is unlikely that DNA damage is a central mediator of DHODH inhibition sensitization to BCL-X_L targeting and as such have not further pursued this line of investigation in our study. However, we have now performed further experiments to define the mechanistic links between DHODHi and sensitization to apoptosis and we include here the response we provided to Reviewer 1 for ease of review:

For the revised manuscript, we have performed experiments to understand the mechanistic basis of this combinatorial effect. Taking into consideration Reviewer #1's comments and those of the other reviewers, we investigated two hypotheses: 1) that DHODH inhibition increases sensitivity to BCL-X_L targeting by altering the activity of a transcriptional regulator of apoptosis genes, and 2) that DHODH inhibition alters mitochondrial function thereby enhancing apoptosis sensitivity.

In our original manuscript, we showed that DHODH inhibition (DHODHi) downregulated protein-level expression of BCL-2-family proteins as determined by quantitative proteomics and immunoblotting (**Fig. 4c, Supplementary Fig. 8c-d, Supplementary Table 2**). To understand if the changes at the proteome level correlated with transcriptional responses, in new revision experiments, we quantified expression of apoptosis-related genes by qRT-PCR (**Response Fig. 1a; included in the manuscript as Fig. 4f**). Consistent with protein-level expression changes, PDAC cells treated with DHODHi downregulated expression of anti-apoptotic genes (*BCL2L1* (*BCL-X_L*), *MCL1* and *BCL2*) and upregulated expression of pro-apoptotic genes (*BCL2L11* (*BIM*)).

Response Figure 1 DHODHi modulates expression of apoptotic genes by impairing activation of the NF-κB pathway. (a) Apoptosis regulatory gene expression assessed by qRT-PCR in PaTu-8902 and PaTu-8988T cells treated as indicated (5 μM BQ, 24 hours). Expression levels are normalized to *GAPDH* and presented as mean ± s.d. of 3 independent replicates (representative of three independent experiments). Significance determined by t-test, ****p<0.0001. (b) Immunoblot analysis of p65 (RelA subunit of NF-κB) and p105 (precursor of the NF-κB p50 subunit) activation in lysates from PaTu-8902 and PaTu-8988T cells treated for 24 and 48 hours with BQ (5 μM).

These data suggest that the pro-apoptotic shift induced by DHODHi may be regulated at the transcriptional level. We investigated whether DHODH inhibition has been previously linked to transcription factor pathways that regulate apoptosis genes. Prior research demonstrated that DHODH inhibition in the liver of mice downregulated NF-κB activity, a known pro-survival transcription factor (PMID: 15455409, PMID: 9973483). Similarly, patients with Miller Syndrome, caused by mutation of DHODH, have decreased NF-κB pathway activity (PMID: 19915526). NF-κB is a pro-survival effector that increases expression of anti-apoptotic genes, including *BCL-X_L*, *MCL1*, and *BCL2* (PMID: 32231206; PMID: 29379212). Conversely, downregulation of NF-κB is associated with increased sensitivity to apoptosis (PMID: 16751281).

Interestingly, our GSEA proteome analysis of BQ-treated PDAC lines demonstrated significant downregulation of pathways associated with TLR-related innate immunity and inflammation (**Fig. 1f, Supplementary Fig. 2e, Supplementary Table 3**), which activate or are regulated by NF- κ B transcription factor activity, respectively. To directly evaluate NF- κ B pathway activity in response to DHODHi, we analyzed phosphorylation of p65 and p105, key proteins in the canonical and non-canonical NF- κ B pathways, respectively. PDAC cells treated with DHODHi demonstrated a decrease in phosphorylation of both p65 and p105, consistent with decreased activation of the NF- κ B pathway (**Response Fig. 1b; Fig. 4g**). Based on these findings, we propose a model in which DHODHi suppresses NF- κ B activity, leading to a pro-apoptotic transcriptional response. This transcriptional shift downregulates anti-apoptotic genes, such as *BCL-X_L*, and upregulates pro-apoptotic genes, sensitizing PDAC cells to *BCL-X_L* inhibition and apoptosis induction.

To evaluate the hypothesis that DHODH inhibition increases sensitivity to *BCL-X_L* targeting by affecting mitochondrial function and thereby mitochondrial apoptosis priming, we performed new experiments to measure Oxygen Consumption Rate (OCR) and membrane potential of PDAC cells treated with DHODHi (**Response Fig. 2; Supplementary Fig. 1e-j**). As measured by Seahorse, DHODHi impaired OCR at 24 h but to a lesser extent than complex I inhibition with IACS-010759. This is in line with previous studies showing a role for DHODH in respiration (PMID: 38547260). Long-term DHODHi treatment demonstrated more profound inhibitory effects suggesting resistance to DHODHi is not related to reactivation of respiration in these cells.

An increase in mitochondrial membrane potential is considered an early event in apoptosis that may be independent from caspase activation, precedes outer mitochondrial membrane disruption, and is associated with increased ROS production (PMID: 9973403, 9393856). As DHODHi decreased GSH/GSSG ratios (**Fig. 1c**), indicative of a decrease in antioxidant capacity, we next measured effects of DHODHi on mitochondrial membrane potential by Tetramethylrhodamine methyl ester (TMRM) staining in PDAC cells. DHODHi induced an increase in mitochondrial membrane potential (**Response Fig. 2g-h; Supplementary Fig. 8a-b**) suggesting that DHODHi-induced mitochondrial dysfunction and hyperpolarization may sensitize PDAC cells to apoptosis inducers. Interestingly, *BCL-X_L* is known to prevent changes in mitochondrial membrane potential in response to stimuli (PMID: 21987637, 9393856); therefore, whether the increase in mitochondrial membrane potential is due to a direct effect of DHODHi on mitochondrial respiration or antioxidant function or is a downstream consequence of modulation in *BCL-X_L* levels is unclear and will be the subject of future investigation.

a - (also Supp Fig. 1e)

b - (also Supp Fig. 1f)

c - (also Supp Fig. 1g)

d - (also Supp Fig. 1h)

e - (also Supp Fig. 1i)

f - (also Supp Fig. 1j)

g - (also Supp Fig. 8a)

h - (also Supp Fig. 8b)

Response Figure 2 DHODHi impairs mitochondrial respiration and mitochondrial membrane potential (a-b) Oxygen Consumption Rate of PaTu-8988T and PaTu-8902 cells treated with BQ for 24 h or 7 days. The complex I inhibitor IACS-010759 was used as a positive control for impaired mitochondrial respiration. **(c-f)** Measurements of basal respiration and ATP production for PaTu-8988T and PaTu-8902 cells. Error bars represent s.d. of six technical replicates (one representative of three independent experiments). Significance was determined with t-test. **** $p < 0.0001$. **(g-h)** Mitochondrial membrane potential was measured in PaTu-8902 **(g)** and PaTu-8988T **(h)** after 24 h or 7 days of treatment with Brequinar by flow cytometry. Mean fluorescence Intensity (MFI) of TMRM (Tetramethylrhodamine methyl ester) was calculated by subtracting fluorescence of FCCP-treated cells, normalizing to mitochondrial content measured with MitoTracker Green and normalized to DMSO. Error bars represent s.d. of average of 6 independent experiments. Significance was determined by t-test. ** $p < 0.01$, *** $p < 0.001$, **** $p < 0.0001$.

Finally, we explored whether co-dependency with BCL-X_L may be related to the disruption of mitochondrial respiration by DHODHi⁸. IACS-010759 in combination with DT2216 showed no additive effects on growth or Annexin V staining (**Response Fig. 3a-d; Supplementary Fig. 7i-l**) suggesting DHODHi effects on mitochondrial respiration alone are not responsible for sensitization to BCL-X_L inhibition.

Response Figure 3 Complex I inhibitor does not recapitulate the pro-apoptotic phenotype induced by DHODHi (a-b) Percentage cell viability of PDAC cells after treatment with increasing concentrations of IACS-010759 with DT2216 for 5 days. IC50 values are shown for a representative experiment out of two independent experiments. **(c-d)** Real-time accumulation of Annexin V fluorescence in PaTu-8902 and PaTu-8988T cells was monitored using the Incucyte system. Cells were labeled with Annexin V Red Dye and treated with DT2216 (2 μM) or IACS-010759 (10 nM) alone or in combination for 72 hours. Error bars represent s.d. of three technical replicates (representative of two experiments). Significance was determined using ordinary one-way ANOVA. **** $p < 0.0001$.

We have summarized these new findings in the discussion as follows:

“We propose a model in which pyrimidine synthesis inhibition suppresses NF-κB activity, leading to a pro-apoptotic transcriptional response. This transcriptional shift downregulates anti-apoptotic genes, such as BCL-X_L, and upregulates pro-apoptotic genes, sensitizing PDAC cells to BCL-X_L degradation and promoting apoptosis. A similar profile of anti-apoptotic protein alteration was previously demonstrated in response to gemcitabine suggesting a conserved response to nucleotide metabolism targeting³². Functionally, alteration in

anti-apoptotic proteins in PDAC, whether by DHODH inhibition or gemcitabine, conferred increased sensitivity specifically to BCL-X_L targeting. Given DHODHi is also associated with an increase in mitochondrial membrane potential, it is possible that in addition to altering NF-κB pathway activity to increase apoptosis sensitivity, DHODHi also primes cells for apoptosis by inducing mitochondrial dysfunction linked to ROS accumulation and mitochondrial hyperpolarization. Whether the increase in mitochondrial membrane potential is due to direct DHODHi effects on mitochondrial respiration and antioxidant function or it is a downstream consequence of modulation in BCL-X_L⁵⁵ will require further investigation.”

3. The PDAC cells exhibit unique dependency on BCL-XL, but not BCL2 or MCL. What’s the impact of BCL-XL, BCL2 or MCL depletion on cytochrome c release and caspase activation?

Based on our BH3 profiling results (**Fig. 4a-b**), we observed that in the PaTu-8902 cell line (DMSO-treated samples), BCL2 inhibition led to a greater release of cytochrome c compared to BCL-X_L and MCL1 targeting. Conversely, in the PaTu-8988T cell line, BCL-X_L inhibition resulted in the most cytochrome c release, followed by MCL1, and then BCL2, suggesting a greater dependency on BCL-X_L for preventing apoptosis. DepMap data show that PDAC cells generally exhibit higher dependency on BCL-X_L, followed by MCL1 and then BCL2, underscoring the critical role of BCL-X_L in pancreatic cancer cell survival (**Supplementary Fig. 9e**).

4. Additional genetic studies showing the synergy between BCL-XL depletion and DHODH inhibition are preferred.

We thank the reviewer for this excellent suggestion. We have now generated BCL-X_L knockout clones using CRISPR/Cas9 genome editing in PaTu-8902 cells (**Response Fig. 9a; Supplementary Fig. 5a**). While BCL-X_L KO alone did not significantly impact cell growth in vitro (**Response Fig. 9b; Supplementary Fig. 5b**), treatment with BQ demonstrated a significant decrease in cell proliferation (**Response Fig. 9c; Supplementary Fig. 5c**) and increased Annexin V staining compared to PaTu-8902 cells transduced with a non-targeting sgRNA (sgControl) (**Response Fig. 9d-e; Supplementary Fig. 5d-e**). These results are consistent with our data using BCL-X_L inhibitors and degraders (**Supplementary Fig. 9a-d**). To assess the potential for tumor-specific targeting of BCL-X_L in vivo, we implanted sgControl (non-targeting) and BCL-X_L KO cells in mice and treated with vehicle or BQ. BCL-X_L KO tumors grew slightly slower compared to non-targeting tumors (**Response Fig. 9f-g; Fig. 6e-f**). BQ-treated BCL-X_L KO tumor-bearing mice demonstrated significant tumor growth inhibition compared to BQ-treated sgControl tumor bearing mice (**Response Fig. 9f-g; Fig. 6e-f**). Importantly, mice maintained stable body weight throughout treatment (**Supplementary Fig. 10b**). Additionally, BQ treatment in mice with BCL-X_L KO tumors did not lead to significant anemia or thrombocytopenia (**Supplementary Table 8**). Altogether, these new genetic studies demonstrate synergy between BCL-X_L depletion and DHODH inhibition validating our pharmacologic studies. Furthermore, this suggests that a strategy combining systemic DHODH inhibition with tumor-specific BCL-X_L targeting, such as with an antibody drug conjugate, offers a promising therapeutic approach with the potential for reduced toxicity.

a - (also Supp Fig. 5a)**b - (also Supp Fig. 5b)****c - (also Supp Fig. 5c)****d - (also Supp Fig. 5d)****e - (also Supp Fig. 5e)****f - (also Fig. 6e)****g - (also Fig. 6f)**
Response Figure 9 BCL-XL KO and BQ treatment show additive effects *in vivo*. (a) Immunoblot analysis of BCL-XL in non-targeting cells and BCL-XL KO PaTu-8902 cells (The numbers represent individual clones derived from the cell pool following lentivirus infection and subsequent selection. BCL-XL KO clones 1-4 were derived from the BCL-XL sgRNA-1 targeting cell pool. BCL-XL KO clones 5-8 originated from the BCL-XL sgRNA-2 targeting cell pool). (b) Cell proliferation was monitored in real-time using the Incucyte® Live-Cell Analysis System and represented as relative growth normalized to day 0 for PaTu-8902 non-targeting cell and BCL-XL KO over 72 h. Non-targeting-1 and non-targeting-2 refer to clones 1 and 2 in (a) labeled as non-targeting. BCL-XL KO-1 and BCL-XL KO-2 correspond to clones 1 and 5 from (a) labeled as BCL-XL KO. Clones non-targeting -1 and BCL-XL KO-1 were used in panels (c), (d), (e) and (g). Error bars represent s.d. of three technical replicates (representative of two experiments). (c) Cell proliferation was monitored in real-time using the Incucyte® Live-Cell Analysis System and represented as relative growth normalized to day 0 for PaTu-8902 non-targeting cell and BCL-XL KO with or without 5 μM BQ over 72 h. Error bars represent s.d. of three technical replicates (representative of two experiments). (d) Real-time accumulation of Annexin V fluorescence in PaTu-8902 non-targeting cells and BCL-XL KO cells was monitored using the Incucyte system. Cells were labeled with Annexin V Red Dye and treated with the indicated concentrations of 5 μM BQ for 72 hours. Error bars represent s.d. of three technical replicates (representative of two experiments). (e) Immunoblot analysis of BCL-XL and cleaved PARP in non-targeting cells and BCL-XL KO PaTu-8902 with or without 5 μM BQ. (f) Experimental design of (g). (g) Tumor volume of PaTu-8902 BCL-XL KO tumors (non-targeting n=7, non-targeting + BQ n=7, BCL-XL KO n=7, BCL-XL KO + BQ n=9 per arm) after treatment with vehicle or 10 mg/kg BQ three times a week for 3 weeks. Tumors were measured twice a week. Error bars represent \pm s.e.m. For all panels, statistical significance was determined by ordinary one-way ANOVA test. *p < 0.05, **p < 0.01, ***p < 0.001, ****p < 0.0001.

Minor comment

On page 7, references are missing for the following statement, 'In agreement with recent studies, we also identified an induction in HLA-I and proteins involved in antigen presentation'.

We thank the reviewer for this suggestion. We have now added a reference to this study (Mullen et al., eLife, 2024, PMID: 38973593).

Point-by-point Response to Reviews:

Below, we have copied each reviewer's comments (blue text) and present a point-by-point response (black text). For ease of review, we present data as "Response Figures" that integrate figure panels for the response to reviewers but also note where these were included in the revised manuscript. When appropriate, we have also included relevant text from the manuscript for ease of reference by the reviewers.

Reviewer #1 (Remarks to the Author):

The authors have addressed my concerns. I congratulate the authors on an excellent paper!

We thank the reviewer for their positive and constructive comments during the revision process.

Reviewer #2 (Remarks to the Author):

Authors have addressed my comments properly by additional experiments and discussions.

We thank the reviewer for their contributions to the revision process.

Reviewer #3, withdrawn

Reviewer #4, withdrawn

Reviewer #5 (Remarks to the Author):

The authors conducted extensive experiments and have successfully addressed all my comments. I have no additional comment.

We thank the reviewer for their constructive comments during the revision process.

Reviewer #6 (Replacing Reviewers #3 and #4, Remarks to the Author):

This is an interesting paper with potential important implications. As far as I can assess the authors have adequately answered most of reviewer 3 comments.

We thank the reviewer for their positive comments.

I think this paper should be accepted with a few minor revisions:

1) In the CRISPR screen BCL2L1 and MCL1 score as targets but as reviewer 3 mentions, BCL2L1 is down regulated following BQ treatment. This suggests that BQ works by activating apoptosis (but it is not a complete response) and that further inhibition of either MCL or BCL2L1

is enough to get apoptosis. This is very interesting since these two genes are typically regulated differently and mutually exclusive. What about MCL1 this should be highlighted also in the proteomic data. Since BCL2L1 and MCL1 are very well-known co-dependencies, the authors should comment on the possibility of a trio combination or that MCL1 is restricting BCL2L1 suppression.

We have now highlighted MCL1 in the volcano plots included in **Fig. 1d-e** and **Supplementary Fig. 2c (Response Fig. 1)**. MCL1 was not identified in the dataset included in **Supplementary Fig. 2d**. This data is also presented in **Supplementary Fig. 8c-d**. As noted previously, DHODHi downregulate MCL1 expression at the protein level and the mRNA level (**Fig. 4f**) in both PaTu-8988T and PaTu-8902 cells.

Response Figure 1 MCL1 changes in proteomic. (a-b) Volcano plot for protein abundance in PaTu-8988T cells treated with 5 μ M BQ at 24 h (a) and 7 days (b). Plots display $-\log_{10}$ (FDR) versus \log_2 of relative protein abundance of mean BQ to DMSO-treated samples. Red circles: significantly upregulated proteins (\log_2 fold change ≥ 1 , FDR < 0.01); blue circles: significantly downregulated proteins (\log_2 fold change ≤ -1 , FDR < 0.01); data from 3 DMSO or 3 BQ-treated independent plates. **(c)** Volcano plot illustrates significant protein abundance differences in PaTu-8902 cells treated with BQ 5 μ M at 24h. Volcano plots display the $-\log_{10}$ (FDR) versus the \log_2 of the relative protein abundance of mean BQ to DMSO-treated samples. Red circles represent significantly upregulated proteins (\log_2 fold change ≥ 1 , FDR < 0.01), whereas blue circles represent significantly downregulated proteins (\log_2 fold change ≤ -1 , FDR < 0.01). Data from 3 DMSO or 3 BQ-treated independent plates.

Second, the reviewer makes an excellent point regarding the potential for a triple combination with MCL1 and BCL2L1 inhibition. A decrease in MCL1 expression may, in part, further increase dependency on BCL-X_L for anti-apoptotic function suggesting a further inhibition of MCL1 activity would confer further potency of combination DHODHi and BCL-X_L inhibition. Interestingly, our western blot data also suggest a compensatory upregulation of MCL1 and BCL2 after DT-2216 treatment (**Fig. 4d-e**) alone or in combination with BQ that could limit efficacy of DHODHi and BCL-X_L inhibition combination. However, we have not explored this triple combination given reports of toxicity with MCL1 inhibitors and pre-clinical data suggesting lethal toxicity of combination MCL1 and BCL-X_L targeting. In particular, phase 1 clinical trials with an MCL1 inhibitor (AZD5991) have shown limited clinical activity in hematologic malignancies and resulted in cardiotoxicity that precluded further clinical development of these specific inhibitors (PMID: 39167622, PMID: 23788623). Other studies have shown significant hepatic toxicity when BCL-X_L was co-targeted with MCL1, resulting in lethality likely due to the critical cooperative role of MCL-1 and BCL-X_L in maintaining hepatocyte function (PMID: 19676108). Therefore, we have not explored this triple combination strategy either from a mechanistic or translational perspective.

2) It is very critical that the raw read counts from the CRISPR screen are added as a supplementary table for re-analysis of this data.

We agree with the reviewer this is a critical point. We have now provided raw read counts for our whole-genome CRISPR screen as well as for the in vitro and in vivo mini-library screen in new **Supplementary Data 6**. In addition, we have also provided in the table all the sgRNA sequences included in both the commercial Brunello library as well as in our customized mini-library.